# Integrating Inflammatory and Epigenetic Signatures in IBD-Associated Colorectal Carcinogenesis: Models, Mechanisms, and Clinical Implications

**DOI:** 10.3390/ijms26199498

**Published:** 2025-09-28

**Authors:** Kostas A. Triantaphyllopoulos, Nikolia D. Ragia, Maria-Chara E. Panagiotopoulou, Thomae G. Sourlingas

**Affiliations:** 1Department of Biotechnology, School of Applied Biology and Biotechnology, Agricultural University of Athens, 75 Iera Odos Street, 11855 Athens, Greece; nikoliaragia@gmail.com (N.D.R.); mpanagiotopoulouu21@gmail.com (M.-C.E.P.); 2Laboratory of Nuclear Proteins and Chromatin Function, Institute of Biosciences and Applications, National Center for Scientific Research “DEMOKRITOS”, 15310 Agia Paraskevi, Greece; sourlin@bio.demokritos.gr

**Keywords:** inflammatory bowel disease (IBD), colorectal cancer (CRC), colitis-associated colorectal cancer (CAC), autophagy dysfunction, epigenetic, DNA methylation, histone modifications, non-coding RNAs (ncRNA), pro-inflammatory cytokines, translational biomarkers

## Abstract

The rising global prevalence of inflammatory bowel diseases, including Crohn’s disease and ulcerative colitis, is paralleled by an increased risk of colitis-associated colorectal cancer. Persistent intestinal inflammation promotes genetic instability and epigenetic reprogramming within epithelial and immune cells, driving the multistep transition from inflammation to neoplasia. This review integrates human and preclinical model evidence with literature mining and bioinformatic analyses of genetic, epigenetic, and ncRNA data to dissect molecular mechanisms driving colitis-associated colorectal cancer from chronic inflammation. We highlight how pro-inflammatory cytokines (e.g., TNF-α, IL-6), oxidative stress, and microbial dysbiosis converge on key transcriptional regulators such as NF-κB and STAT3, inducing DNA methylation and histone modifications (e.g., H3K27me3); altering chromatin dynamics, gene expression, and non-coding RNA networks (e.g., *miR-21*, *MALAT1*, *CRNDE*); ultimately reshaping pathways involved in proliferation, apoptosis, and immune evasion. This review updates new potential associations of entities with these diseases, in their networks of interaction, summarizing major aspects of genetic and chromatin-level regulatory mechanisms in inflammatory bowel disease and colorectal cancer, and emphasizing how these interactions drive the inflammatory-to-neoplastic transition. By underscoring the reversibility of epigenetic changes, we explore their translational potential in early detection, surveillance, and precision epigenetic therapy. Understanding the interplay between genetic mutations and chromatin remodeling provides a roadmap for improving diagnostics and personalized treatments in inflammatory bowel disease-associated colorectal carcinogenesis.

## 1. Introduction

### 1.1. The Impact of Chronic Inflammation in Carcinogenesis

Inflammation is a physiological process triggered by the immune system in response to pathogens, inflammatory cytokines, harmful agents (e.g., toxins, chemicals, environmental factors), or physical injury, and is essential for host defense against invasive organisms [1,2,3]. It functions as an immediate protective response to limit infection and tissue damage [4]. Normally, the innate immune system reacts rapidly, and inflammation resolves once infectious or damaging stimuli are eliminated. The networks of inflammatory pathways are precisely regulated to avoid tissue injury; however, persistent immune activation can cause dysregulation and promote chronic inflammation, whereby prolonged inflammation has been closely linked to autoimmune diseases such as systemic lupus erythematosus (SLE) and rheumatoid arthritis (RA), neurodegenerative disorders, viral infections including COVID-19, and cancer [3]. The outcome depends on the balance and crosstalk between pro- and anti-inflammatory cytokines and chemokines, which determine whether the response remains beneficial and non-tumorigenic [4,5].

With respect to cancer, prolonged activation of inflammatory signaling results in chronic inflammation that can induce malignant cellular transformation [4]. Chronic inflammation can be tumor-promoting and is considered a hallmark of cancer. It plays a crucial role in tumor initiation, malignant transformation, invasion and metastasis [6,7,8]. Chronic inflammation predisposes patients to the development of cancer and facilitates practically all stages of carcinogenesis. The cytokines and chemokines produced by immune cells have both pro-tumor and anti-tumor roles, and their complex interaction determines the fate of carcinogenesis [8,9]. These pro-inflammatory cytokines produced as an outcome of the inflammatory processes lead to immunosuppression, promotion of angiogenesis, invasion and metastasis [10].

Chronic inflammation provokes alterations and/or dysregulation or dysfunction of molecular events leading to aberrant or altered signaling pathways. More specifically, chronic inflammation can provoke or induce the accumulation of mutations, aberrant changes in cellular, genetic and epigenetic processes leading to aberrant alterations in signaling events favoring neoplastic transformation and tumor initiation. Altered signaling involves the inactivation and activation of tumor suppression and oncogenic pathways, respectively [11]. Proliferation and survival of cancerous cells will be continuously enhanced by the formation of a tumor microenvironment (TME) that is a complex ecosystem of carcinoma-associated immune cells, signaling molecules, fibroblasts, blood cells (angiogenesis), extracellular matrix components, and other factors. Crosstalk between factors of this microenvironment can lead to the survival and growth of the malignant phenotype and the progression of the stages in the process of carcinogenesis, i.e., cancer initiation, promotion and progression at the expense of normal cells and tissues [5].

### 1.2. Inflammatory Bowel Disease

Inflammatory bowel disease (IBD), including Crohn’s disease (CD) and ulcerative colitis (UC), comprises chronic, relapsing inflammatory disorders of the gastrointestinal (GI) tract. They are remitting disorders which usually result in repeated abdominal pain, diarrhea, bloody purulent stool and weight loss. These disorders subsequently reduce the quality of life and increase the economic burden of IBD patients. The pathogenesis of IBD remains incompletely understood, but current data support the hypothesis that IBD is the result of a complex interplay of genetic predisposition, environmental factors and aberrant immune responses, such as an inappropriate gut mucosal response towards the constituents of the gut microbiota, which cross an impaired epithelial barrier [12,13]. The intestinal epithelium is a highly dynamic tissue whose functional integrity is indispensable for proper gut homeostasis. The intestinal epithelium lines the inner walls of the gastrointestinal tract and establishes the first line of defense from potential pathogens. A dysfunctional intestinal epithelium barrier can lead to severe dysregulation of gut homeostasis and allows microbial antigens to cross the barrier membrane, triggering inappropriate immune activation. This dysregulation, combined with gut microbiota dysbiosis, contributes to the chronic inflammatory state characteristic of IBD. The latter is a hallmark of the chronically relapsing exaggerated inflammation of IBD that involves drastic alterations in microbiome and epithelial barrier [14]. Recent research highlights the role of genetic loci, epigenetic modifications, and environmental triggers in disease development. Furthermore, chronic inflammation increases the risk of malignancies, notably colorectal cancer, necessitating vigilant monitoring and management. IBD is an extremely complicated chronic disease with unclear pathogenesis, and despite its rising incidence rates worldwide and extensive research, precise etiology is still unclear. Both genetic and environmental factors appear to be important in the development of IBD [15]. IBD-related disorders also confer a high risk of development of a number of malignancies, especially colorectal cancer CRC. With respect to cancer, it remains a major public health concern globally, with ≈20 million new cases and 9.7 million deaths worldwide in 2022, while CRC ranked third in incidence (≈1.93 million cases) and fourth in mortality (~0.90 million deaths) globally according to GLOBOCAN estimates (for details see Section 4.1), making research efforts for causes and its relationship with IBD and inflammation of major importance.

## 2. Infectious Agents, Immune Responses, and the Inflammatory Basis of Gastrointestinal Carcinogenesis

The gastrointestinal tract represents a critical interface between host and environment, where diverse microbial communities, dietary factors, and pathogens interact with epithelial and immune cells. Following infection, pathogens deploy strategies to navigate and manipulate the host environment, establishing niches through evasion of innate and adaptive immune responses, subversion of surveillance mechanisms, and modulation of host signaling pathways. This dynamic cascade ultimately determines whether an infection is transient, self-limited, or evolves into chronic inflammation that predisposes to disease.

The following section provides a concise overview of the hierarchical sequence of events that occur following the initial infection, focusing on the physiological and immunological mechanisms governing host–pathogen interactions. It outlines how pathogens navigate and manipulate the host environment to establish a successful niche. This dynamic cascade involves the pathogen’s strategies to subvert innate and adaptive immune responses, evade detection, and overcome host defense mechanisms, ultimately tipping the balance in favor of pathogen survival and persistence within the host.

### 2.1. Infection and Host Defense Mechanisms

#### 2.1.1. The Physiology of Infection: A Brief Overview

The term “infection” refers to any situation in which a microorganism, which is not a member of the local flora, settles and grows in a host, with or without damage to the host. Pathogens, organisms causing disease, initiate infection through mechanisms known as pathogenesis. However, infection is not synonymous with disease, as the presence of a microorganism, even a pathogen, does not always result in harm. Opportunistic pathogens, typically harmless members of normal microflora, can cause disease in hosts with compromised resistance, such as in cancer or AIDS [16]. The physiology of infection involves the body’s reaction to microbial invasions and the immune system’s role in combating pathogens. Upon exposure to pathogens such as bacteria and viruses, the adaptive immune system mounts an antibody response, with neutralizing antibodies that block pathogen entry or flag them for elimination (binding) by immune cells. At the same time, pathogens have evolved diverse immune evasion strategies, including antigenic variation, decoy epitopes, interference with antibody function, and secretion of immunosuppressive surface proteins—all employed to avoid detection and neutralization. Thus, the host response and pathogen countermeasures engage in a complex evolutionary arms race [17].

#### 2.1.2. Infection Stages

The initial stages of infection involve pathogen entry through routes such as the respiratory and gastrointestinal tracts, or skin and mucous membranes. Mechanisms include inhalation, ingestion, direct contact, or vector-borne transmission. The respiratory system is a common entry point, with pathogens spread through coughing, sneezing, or talking [18]. The digestive system allows entry via contaminated food, water, or direct deposition of agents. Pathogens can also penetrate the body through skin-to-skin contact, mucous membranes, or insect bites. The host body counters these invasions with barriers like skin, mucous membranes, and stomach acidity. However, pathogens evade these defenses through mechanisms like surface proteins and enzymes. Once inside, pathogens attach to host cells, avoiding immune detection [17,18].

#### 2.1.3. The Immune Response

The innate immune system serves as the first line of defense, offering non-specific protection through cells like macrophages, neutrophils, monocytes, natural killer cells, dendritic cells, innate lymphoid cells (ILCs), a recently recognized family of lymphocyte-like cells that regulate mucosal immunity and inflammation, together with molecular components like cytokines and complement proteins [19].

These cells can recognize conserved pathogen-associated molecular patterns (PAMPs) through pattern recognition receptors (PRRs) present on immune and epithelial cells. Examples of PAMPs include lipopolysaccharides (LPS) in Gram-negative bacteria, flagellin in bacterial flagella, and viral double-stranded RNA. Cytokines such as interferons, interleukins, and tumor necrosis factors also play vital roles in immune responses [19].

Phagocytosis, a critical innate mechanism, involves specialized cells like monocytes, neutrophils, and macrophages engulfing extracellular material, including pathogens [19]. Signals from the Toll-like receptors (TLRs) further activate innate immunity, with specific TLRs recognizing distinct bacterial components, e.g., TLR2 binds peptidoglycan, TLR4 recognizes LPS, and TLR5 detects flagellin. TLRs trigger signaling pathways involving nuclear factor kappa-light-chain-enhancer of activated B cells (NF-kB), mitogen-activated protein kinases (MAPKs), and interferon regulatory factors (IRFs), leading to the production of pro-inflammatory cytokines and chemokine family members [20].

Adaptive immunity, mediated by T and B lymphocytes, is a highly specific response against pathogens and retains immunological memory for rapid reactivation upon re-exposure. Unlike innate immunity, the adaptive arm engages only upon antigen detection, providing targeted and efficient host defense [21]. Pathogens and their metabolites, including lipopolysaccharide (LPS) and short-chain fatty acids (SCFAs), serve as pathogen-associated molecular patterns (PAMPs) or damage-associated molecular patterns (DAMPs). These molecules bind to pattern recognition receptors (PRRs), initiating downstream signaling cascades that orchestrate protective inflammation. While inflammation is crucial for pathogen clearance, uncontrolled or chronic inflammation—triggered by overactivation of PRRs by microbial molecules such as LPS or SCFAs—can contribute to the development of chronic diseases and potentially fatal immunopathology [22].

#### 2.1.4. Factors Influencing the Outcome of the Infection

The outcome of an infection is influenced by factors such as pathogen virulence, host immune response, age, health status as well as environmental conditions. Infectivity varies among pathogens, with highly virulent agents causing severe disease and less virulent ones often leading to mild or asymptomatic infections [16]. A robust immune response can clear pathogens and prevent disease, whereas a weakened response may result in persistent infections, particularly in immunocompromised individuals, such as those with HIV or undergoing immunosuppressive therapy [22]. Host factors like age and malnutrition also affect susceptibility to severe infections, with poor nutrition impairing immune functions.

Environmental factors, including seasonal variations, influence infectious disease dynamics. For instance, influenza is more prevalent in cold and dry conditions. Stress, genetics, diet and underlying health conditions further modify infection outcomes [23]. Immune dysfunctions, such as allergies, autoimmunity, and immunodeficiencies, occur upon immune system deregulation, highlighting the need to understand infection physiology for effective prevention and treatment strategies [22].

#### 2.1.5. Physiological Human-Bacteria Interactions

The human microbiome comprises trillions of microorganisms residing on or within the human body, forming a symbiotic relationship with the host [24]. These microbes, collectively known as the normal microflora, adapt to the host’s physiology to create body-specific ecosystems. Introduced at birth, the microbiome includes skin, oral, and gastrointestinal microbiota, with intestinal microbes performing critical metabolic functions, such as producing vitamins B12 and K, which humans cannot synthesize [24].

The microbiome supports host health by producing beneficial compounds and inhibiting harmful microorganisms, while the host provides microenvironments for microbial growth. Dysbiosis, or an imbalance in the microbiome, has been associated with diseases like inflammatory bowel disease, diabetes, asthma, and cancer, highlighting the critical role of a balanced and stable microbiome in maintaining overall health [24], as discussed below.

### 2.2. Pathogen Evasion and Immune Dysregulation

#### 2.2.1. Antigens and Subversion of Immune Response

Bacterial pathogens have developed sophisticated mechanisms to evade phagocytosis and survive within host cells. These pathogens, in order to counteract phagocytes, employ strategies such as escaping the phagosome, blocking phagosome-lysosome fusion or surviving within phagolysosomes. For example, *Shigella* sp., *Listeria monocytogenes* and some *Rickettsia* species secrete lysins to dissolve vacuolar membranes, facilitating their escape [25].

Many intracellular pathogens reside in modified vacuoles that deviate from typical microbicidal phagolysosomes. These modifications often rely on type III and IV secretion systems to disrupt host vesicle trafficking. For example, *Legionella pneumophila* employs the Dot/ICM system, where its effector RalF activates ARF-1 GTPase, facilitating the pathogen’s intracellular survival. Similarly, *Salmonella* uses Spi-2 secretion to release effectors like SifA, altering vacuole composition. *Mycobacterium tuberculosis* (discussed in a later section), a highly successful pathogen, prevents phagosome acidification through surface glycolipids and carbohydrates [25]. Beyond evasion strategies, pathogens manipulate host inflammatory responses for survival. While these strategies block inflammatory signaling pathways, such as MAP kinase and NF-κB, others actively induce inflammation to recruit host cells that serve as proliferation sites. Recent findings also highlight how non-bacterial antigens, such as commensal and food-derived yeasts, can drive aberrant cytotoxic CD4^+^ TH1 responses in CD, illustrating how antigenic cross-reactivity fuels pathogenic immune activation [26]. Certain pathogens even secrete molecules to attenuate excessive inflammation, displaying their adaptive capabilities [25].

Bacterial pathogens have developed mechanisms to modulate the host immune response, including altering downstream Toll-like receptor (TLR) signaling and cytokine responses critical to innate immunity [25]. Endogenous antimicrobial peptides like defensins and cathelicidins play a vital role in controlling infections by disrupting bacterial membranes and regulating immunity. However, pathogens counteract these defenses by modifying surface structures to prevent peptide binding, encoding transport systems for peptide removal and secreting proteases to degrade them [25].

Phagocytes combat intracellular pathogens by producing oxygen reactive species, such as nitric oxide (NO), mediated by inducible nitric oxide synthase (iNOS). NO serves both as a microbicidal agent and a signaling molecule. Pathogens evade NO-mediated killing by detoxifying reactive nitrogen intermediates, repairing damage, or inhibiting iNOS activity, demonstrating their evolutionary adaptation to host defenses [25]. Along with bacterial strategies, fungal antigens (e.g., yeasts) can also act as potent modulators of host immunity by continuously selecting cross-reactive T cell clones in the inflamed gut, as recently shown in CD [26].

#### 2.2.2. Bacterial Infection and Immune Dysregulation

Bacterial infections pose a serious threat to human health, progressing through host cell adhesion, bacterial growth and multiplication, tissue penetration of the host, and toxin-induced damage. Host defenses counteract these stages, but excessive immune responses can worsen the outcome [27]. Furthermore, bacterial toxins trigger cytokine overproduction, leading to conditions like septic shock and toxic shock syndrome. Gram-negative endotoxins stimulate macrophages to release IL-1 and TNF-α, while staphylococcal exotoxins act as superantigens, inducing excessive cytokine release by T cells [27].

Sepsis progresses through hyperinflammatory and hypo-inflammatory phases. The initial “cytokine storm” leads to clinical symptoms, followed by monocyte dysfunction and lymphocyte apoptosis, impairing infection control. Certain bacteria, such as *Mycobacterium tuberculosis*, evade immunity by surviving intracellularly, causing chronic activation of CD4^+^ T cells, macrophage activation, and granuloma formation, often leading to tissue necrosis [27]. Deregulated and uncontrolled inflammation, while crucial for pathogen elimination, can escalate to systemic damage and septic shock, emphasizing the importance of balanced immune responses. In the following sections, we examine specific infectious diseases that serve as key examples of bacterial infections in the gastrointestinal tract, where the human immune system rarely succeeds in completely eliminating the pathogen, which remains active in most cases.

### 2.3. Infectious Agents Linked to GI Inflammation and Cancer

#### 2.3.1. *Mycobacterium tuberculosis* and GI Inflammation

Gastrointestinal tuberculosis (GITB) is a form of extrapulmonary tuberculosis that can affect any organ of the gastrointestinal tract [28,29,30,31]. GITB may occur from primary or secondary infection [32]. Primary infection consists of ingestion of food or milk that contains the bovine bacillus [33]. Secondary infection arises from swallowing of contaminated sputum in a patient with active pulmonary tuberculosis, through the spread of the bacteria via the bloodstream and lymphatic system or reactivation of latent tuberculosis infection (LTBI) [29,31]. Once in the gastrointestinal tract, Mycobacterium penetrates the mucosal layer and invades the intestinal submucosa [32,33]. The bacillus colonizes the Peyer’s patches and triggers an inflammatory response, which leads to the formation of granulomas [33]. These granulomas undergo caseous necrosis, releasing bacteria to the neighboring lymph nodes. While the granulomas grow in size, the bowel wall thickens and papillary elevations appear in the mucosa [33]. Consequently, mucosa becomes edematous and ulcerative, which can either progress to perforation or heal through fibrosis [28,33].

Tuberculosis (TB) can affect any part of the gastrointestinal tract from esophagus to the rectum [28,32]. The most common site of involvement is the ileocecal region due to the abundance of lymphoid tissue at this site [7,31,34]. The diagnosis of GITB is often delayed due to its varying and non-specific clinical manifestations, making it hard to distinguish from other intestinal diseases [28,30,35]. GITB in most patients results in chronic intestinal inflammation with the following symptoms: abdominal pain, fever, weight loss, loss of appetite, nausea/vomiting, diarrhea, change in bowel habits and blood in stool [28,29,30,35]. However, some patients may appear asymptomatic [35]. Clinical examination may reveal ascites, splenomegaly or a palpable abdominal mass in the lower quadrant area. If GITB is not treated promptly and properly, complications such as intestinal bleeding, fistula and perforation may occur [35].

#### 2.3.2. Other Bacterial Drivers of GI Carcinogenesis

Colonic tuberculosis is rare and can affect any part of the colon including the cecum, anus and rectum [31]. It appears that the cecum is the most common site of involvement [31]. Anal tuberculosis is also uncommon, accounting for only 1% of the abdominal tuberculosis cases [30]. The reported symptoms of colonic TB often include intestinal obstruction, perforation, fistulae, bleeding, fever, weight loss, diarrhea and the presence of a palpable abdominal mass [28,31]. Colonic perforation is a serious complication that requires surgical intervention [31]. Colonic TB is difficult to diagnose as it can mimic other abdominal diseases, tumors and Crohn’s disease [31,36]. Endoscopy, colonoscopy and CT scan are used to diagnose colonic TB, although the final diagnosis should be based on histological or bacteriological findings [29,37,38]. Differentiating colonic TB from Crohn’s disease is crucial because an immunosuppressive treatment for a TB patient may lead to detrimental effects, such as miliary TB [29,37]. It is also noteworthy to mention that colonic TB may mimic or masquerade as precancerous or cancerous states [39]. Furthermore, people with TB have increased risk of both pulmonary and gastrointestinal cancers. For example, there are reports of cancers developing in about 10% of gastric TB cases [40]. Therefore, it is very important for clinicians to keep in mind this association between TB and neoplastic lesions.

TB also affects mammals and is caused by bacteria that belong to the *Mycobacterium tuberculosis* complex (MTBC) [41]. Gastrointestinal tuberculosis in domestic animals such as cattle and goats can provoke detrimental economic and public health problems to the local communities [41,42]. Companion animals such as dogs have been reported to be infected with *Mycobacterium tuberculosis* and subsequently develop gastrointestinal tuberculosis [43,44,45]. Dogs are infected by *M. tuberculosis* by swallowing human sputum or ingestion of food that contains the tubercle bacilli, thereby the main infection site is the abdomen and intestine [44]. There are no reported cases of tuberculosis spreading from dogs to humans. Thus, the disease is thought to be anthropozoonosis [43]. Tuberculosis in birds caused by *M. avium* and *M. genavense* mainly affects organs like the spleen and intestine, and rarely involves the lungs. *M. avium* is also known to colonize the GI tract of HIV infected patients [46]. Granuloma formation in the gastrointestinal tract following a mycobacterial infection has been observed in a series of animal species such as dogs, rabbits (*Oryctolagus cuniculus*), birds, cattle and Kenyan sand boas (*Eryx colubrinus loveridgei*) [46,47,48]. Moreover, intestinal perforation has been reported in a free-ranging Australian Sea Lion (*Neophoca cinerea*) subsequent to *Mycobacterium pinnipedii* infection [49]. Infection with *M. bovis* in cattle may cause intermittent diarrhea and constipation which is considered the causative (bacterial) agent of tuberculosis in the cattle (known as bovine TB) (ICD-10 A16), although it can produce infection in other animals [41].

#### 2.3.3. The Involvement of Microbes in the Mechanisms of Carcinogenesis

Infections contribute significantly to human tumors, as mentioned in the previous subsection. Most of these infections are attributed to viruses, leading to the oversight of bacterial contributions. While bacterial infections are epidemiologically linked to certain cancers, inflammation resulting from these infections has been traditionally considered the primary cause of tumor formation. However, bacteria can directly manipulate host cells during their infection cycles, impacting cellular integrity and potentially contributing to cancer development [50,51]. Cancer progression involves genetic alterations disrupting normal cell growth and survival controls. Viral genomes found in tumors and epidemiological studies establish strong links between viruses and cancers, such as human papillomavirus with cervical cancer and hepatitis B/C viruses with liver cancer. These viruses are part of a broader microbiome that interacts with host cells to ensure their survival. Although microbial infections like bacteria, molds, and helminths do not leave genetically identifiable marks in host genes, strong links exist between these infections and cancers [52]. Notable examples include *Schistosoma haematobium* with bladder cancer, *Helicobacter pylori* (*H. pylori*) with gastric cancer, and chronic *Salmonella typhi* infections with gallbladder carcinoma.

Studies in animals highlight the carcinogenic effects of microbiota, with germ-free or antibiotic-treated models showing reduced tumor development, underscoring the role of the microbiome in cancer [52]. *H. pylori* is the most documented bacterium with epidemiological data linking it to carcinogenesis, although other bacteria have also been associated with human cancers through interactions within the human microbiome [50].

The bacterial protein CagL is a type IV pilus adhesin of *H. pylori* that ensures the attachment of *H. pylori* to gastric epithelial cells. Notably, CagL from *H. pylori*, binds to gastric epithelial cells and then controls a signaling cascade that increases gastrin secretion, resulting in hypergastrinemia, an important risk factor for the development of gastric adenocarcinoma [53]. Bacteria can interfere with p53 activities and DNA repair mechanisms, promoting DNA damage accumulation and tumor growth [50]. Animal studies show reduced tumor burden when gut microbiota are manipulated with antibiotics, emphasizing the potential role of bacteria in cancer [50]. Clinical studies link *Fusobacterium nucleatum* with colorectal cancer, *Chlamydia trachomatis* with cervical cancer, and mycoplasmas with prostate and colorectal cancer, as well as with non-Hodgkin’s lymphoma in HIV-seropositive subjects [50,51].

Mycoplasma infections, in particular, have been shown to inhibit p53 and cooperate with oncogenic Ras, leading to oncogenic transformation in vitro. This strongly suggests that they can be the leading candidate bacteria with oncogenic potential. Persistent mycoplasma infections can lead to decreased expression of tumor suppressors p53 and p21, causing pathological changes and potentially facilitating tumorigenesis. *Mycoplasma fermentans*, for example, induces chromosomal alterations leading to malignant properties [50].

Recent research on the tumor microbiome highlights the impact of bacteria on tumor progression and therapy. Bacteria employ immune evasion strategies and intracellular infection mechanisms to survive and propagate. For instance, a protein from *M. fermentans*, DnaK, impairs DNA repair by reducing PARP1 activity, promoting cellular transformation [50]. DnaK also interacts with Ubiquitin Specific Peptidase 10 (USP10), reducing p53-dependent antitumor functions and counteracting anticancer drugs reliant on p53. Phylogenetic analysis suggests a common mechanism of cell transformation among bacteria like *Mycoplasmas*, *Helicobacter pylori*, *Fusobacterium nucleatum*, and *Chlamydia trachomatis*.

It was shown that exogenous DnaK induces inappropriate protein phosphorylation, adding to current knowledge about the role of bacteria in the tumor microenvironment in dysregulating cellular functions to ultimately promote cancer progression. These findings indicate that bacteria with similar DnaK proteins might contribute to tumor progression and therapy resistance by altering DNA repair and anticancer drug actions [50]. Bacterial manipulation of host cells likely results in cancer as an unintended consequence of infection cycles, as cancer typically arises after the bacteria have left the host [52]. While viral proteins involved in carcinogenesis are well-documented, bacterial mechanisms remain less understood. However, similarities between cancer-associated bacteria and oncogenic viruses are becoming clearer, suggesting shared virulence pathways, meaning that bacteria also alter critical cellular proteins and DNA repair processes, leading to cancer [50,51]. Understanding these mechanisms could enhance knowledge of cancer origins and be of benefit to preventive, diagnostic, and therapeutic strategies [50].

#### 2.3.4. Effect of Bacterial Infection on Gastrointestinal Cancer

The human gastrointestinal tract (GIT) is a highly intricate system housing trillions of microorganisms, including bacteria, archaea, fungi, parasites, and viruses [24]. Among these, bacteria represent the predominant microflora colonizing the GIT. Cancers of the GIT are recognized as a significant global health challenge, with high incidence and mortality rates, as reported in the World Cancer Statistics 2018 [54].

Compelling evidence highlights the role of bacterial infections in the development and progression of various GIT diseases, including cancers. Additionally, emerging research suggests that the GIT microbiota plays a critical role in influencing tumor responses to anticancer therapies, such as conventional chemotherapy and molecularly targeted treatments. As a result, targeting the bacterial microbiota offers promising potential for the prevention and treatment of GIT cancers [55].

#### 2.3.5. Colorectal Cancer

Colorectal cancer ranks as the third most commonly diagnosed cancer in men and the second in women, with 1.8 million new cases and 881,000 deaths reported in 2018 [51,54]. Adenocarcinoma is the prevalent histopathological subtype. While the exact causes remain unclear, environmental factors such as smoking, diet, and lifestyle are known risk factors. Age increases CRC incidence, with certain genetic disorders like Adenomatous Polyposis Coli (APC) and family history being significant contributors. Conditions like ulcerative colitis and Crohn’s disease also elevate CRC risk, though 80% of cases are sporadic [55].

The human gut hosts over 500 bacterial species, predominantly anaerobes like *Bacteroides*, *Eubacterium*, and *Fusobacterium*, with the colon having the highest concentration. Facultative anaerobes like *Enterococci* and *Lactobacilli* form a smaller portion. Dysbiosis, an imbalance in the gut microbiome, is implicated in colon diseases, including CRC. Despite extensive research, the specific mechanisms by which intestinal flora induce CRC remain unclear [55].

Studies have highlighted the role of the gut microbiome in CRC. McCoy and Mason first linked enterococcal endocarditis with cecal carcinoma, suggesting *Streptococcus gallolyticus* (formerly *S. bovis*) as a contributing factor. A significant percentage of *S. gallolyticus* bacteremia patients also have CRC, with prevalence rates in CRC patients ranging from 33% to 100%, compared to 2.5% to 15% in the normal population. Animal studies show that *S. gallolyticus* increases proliferation markers and polyamines, with colonic adenomas observed in 50% of affected rats. Increased IL-8 production, promoted by *S. gallolyticus*, enhances free radical generation, aiding the neoplastic process. *S. gallolyticus* colonizes colonic tissues through collagen-binding proteins and histone-like protein A. Consequently, patients with *S. gallolyticus* bacteremia are recommended to undergo complete colonoscopy [55]. Collectively, these findings support the notion that chronic inflammation is not merely the milieu to cancer development but an active participant in tumor initiation and progression. Understanding these immune and cytokine networks is therefore pivotal for identifying preventive and therapeutic targets in inflammation-associated CRC.

### 2.4. Chronic Inflammation as a Driver of Colorectal Carcinogenesis

Chronic inflammation is a critical driver of colorectal carcinogenesis, particularly in the setting of inflammatory bowel diseases such as ulcerative colitis and Crohn’s disease. Patients with long-standing IBD exhibit a significantly increased risk of developing colitis-associated colorectal cancer (CAC), which is characterized by inflammation-induced genetic and epigenetic alterations in colonic epithelial cells [56,57]. Unlike sporadic CRC, where the adenoma—carcinoma sequence predominates, inflammation-driven CRC follows an inflammation—dysplasia—carcinoma pathway, rooted in persistent immune activation and tissue injury [58].

A hallmark of this inflammatory environment is the dysregulation of cytokines and immune signaling cascades. Pro-inflammatory cytokines such as interleukin-6 (IL-6), tumor necrosis factor-alpha (TNF-α), and interleukin-1β (IL-1β) are markedly elevated in the inflamed colonic mucosa and contribute to epithelial cell transformation. In addition to these cytokines, innate lymphoid cells (ILCs) represent key regulators of mucosal homeostasis and chronic inflammation. In particular, ILC1 and ILC3 subsets are tightly controlled by transcription factors such as T-bet (Tbx21) and RORγt (Rorc), which influence their plasticity and effector cytokine production. Dysregulation of this transcriptional balance has been implicated in sustaining mucosal inflammation and promoting CAC development [59].

These cytokines activate downstream transcription factors, notably nuclear factor-kappa B (NF-κB) and signal transducer and activator of transcription 3 (STAT3), which orchestrate the transcription of genes involved in cell survival, proliferation, angiogenesis, and immune evasion [6]. NF-κB plays a central role in sustaining chronic inflammation and promoting oncogenesis. Under normal conditions, NF-κB is held inactive in the cytoplasm by IκB proteins. However, in the presence of inflammatory stimuli—including cytokines, bacterial products, and oxidative stress—NF-κB is rapidly activated and translocated to the nucleus. There, it induces the expression of anti-apoptotic genes (e.g., *Bcl-xL*, *XIAP*), inflammatory mediators (e.g., COX-2, IL-6), and enzymes that contribute to genomic instability (e.g., iNOS, ROS-generating enzymes) [60,61].

Similarly, IL-6 activation of the JAK/STAT3 pathway further exacerbates carcinogenic signaling. Activated STAT3 enhances the transcription of *cyclin D1*, *c-Myc*, and survivin gene (*BIRC5* gene), thereby promoting cellular proliferation and inhibiting apoptosis [62]. Notably, the crosstalk between NF-κB and STAT3 creates a positive feedback loop, reinforcing a tumor-promoting inflammatory niche. Moreover, ILC-derived cytokines such as IFN-γ, IL-17, and IL-22 add another layer of regulation in this process, further shaping the inflammatory tumor microenvironment [59]. This loop not only supports early tumor development but also fosters immune tolerance and resistance to therapy.

In addition to epithelial alterations, the inflammatory microenvironment recruits various immune cells, including macrophages, neutrophils, and Th17 cells, which release additional cytokines and reactive oxygen species (ROS). ROS induce oxidative DNA damage, telomere shortening, and DNA methylation changes, contributing to genomic instability [63]. Increasingly, ILC2s have also been implicated in cancer biology, though their role appears context-dependent. In some solid tumors, ILC2s promote tumor growth by fostering type 2 cytokine environments, while in others they enhance antitumor immunity, suggesting a dual role that could influence prognosis and therapy responsiveness [64].

Over time, this inflammatory pressure leads to architectural and functional distortion of the mucosa, transitioning from chronic inflammation to low-grade dysplasia, then to high-grade dysplasia, and ultimately to the invasive carcinoma stage. The emerging understanding of ILCs—including ILC1, ILC2, and ILC3 subsets—suggests they may be integral players in this sequence, positioning them as potential diagnostic markers and therapeutic targets in inflammation-associated CRC [59,64].

### 2.5. Mechanistic and Historical Perspectives on Inflammation-Induced Cancer

The association between chronic inflammation and cancer has been recognized for centuries. In 1863, Rudolf Virchow proposed that recurrent inflammatory responses could drive tumor development after observing immune cell infiltration within cancerous tissues [65]. Building on this idea, Katsusaburo Yamagiwa demonstrated in 1915 that experimentally induced chronic inflammation could initiate tumor formation in animal models [66]. Later, Harold F. Dvorak described tumors as “wounds that do not heal,” emphasizing that the same molecular programs involved in tissue repair and regeneration may also underlie tumorigenesis [67,68]. Advances in molecular biology and genetically engineered mouse models have since clarified the cellular and molecular framework linking inflammation and cancer. This includes the diverse roles of immune cell subsets and complex signaling cascades governed by cytokines, chemokines, and growth factors [6]. A deeper understanding of these pathways is crucial for the development of effective therapeutic strategies. In the next subsection updated information is provided through statistics and epidemiology search, the severity of the most prevalent cancer incidents, the rise in their occurrence and death rates as recently recorded, and compared to the incidences of GI cancers internationally.

#### Inflammation-Induced Cancers Associated with the GI Tract

Studies suggest that around 20% of cancers are associated with chronic inflammation that is linked to different stages of oncogenesis: cellular transformation, tumor progression, invasion, angiogenesis and metastasis [5]. Approximately 15–20% of all cancer cases develop at the same tissue or organ site that previously had some type of serious infection and/or chronic inflammation. In such conditions, cancer-promoting inflammation is often established well before any tumor arises [69]. Notable examples include inflammatory bowel disease, chronic hepatitis, Helicobacter-associated gastritis, and Schistosoma-induced bladder inflammation, all of which markedly elevate the risk of developing colorectal, liver, gastric, and bladder cancers, respectively [11]. These particular cancers that are associated with inflammatory disorders are cancers of organs of the GI tract.

The GI tract, also known as the gut or the digestive tract, is where food and liquids travel through and are processed, i.e., swallowed, digested, absorbed and wastes expelled from the body. The GI tract is made up of the hollow organs (mouth, esophagus, stomach, small intestine, large intestine (bowel, colon), rectum and anus. The solid organs of the GI tract are the liver, pancreas and gallbladder.

Generally speaking, for GI tract cancers, i.e., colorectal cancer (cancers of the colon and/or rectum, CRC), liver cancer, pancreatic cancer and stomach cancer are leading causes of cancer-related deaths worldwide [70]. More specifically, worldwide, in 2022, lung, breast, and colorectal cancers represented the three most prevalent cancer types. According to the International Agency for Research on Cancer’s Global Cancer Observatory (IARC, WHO) https://gco.iarc.fr/en (data accessed 16 June 2025), ten major cancers together accounted for roughly two-thirds of all newly diagnosed cases and cancer-related deaths across 185 countries and 36 cancer types. Lung cancer topped the list, with 2.5 million new diagnoses (12.4% of global cases). Female breast cancer followed with 2.3 million cases (11.6%), and colorectal cancer ranked third with 1.9 million cases (9.6%). Prostate cancer contributed 1.5 million cases (7.3%), and stomach cancer 970,000 cases (4.9%). In terms of mortality, lung cancer was also the primary cause, responsible for 1.8 million deaths (18.7%). Colorectal cancer caused about 900,000 deaths (9.3%), liver cancer 760,000 (7.8%), breast cancer 670,000 (6.9%), and stomach cancer 660,000 (6.8%) [71]. The global re-emergence of lung cancer as the most frequently diagnosed malignancy is thought to reflect sustained tobacco consumption, particularly in Asia.

With respect to GI tract cancers (i.e., colorectal, liver, gall bladder, pancreas and stomach) only, CRC ranked first for new cases of cancers. Moreover, CRC ranked first for leading causes of cancer death, followed by liver and stomach cancer [72,73]. 1.6 million deaths by 2040 [74].

Moreover, though it is known that the risk of CRC increases with age, with most cases affecting people over the age of 50, the incidence of new cases and deaths of CRC in younger age groups under the age of 50 has been rising steadily. In fact, the disease has become a leading cause of cancer deaths for Americans 20 to 49 years old according to the National Cancer Institute, https://www.cancer.gov/types (data accessed 16 June 2025). This may be due to changes brought about by modern lifestyle, which add an increased burden to the risk factors already involved (e.g., infections/pathogens). These include diet and industrialized food (e.g., meat processed food, etc.) rather than home cooking, as well as other environmental risk factors (chemicals, toxins, newly evolved pathogens, atmosphere—atmospheric changes—e.g., increased UV irradiation, etc.). The environmental factors which may be causal elements in inflammation, combined with genetic predisposition, may partly explain the increasing incidence of early onset CRC [72,73].

All cancers may or may not have genetic and/or epigenetic predispositions, and all may or may not be inflammation-induced. However, data support that chronic inflammation can induce carcinogenesis in individuals with susceptibility to infection, which increases the cancer risk, but also in those without. Specifically, CRC can develop with genetic susceptibility only, also taking into account its location, the colon/bowel, which is prone to numerous inflammatory conditions; it is also a prime candidate to be induced by chronic inflammation without genetic risk. As such, CRC may be considered a typical inflammation-dependent cancer, and the risk of developing CRC increases in patients with IBD [75].

Figure 1 presents a schematic mechanism of inflammation–dysplasia–carcinoma sequence model, anchoring, (1) microbial triggers (e.g., dysbiosis, etc.), (2) immune cell recruitment (macrophages, T cells, etc.), cytokines (e.g., TNF-α, IL-6, etc.), (3) persistent immune activation and cytokine overproduction compromise the viability of intestinal epithelial cells (IECs) and impair their autophagy machinery. This is accompanied by the loss of goblet cells and disruption of their mucin-secreting function, which weakens the mucosal barrier and facilitates bacterial infiltration and tumor-promoting inflammation. The breakdown of goblet cell–mediated mucosal protection is a hallmark of both active IBD and the transition to colorectal neoplasia [76,77,78,79]. (4) Epigenetic marks leading to silencing of tumor suppressors or activation of oncogenes and (5) transition to cancer, as the main components leading normal colon to IBD and dysplasia and progressing to colorectal cancer.

In more detail, Figure 1 graphically assigns and emphasizes the importance of the following factors in the inflammatory signalling involvement towards colorectal carcinogenesis, as summarized below:

Initiating factors such as pathogenic bacteria (*Helicobacter pylori*, *Mycobacterium tuberculosis*) and gut microbiota dysbiosis activate host immune surveillance via pattern recognition receptors, promoting sustained immune activation. This leads to the secretion of pro-inflammatory cytokines, including TNF-α, IL-6, and IL-1β, which in turn activate intracellular signaling pathways (e.g., NF-κB, STAT3), maintaining a chronic inflammatory microenvironment within the colonic mucosa. Persistent inflammation results in accumulated genetic mutations and epigenetic reprogramming. Epigenetic alterations include promoter hypermethylation of tumor suppressor genes (e.g., *SEPT9*, *CDKN2A*), histone modifications such as H3K27 trimethylation and histone deacetylation, and dysregulated expression of non-coding RNAs. Upregulation of oncogenic microRNAs (e.g., *miR-21*, *miR-155*, *miR-214*) and aberrant expression of long non-coding RNAs (e.g., *HOTAIR*, *MALAT1*, *CRNDE*) [76] disrupt gene regulation, chromatin accessibility, and epithelial differentiation. Critically, the sustained inflammatory response impairs autophagy and apoptosis within intestinal epithelial cells (IECs), undermining mucosal homeostasis. Loss of autophagic control contributes to abnormal cell survival, dysregulated turnover, and unrestrained epithelial cell proliferation. Goblet cell dysfunction—characterized by reduced mucin production—further weakens the intestinal barrier, increasing susceptibility to microbial translocation and perpetuating the pro-tumorigenic immune cascade. These epithelial impairments have been increasingly implicated in IBD pathogenesis and the development of colitis-associated colorectal [76,77,78,79].

Figure 1 also depicts the stepwise events and molecules linking chronic intestinal inflammation to colorectal cancer (CRC). Key initiators include microbial pathogens such as *Helicobacter pylori* and *Mycobacterium tuberculosis*, as well as gut microbiota dysbiosis. These factors activate inflammatory pathways leading to sustained production of cytokines and transcription factors (*IL-6*, *TNF-α*, and *NF-κB* activation), resulting in a state of chronic inflammation. There are also highlighted key molecular biomarkers (e.g., *SEPT9* methylation, *miR-21* expression) and emerging therapeutic targets such as 5-ASA (anti-inflammatory agent), vorinostat (histone deacetylase inhibitor), and decitabine (DNA methyltransferase inhibitor), which intervene at various stages of the inflammation—epigenetics—tumorigenesis axis. Understanding this complex interplay provides a dynamic framework for novel diagnostic and potential therapeutic strategies in inflammation-induced colorectal cancer.

More importantly, Figure 1 also emphasizes how chronic inflammation and epigenetic dysregulation intersect to drive CRC progression and identifies critical intervention points for clinical application. Important insights into the complex mechanisms are highlighted below:Persistent inflammation promotes epigenetic changes, including:DNA methylation of tumor suppressor genes (e.g., *MLH1*, *CDKN2A/p16*).Histone modifications such as hypoacetylation and trimethylation of histone H3 on lysine 27 (H3K27me3).

Altered expression of non-coding RNAs (miRNAs and lncRNAs), which regulate key inflammatory and tumorigenic genes. These molecular alterations drive the transition states from a healthy colon epithelium to IBD, dysplasia, and ultimately colorectal carcinoma. Clinical markers that can be detected at transitional stages include SEPT9 gene methylation (plasma biomarker) and *miR-21* expression (in tissue or circulation), both of which have diagnostic and prognostic potential. Current therapeutic interventions are also illustrated:Anti-inflammatory agents, such as mesalamine (5-ASA) and infliximab, which reduce inflammatory cytokine activity.Epigenetic drugs, including decitabine (a DNA methyltransferase inhibitor) and vorinostat (a histone deacetylase inhibitor), are being explored for their potential to reverse aberrant epigenetic states in cancer and inflammation.

According to the Genetic Testing Registry (GTR) resource (NCBI), in inherited colon cancer, https://www.ncbi.nlm.nih.gov/gtr/tests/552303/ (data accessed 3 February 2025), the following genes have been involved:Tumor suppressors and mismatch repair genes: *APC* (5q22.2), *MLH1* (3p22.2), *MSH2* (2p21-16.3), *MSH6* (2p16.3), *PMS2* (7p22.1).Polymerases and modifiers: *POLD 1* (19q13.33), *POLE.*Other associated loci: *MUTYH* (1p34.1), *EPCAM* (2p21), *GREM1* (15q13.3).

As shown in Figure 1, the aforementioned genes highlight the hereditary predisposition to colorectal neoplasia, interacting with both environmental and epigenetic drivers. Arrows indicate multidirectional crosstalk between genetics, epigenetics, and inflammation, emphasizing the multifactorial nature of IBD-to-CRC transition and the importance of integrated, personalized interventions.

## 3. Animal Models of IBD and Colitis-Associated Colorectal Cancer (CAC)

A comprehensive understanding of the pathogenesis and progression of inflammatory bowel disease (IBD) and its transition into colitis-associated colorectal cancer (CAC) necessitates the use of various mouse models for IBD, UC, etc. These models are tailored to mimic different facets of human disease, including innate and adaptive immune dysfunction, epithelial barrier disruption, and inflammation-induced carcinogenesis. In addition, this is an overview of the experimental animal models that have been developed and utilized to investigate the pathogenesis, progression, and therapeutic responses associated with gastrointestinal cancers (GIC), with a focus on inflammation-associated colorectal cancer. It highlights the strengths and limitations of both chemically induced models (e.g., AOM/DSS), genetically engineered models (e.g., APC^Min/+^ mice), and xenograft systems. Emphasis is placed on how these models recapitulate key features of human disease—including tumor microenvironment, immune responses, and molecular alterations—making them indispensable tools for preclinical research into cancer biology, drug testing, and biomarker discovery. The section also outlines the translational relevance of these models for studying the interplay between chronic inflammation, epigenetic regulation, and gastrointestinal tumorigenesis. Below, Table 1 highlights a summarized overview of widely employed murine models used in IBD and colitis-associated colorectal cancer (CAC). It contains information about the model, the method, which is the immune system mainly involved, the advantages, limitations, and is supported by previous work (PMID).

### 3.1. Murine Models of Gastrointestinal Cancer (GIC) Through Pathogen Infection

*Bacteroides fragilis* (*B. fragilis*), though representing only 0.1% of the normal colonic flora, is present in 80% of children and adults. However, enterotoxigenic *B. fragilis* (ETBF) strains producing the metalloprotease fragilysin are elevated in stool and colonic mucosal tissues of CRC patients. *B. fragilis* disrupts cell–cell adhesion by cleaving E-cadherin, a suppressor of invasion [55]. In vitro studies demonstrated that *B. fragilis* toxin stimulates cell proliferation via the β-catenin pathway, leading to the transcription of oncogenes *c-MYC* and *cyclin D1*. Mutations in Adenomatous Polyposis Coli (APC) complex proteins that activate β-catenin signaling are linked to hereditary and sporadic CRC forms in humans. Clinical studies found higher expression of the enterotoxin gene in mucosal samples from CRC patients. ETBF induced CRC in Min mice through STAT3 activation and TH17 cell response, with tumor growth inhibited by blocking IL-17 and IL-23 receptors [55]. *Escherichia coli* (*E. coli*), part of the normal colonic flora, shows increased carriage in adenomas and carcinomas of CRC patients. *E. coli* produces cytotoxic necrotizing factor (Cnf), cytotoxic distending toxin (Cdt), and colibactin, a polypeptide genotoxin associated with CRC. *E. coli* strains from phylogenetic group B2 produce colibactin via the enzyme complex “PKS”. Animal studies showed *E. coli* with PKS enzymes induced sporadic CRC in mice, with colibactin promoting epithelial cell proliferation through DNA damage and genomic instability [55].

*Fusobacterium nucleatum* (*F. nucleatum*) is linked to colorectal adenomas and CRC, with higher levels in CRC tissues and stool samples compared to controls [80]. It is associated with high CRC mortality, low overall survival, and increased metastasis. *F. nucleatum* stimulates CRC expansion via the Fap2 protein, which interferes with the immune system’s antitumor activity. The virulence factor FadA mediates adhesion to E-cadherin, activates β-catenin signaling, and enhances inflammatory and tumorigenic responses. *F. nucleatum* promotes proliferation and invasion of CRC cell lines through TLR4 signaling, while there is also NF-κB stimulation, and increase in *miR-21* marker expression [55].

*Enterococcus faecalis* (*E. faecalis*) is a human pathogen found at higher levels in CRC patients’ stool samples. It generates reactive oxygen and nitrogen species (RONS), causing DNA breakage, mutations, and chromosomal instability, contributing to its oncogenic activity [55].

*Helicobacter pylori* (*H. pylori*) and its role in CRC are less clear, but statistically significant associations exist. *H. pylori* infection is increased in patients with colon cancer and adenomatous polyps. Cytotoxin-associated gene A (CagA)-positive or CagA seropositivity correlates with severe gastrointestinal disease and higher CRC risk [55]. Alterations in other bacterial species, such as *Bacteroides*/*Prevotella*, *Coriobacteridae*, *Roseburia*, and *Fusobacterium*, are noted in CRC patients. Studies suggest that miRNAs may influence gut microbes’ gene expression and growth, impacting cancer pathogenesis [55]. Gastrointestinal cancers have high incidence and mortality rates, with bacterial infections playing a significant role in their development [54,55].

### 3.2. Non-Infectious Animal Models for Gastrointestinal Cancer (GIC)

In addition to pathogen-induced models, several non-infectious animal models have been extensively used to study the initiation, progression, and treatment responses of gastrointestinal cancers, particularly colorectal cancer (CRC). These models broadly fall into three categories: chemically induced models, genetically engineered mouse models (GEMMs), and xenograft systems. Each model recapitulates different facets of human disease and serves unique experimental objectives.

#### 3.2.1. Chemically Induced Models: AOM/DSS

The azoxymethane (AOM)/dextran sulfate sodium (DSS) model is one of the most widely used systems for studying inflammation-associated colorectal carcinogenesis. AOM is a potent procarcinogen that induces DNA alkylation, resulting in O6-methylguanine (O6-meG) adducts and subsequent G:C → A:T transitions, often leading to activating mutations in oncogenes such as *Kras*. When combined with DSS—an irritant that induces colitis—the model mimics the pathophysiological features of colitis-associated CRC [81]. In the context of modeling inflammation-driven colorectal cancer, DSS (dextran sulfate sodium)-induced colitis and the AOM/DSS (azoxymethane combined with DSS) model remain widely adopted experimental systems due to their simplicity, reproducibility, and close histopathological resemblance to human disease. This model faithfully reproduces key steps of tumor development, including crypt abscesses, epithelial injury and acute inflammation in the distal colon, hyperplasia, inflammatory cell infiltration, particularly useful for studying ulcerative colitis-like damage, dysplasia, and adenocarcinoma formation. Furthermore, it allows investigation of molecular events such as cytokine signaling (e.g., IL-6, TNF-α) and epigenetic alterations, such as promoter methylation and histone modification [82].

In one of our studies pertinent to the AOM/DSS models, we leveraged publicly available RNA-seq datasets derived from these murine models—DSS-induced colitis and AOM/DSS-induced CAC—in comparison to human CRC, in which we found close similarities (Triantaphyllopoulos et al., unpublished results) (see more details in Section 7.5).

#### 3.2.2. Genetically Engineered Mouse Models (GEMMs)

GEMMs offer the advantage of dissecting the functional roles of specific genes implicated in GIC. The most common knockout (KO) genes used in the murine model of intestinal inflammation are *IL-10*, *IL-23R*, *CD4^+^CD25^+^*, *NOD2/CARD15*, *TGF-β1*, *RAG*, *ATG16L1*, *APC^Min/+^*, *IL-2*, *TNF-α*, *STAT3*, *NFκB*, *Muc2*, *IFN-γ*, *MyD88* and *TLR* [83]. Among these, the *APC^Min/+^* mouse is the most commonly used model for studying sporadic and familial adenomatous polyposis (FAP) [84]. These mice carry a heterozygous truncating mutation in the tumor suppressor gene, *Apc*, leading to constitutive activation of the Wnt/β-catenin signaling pathway, and develop multiple intestinal neoplasms spontaneously [85]. Although tumors predominantly arise in the small intestine, combinations with other mutations (e.g., *Kras*, *p53*, or *Smad4*) or with inflammatory agents can shift tumorigenesis toward the colon and more accurately reflect human CRC. Importantly, GEMMs enable time- and tissue-specific gene modifications using Cre-loxP technology, allowing precise modeling of multistage tumor development and microenvironmental interactions.

#### 3.2.3. Xenograft and Patient-Derived Xenograft (PDX) Models

Xenograft models, involving transplantation of human CRC cell lines into immunocompromised mice (e.g., *NOD*/*SCID* or nude mice), are widely used for preclinical drug testing and evaluation of tumor growth dynamics. Subcutaneous xenografts offer ease of monitoring tumor size, while orthotopic models—involving implantation into the cecum or colon—better simulate tumor microenvironment and metastatic spread. More recently, patient-derived xenografts (PDX) have gained popularity, as they preserve the genetic, epigenetic, and histopathological characteristics of the original human tumors, offering enhanced predictive value for personalized medicine approaches [86]. However, the lack of a functional immune system in these models limits their utility for immuno-oncology studies.

Animal models have played a pivotal role in dissecting the complex pathophysiology of inflammatory bowel disease (IBD) and its progression towards colitis-associated colorectal cancer (CAC). These animal models allow for controlled experimentation on genetic, environmental, and immunological contributors to disease progression; more importantly, they provide complementary systems for studying the complex interactions between genetic mutations, epigenetic changes, inflammation, and tumor progression. Their continued development and refinement remain essential for translational research aimed at identifying therapeutic targets and validating biomarkers for GICs.

Below, the sidebar infographic Figure 2 presents a concise schematic overview of the more detailed Table 1, for the commonly used murine models of IBD and colitis-associated colorectal cancer (CAC), including method names and supportive citations.

## 4. Overview of Chromatin and Epigenetic Modulations

Nuclear DNA is organized into chromatin, which consists of nucleic acids (genomic DNA and different types of RNAs); the histone proteins H2A, H2B, H3, H4 and H1; and non-histone chromatin-associated proteins [12,87,88,89]. The basic structural and functional unit of chromatin is the nucleosome. The nucleosome consists of a core histone octamer (two H2A-H2B dimers and one (H3-H4)_2_ tetramer) around which are wrapped 146 base pairs (approximately 1.65 turns of DNA). Histone H1 is found outside of the nucleosome on the linker DNA region and seals the entrance and exit of the DNA around the nucleosome [90,91]. All biological processes such as replication and transcription take place on the DNA template, which must be in an ‘open’ structural form so that proteins of the replication machinery and transcription factors, and other proteins involved in transcription can have access. Thus, chromatin and nucleosomal structure must be (and is) dynamic in order for proteins of the transcriptional machinery to have access or be blocked as necessary. Epigenetics is the study of heritable changes in gene expression and function without changes in DNA sequence [92]. Epigenetic mechanisms are responsible for the regulation of transcription, i.e., what genes are expressed, or dynamically have the structural potential to be expressed or what genes are silenced (permanently or temporarily). These epigenetic mechanisms are the histone post-translational modifications (PTMs), changes in the histone variant constitution of the nucleosome, DNA methylation, nucleosomal remodeling and positioning factors (activating complexes such as SWI/SNF). More importantly, interactions with proteins of the nuclear matrix (scaffold proteins) and regulation via long non-coding RNAs (lncRNAs), microRNAs (miR) and other non-coding RNAs complicate the picture of the multifactorial network of interactors that are involved in genomic regulation at the chromatin level [12,93,94,95]. These mechanisms of transcriptional regulation establish epigenetic heritable patterns of differential gene expression and silencing profiles from the same genome, which are cell-type specific. Cells can change these gene expression signatures in response to stimuli, such as the changing conditions due to changes in the micro and macro environments [12,95,96].

Post-translational modifications of the histone proteins (histones H1, H2A, H2B, H3, H4) take place mostly on their N-terminal tails, which protrude from the nucleosome. Notably, some histone modifications also occur on the C-terminal tails, which do not protrude from the nucleosome but are embedded inside the octamer core in the globular domain of the histone, e.g., H3K79 methylation [97]. These modifications are reversible reactions and include acetylation, methylation, phosphorylation, ubiquitination, poly(A)ribosylation and sumoylation, among other more recently identified histone modifications [12,93,98,99], which have not been thoroughly investigated (i.e., GlcNAcylation, citrullination, crotonylation and isomerization) [92]. They can function alone or in combination with other histone modifications. The latter has been referred to as the ‘histone code’. Two or more histone modifications, e.g., on the promoter of a gene, can either enhance, reduce/inhibit or alter the function of another histone modification. The ‘histone code’ is a hypothesis that states that DNA transcription is largely regulated by post-translational modifications to the histone proteins [100,101].

Histone modifications occur at specific amino acid residues. Histone acetylation is one of the most studied and also most prevalent histone modifications [4]. Acetylation occurs only on specific lysine residues of all histones. This modification reduces the positive charge of the histone lysine residues, thus weakening the DNA-histone interactions, establishing an ‘open’, permissive towards transcription, chromatin structure and/or a transcriptionally active chromatin landscape. Acetylation, in fact, is a prerequisite for the activation of gene expression. Acetylated chromatin is ‘poised’ chromatin, ready for transcription [102]. The enzymes responsible for the transfer of the acetyl group from acetyl-coenzyme A are the histone acetyltransferases (HATs—comprising at least six groups of acetyltransferases), and those responsible for their removal are the histone deacetylases (HDACs—comprising four families). The enzymatic activity of HATs and HDACs alters chromatin configuration so as to allow activation or inactivation of a gene, respectively. Histone methylation also occurs at specific amino acids (lysines 4, 36, 79 of histone H3 at active chromatin sites and lysines 9, 27, 20 of histone H3 at inactive chromatin sites and lysines 5 and 20 of histone H4) or arginines (arginines 2, 8, 17, 26 of histone H3 and 3 of histone H4). To increase the complexity, lysines may be mono-, di- or trimethylated, whereas arginine residues may be mono- or dimethylated (symmetric or asymmetric). Unlike acetylation, histone methylation does not alter the charge of the histone protein. A variety of enzymes catalyze the addition or removal of the methyl group: the histone methyltransferases (HMTs) and the histone demethylases (HDMs), respectively. Histone phosphorylation occurs on tyrosine, serine, and threonine residues located within the N-terminal tails of histones. In particular, modifications are found at serine 10 and 28, threonine 3, 6, 11, 45, and tyrosine 41 of histone H3, as well as serine 32 of histone H2B [4]. This process involves the transfer of a phosphate group from ATP to the hydroxyl moiety of the target amino acid, resulting in the accumulation of negative charges on histones, thereby weakening their interaction with DNA, facilitating an ‘open’ transcriptionally permissive chromatin structure. Protein kinases and phosphatases add or remove, respectively, the phosphate group from the histone proteins (as well as from many other cellular proteins) [4]. Histone ubiquitination can be found in all core histone subtypes. Most prominent are histone H2A ubiquitination on lysine 118 or 119 (H2AK118119/ub) and H2B lysine 120 (H2BK120ub), which account for 5–15% of H2A and 1% of H2B, respectively [103]. It is mediated by the sequential interactions of the E1, E2 and E3 ligase enzymes. Histone ubiquitination plays a role in chromatin compaction and transcriptional regulation and can also interact with other histone modifications. Similarly, the reactions leading to the aforementioned histone modifications are catalyzed by other modification-specific enzymes. Various histone modifications, alone or in combination, alter the three dimensional (3D) structure of the nucleosome and affect the transcriptional control of genes by inducing either an ‘inactive’ closed heterochromatin conformation, inaccessible to the transcriptional machinery, or an ‘active’ open euchromatin conformation [104,105,106,107,108,109] or a facultative heterochromatin conformation (forms the poised chromatin with the potential to become euchromatin). Notably, environmental factors can induce changes in histone modifications, thereby altering gene expression signatures.

DNA methylation is the covalent transfer of a methyl group to the carbon atom at position 5 of cytosine. This forms the 5-methylcytosine (5mC), which occurs most frequently at the dinucleotide CG [12,95,110,111]. DNA regions that are ≥200 bp long and show a CG:GC ratio ≥ 0.6 are defined as CpG islands [12,112]. Methylated DNA is a closed structure, and transcription factors cannot reach gene promoters. Genes in such methylated DNA are silenced [12,113]. CpG islands are dinucleotide repeats prevalent in mammalian genomes, typically unmethylated and associated with gene promoters located in genetic regulatory elements. DNA methylation typically begins at one end of CpG islands and extends into gene promoters and transcription start sites (TSS). This process modifies the three-dimensional conformation of DNA, restricting access to transcription factors and thereby leading to transcriptional repression through hypermethylation. Conversely, hypomethylation facilitates transcriptional activity, promoting gene expression [109]. The enzymes that catalyze the addition of methyl groups to DNA are carried out by a family of enzymes known as DNA methyltransferases (DNMTs), comprising DNMT1, DNMT2, DNMT3a, DNMT3b, and DNMT3L. DNMT1 catalyzes DNA methylation during DNA replication and cell division, DNMT3A/3B (de novo methylation) are responsible for methylation of DNA during development and differentiation, while DNMT3L is an “aide” in DNA methylation interacting with DNMT3A/3B to stimulate the de novo reactions, as it lacks the conserved catalytic domain, thus, is not directly involved in methylation [114]. Methyl groups are donated by S-adenosyl-L-methionine (SAM) and attached to cytosine residues within DNA [115]. DNA demethylation is mediated by the ten-eleven translocation (TET) family of enzymes, which hydroxylate the methyl group of 5-methylcytosine (5mC) to generate 5-hydroxymethylcytosine (5hmC) [12,95]. TET proteins can further oxidize 5hmC to 5-formylcytosine (5fC) and subsequently to 5-carboxycytosine (5caC). Both 5fC and 5caC can then be excised from DNA through the base excision repair pathway and substituted with unmethylated cytosine, restoring the original base sequence. TET enzymes have central roles in DNA demethylation required during embryogenesis, gametogenesis, memory, learning, addiction and pain perception [116,117]. Deregulation of DNMTs or demethylases can cause widespread cellular detrimental effects, leading to global and gene-specific hypomethylation, as well as regional hypermethylation, which is linked to cancer [118,119].

Epigenetic regulation can also involve noncoding RNAs (ncRNAs), which are RNAs that are not translated into proteins. The well-known microRNAs (miRNAs) and long noncoding RNAs (lncRNAs) are short molecules with a length of approximately 18–25 nucleotides, while lncRNAs are over 200 bases long, respectively [120]. Although long non-coding RNAs (lncRNAs) may span an open reading frame (ORF) and contain a single exon—a minimal distinguishing feature from other non-coding RNAs—they share similarities with protein-coding genes. However, lncRNAs are generally shorter, composed of fewer but longer exons, and exhibit low evolutionary conservation. This limited conservation complicates the identification of functional domains and hinders comparative studies across species, even when lncRNAs are located within highly conserved genomic regions.

Non-coding RNAs (ncRNAs) encompass several classes, including transfer RNAs (tRNAs), ribosomal RNAs (rRNAs), and small RNAs such as microRNAs, siRNAs, piRNAs, snoRNAs, snRNAs, and extracellular RNAs (exRNAs). They also include long ncRNAs (lncRNAs), long intergenic non-coding RNAs (lincRNAs), and circular RNAs (circRNAs), with well-characterized examples like Xist and HOTAIR. CircRNAs are single-stranded, closed-loop RNA molecules that are highly stable and evolutionarily conserved [76,121]. Structurally, four main types have been identified: exonic circRNAs (ecircRNAs), circular intronic RNAs (ciRNAs), exon–intron circRNAs (EIciRNAs), and intergenic circRNAs [122,123]. Functionally, circRNAs act as miRNA sponges, modulating gene expression by sequestering miRNA targets [124]. In addition, they can bind to RNA-binding proteins (RBPs) to regulate physiological processes [125] and may also serve as regulators of transcription [126]. Both microRNAs (miRNAs) and long non-coding RNAs (lncRNAs) act as post-transcriptional regulators and chromatin remodelers, respectively. These molecules can regulate gene expression by interfering with messenger RNA (mRNA) translation by way of degrading the mRNAs or through interactions with protein complexes involved in the regulation of gene expression [12,120,127]. Chronic inflammation modulates the expression of several oncogenic miRNAs, such as *miR-21*, which is overexpressed in IBD and CRC tissues and associated with inhibition of tumor suppressors like Programmed cell death protein 4 (PDCD4) and Phosphatase and Tensin homolog (PTEN) [128,129]. Micro RNA-21 is currently under investigation as a diagnostic biomarker and therapeutic target.

Similarly, inflammation-sensitive lncRNAs such as *Hox* transcript antisense intergenic RNA (*HOTAIR*) and Long Intergenic Non-protein-coding RNA, P53 induced transcript (*LINC-PINT*) participate in epigenetic gene silencing through interaction with histone-modifying complexes, contributing to the persistence of an oncogenic transcriptional landscape.

### 4.1. Integration of Epigenetic Alterations and Inflammatory Pathways

The requirement to expand our view of the contributors in carcinogenesis by integrating the epigenetic alterations and inflammatory pathways provides the foundation for the growing understanding of inflammation-driven epigenetic alterations, which has been conceptually summarized in Figure 1. The latter graphical annotation portrays the interconnected steps leading from microbial triggers and immune activation to tumorigenesis through layered epigenetic reprogramming. It also emphasizes how chronic inflammatory signaling not only reshapes immune and epithelial responses but also establishes persistent chromatin changes that underpin the dysplastic transformation of colonic mucosa. Moreover, the Mechanistic Pathway—From Inflammation to Colorectal Carcinogenesis shows that epigenetic changes form a crucial axis that links early inflammatory triggers—such as *Helicobacter pylori*, *Mycobacterium tuberculosis*, or gut dysbiosis—leading to sustained epithelial transformation. This proposed model illustrates how persistent cytokine signaling (e.g., IL-6, TNF-α, NF-κB) drives epigenetic remodeling, leading to progressive histopathological stages from normal mucosa to IBD, to dysplasia and to CRC.

### 4.2. Epigenetic Alterations in Inflammation-Associated Pathologies of the GI Tract—An Overview

A range of epigenetic mechanisms is implicated in the initiation, progression, and persistence of IBD, often activated by diverse environmental influences. Notably, three critical windows have been identified in which environmental exposures may predispose individuals to disease: (1) the prenatal stage, shaped by maternal lifestyle factors; (2) the early postnatal stage, coinciding with gut microbiota colonization; and (3) the period immediately preceding disease onset [130]. As already mentioned, chronic inflammation can promote the occurrence and progression of colorectal cancer, and epigenetic mechanisms, both inherited and acquired by environmental factors, participate in the transformation of inflammation into CRC.

In CRC, abnormal patterns of histone acetylation and methylation at specific residues have been identified, accompanied by widespread dysregulation of the enzymes responsible for these modifications. Mutations, deletions, or changes in expression levels can alter the activity of several histone-modifying proteins, underscoring the pivotal role of epigenetic regulators in CRC development. Their involvement in inactivation and activation of tumor suppressor genes and oncogenes, respectively, and their potential as biomarkers [131]. Moreover, in CRC, the commonly observed types of DNA methylation include hypermethylation of anti-oncogene DNA and hypomethylation of oncogene DNA [109]. Non-coding RNAs have also been found to be associated with the transformation of inflammation and the transition towards CRC. MicroRNAs have been found to be involved in the aforementioned, as well as in chemotherapeutic resistance [109], while lncRNAs have also been implicated with the transformation of chronic inflammation into CRC [109].

Moreover in GITB, it has been shown that non-coding RNAs have emerged as crucial regulators of various infectious diseases, including tuberculosis [132]. In patients with GITB, miR-375-3p expression levels were noted to be higher in the plasma but lower in the ileal/ileocecal tissue compared to those who suffered from Crohn’s disease [133]. To date, few studies have explored the role of ncRNAs in the development and progression of gastrointestinal tuberculosis. However, research has revealed that gut microbiota influences immunological responses to tuberculosis by regulating non-coding RNAs [134,135]. For example, a study conducted by Yang and colleagues revealed that *Bacteroides fragilis* regulates lncRNA CGB, which in turn modulates IFN-γ expression, enhancing anti-TB immunity [134].

The rising incidence of IBDs, their difficulties in diagnosis and treatment and their link to CRC, which is a high-risk cancer, both as to its occurrence, its increasing incidence in younger age-groups (under 50 years old) and its high rank in cancer deaths, underscore the need for a better understanding of the molecular mechanisms underlying these diseases and their association. At the molecular level, both genetic (gene mutations) and epigenetic alterations have been found or implicated in IBDs and in CRCs (with and without genetic predisposition) [13,70,92,109]. These epigenetic alterations can drive initiation and progression of the inflammatory, the precancerous and cancer state(s) by altering the gene expression profile(s) of noncancerous and cancer cells of the Tumor Microenvironment (TME) and elsewhere [70]. GI cancer syndromes can arise from germline (inherited) epigenetic alterations [70]. However, familial epigenetic syndromes are rare and appear to be transmitted to offspring [70]. On the other hand, environmental factors have the potential to modify epigenetic states. These environmental factors, as previously mentioned, include infectious pathogens, diet, smoking, atmosphere, etc., and thus, can alter the epigenome. These epigenetic alterations (depending on the changed epigenetic factor and/or profile may also be referred to as ‘aberrant’) can be part of the inflammation and cancer profile, either as causative factors or as resulting factors of the cancer phenotype [70]. The prevailing consensus suggests that epigenetic alterations in cancer occur and are more common than genetic alterations (mutations). Compared with gene mutations, which are irreversible, epigenetic alterations, either inherited or acquired, are largely reversible by intervention. Advances in genomic and epigenomic analysis technologies have led to the identification of epigenetic alterations in IBD and CRC. These epigenetic changes can have significant roles as biomarkers in the clinical setting and as important tools for the early detection, diagnosis, prognosis and management of precancer and cancer states in IBD and CRC [70].

## 5. Epigenetic Mechanisms Linking Inflammation in IBD to Colorectal Carcinogenesis

This section explores the role of epigenetic modifications as critical intermediaries in the transition from chronic inflammation to colorectal cancer. It highlights key changes such as aberrant DNA methylation with special emphasis on how inflammatory signaling cascades modulate the epigenome, and how these alterations influence gene expression, immune evasion, and malignant transformation.

### An Overview of Inflammation-Driven Carcinogenic Transition in Humans and Murine Models

Inflammation is a well-known risk factor for cancer, and epigenetic modifications such as DNA methylation and acetylation play crucial roles in this process [3,4]. In humans, chronic inflammation often leads to global DNA hypomethylation and regional hypermethylation, which can result in chromosomal instability and altered gene expression [3,4]. For instance, in colitis-associated cancer (CAC), oxidative stress and pro-inflammatory cytokines like IL-6 and TNF-α induce DNA methylation changes that contribute to malignant transformation [4]

In mouse models (for details see Section 3), research has shown that inflammation-driven changes in DNA methylation and hydroxymethylation patterns can lead to an imbalance in DNA methylation-demethylation dynamics [83,84,116]. This imbalance can shift histone acetylation patterns, further promoting cancer initiation and progression [116]. Two examples supporting the imbalance are as follows: (a) There is compelling evidence indicating that altered methylation and demethylation dynamics contribute to the pathophysiology of acute kidney injury (AKI). Also, in mouse models of ischemia–reperfusion injury (IRI), endotoxin, or maleate-induced AKI, a global reduction in 5hmC has been observed, whereas overall 5mC levels remain largely unchanged [117]. (b) Moreover, aberrant DNA methylation is a hallmark of cancer, driving abnormal gene expression through hypermethylation and silencing of tumor suppressors, alongside hypomethylation and activation of prometastatic genes [118].

Histone modification abnormalities arise during the transformation of inflammation into CRC [109]. Recent work has shown that dysregulation of histone modifications is closely associated with pathogenesis of gastrointestinal disorders [88]. Aberrant histone modifications have been identified in colonic mucosa of patients with IBD, which may contribute to the chronic inflammation that characterizes these diseases [136,137]. Similarly, alterations in histone modifications have been associated with the development and progression of CRC [138,139].

For example, transcriptomic studies have demonstrated elevated activation of inflammatory pathways in organoids and tissues derived from ulcerative colitis patients. Specifically, enhancer profiling marked by H3K27ac enrichment revealed that UC-derived organoids were enriched for signaling pathways linked to gastrointestinal cancer. This included S100 calcium-binding protein P (S100P) and also identified novel gastrointestinal cancer markers such as lysozyme (LYZ) and neuropeptide S receptor 1 (NPSR1). Immunolocalization further confirmed increased expression of LYZ, S100P, and NPSR1 proteins in UC and CAC. Collectively, these findings indicate that precancerous molecular programs are already activated in UC [140]. The above work focused on the genome-wide enhancer state in cells and tissues derived from CRC patients. However, Chen et al. [141] wanted to further elucidate how the dynamic states of chromatin contribute to the inflammation-cancer transition in colitis-associated CRC. To this end, they performed epigenomic and transcriptomic studies in a colitis-associated CRC mouse model [142] induced by azoxymethane (AOM) and dextran sodium sulfate (DSS) [142]. Combining the data from the above analyses, they generated a genome-wide landscape of chromatin states during inflammation-cancer transition. They support that their work provides important datasets for CRC studies and reveals new regulatory mechanisms and potential targets for clinical investigations. Their results are not only interesting but, more importantly, reveal key modifications and chromatin positions and states in the inflammation-cancer transformation.

In a functional analysis study for differentially expressed genes (DEGs) compared with control tissues, it was revealed that DEGs from the 2- and 4-week samples were enriched in inflammatory pathways, whereas those from the 7- and 10-week samples were enriched in both inflammatory and cancer-associated pathways. Based on this, the 2- and 4-week stages were defined as representing inflammation, while the 7- and 10-week stages reflected tumor development. This classification was supported by comparison with a previous mouse model study, in which the 2- and 4-week datasets clustered with inflammatory bowel disease samples, and the 7- and 10-week datasets clustered with CRC samples.

To examine chromatin dynamics in this model, Chen et al. [141] performed ChIP-seq for H3K27ac, H3K4me1, H3K4me3, H3K27me3, and H3K9me3 at five time points. Chromatin states were defined by combinations of these modifications: quiescent regions (no detected marks), heterochromatin (dominated by H3K9me3), transcriptionally repressed regions (strong or weak H3K27me3), active enhancers (high H3K4me1 and H3K27ac), poised enhancers (high H3K4me1 with low H3K27ac), bivalent enhancers (H3K4me1, H3K27ac, and H3K27me3), weakly active enhancers (low H3K4me1 and H3K27ac), active promoters (all containing H3K4me3 and located near TSS, with four subtypes identified), and poised promoters (marked by both high H3K4me3 and H3K27me3). Comparison across time points showed that these chromatin states were highly dynamic during the inflammation-to-cancer transition, with enhancer regions increasing progressively, particularly at the late tumor stage.

With respect to histone phosphorylation, Xiao et al. [143] found that reduced levels of phosphorylated histone H3 at Ser 10 (H3S10ph) were observed in mouse and human cancer cell lines. Their work showed that phosphorylation events with T-LAK cell-originated protein kinase (*TOPK*) facilitated carcinogenesis of colon cancer [143]. Moreover, histone phosphorylation does not act alone, but partners with other histone modifications to control gene regulatory processes. In vitro experiments using mouse and human cancer cell lines demonstrated that the histone acetyltransferase (HAT) GCN5 shows a preference for phosphorylated H3S10 in contrast to non-phosphorylated histones [4,144]. Phosphorylation of H3S10 (H3S10ph) can also stabilize histone H4 acetylation, whereas its dephosphorylation acts in concert with HDAC1, HDAC2, and HDAC3 to promote H4 deacetylation under stress conditions [145]. Moreover, H3S10ph has been reported to facilitate the expansion of genomic regions enriched with H3K4 methylation, a marker of open chromatin, while at the same time limiting the spread of heterochromatin characterized by H3K9me2 and DNA methylation, both associated with closed chromatin [146]. These findings highlight the extensive crosstalk between histone phosphorylation and other histone modifications, which together regulate gene expression in the contexts of inflammation and cancer [4].

Histone H2B ubiquitination of lysine 120 (H2BK120ub) has been shown to have a role in inflammation-related colorectal cancer. Specifically, in both human colonic tissue cultures and mouse animal models, reduced levels of H2BK120ub and its E3 ligase, RNF20, were found to activate colonic inflammation and tumorigenesis by way of recruiting *NF-κB*, a major transcription factor regulating inflammation signaling in both mice and humans [147]. Other studies also demonstrated that dysregulated H2BS120ub causes genomic instability and promotes tumorigenesis and cancer progression in other cancer types [131,148]. Similarly to phosphorylation, which has been shown to interact with other histone modifications or histone-modifying enzymes in inflammation-associated cancers, histone ubiquitination can also crosstalk and influence other histone modifications. For example, H2BK120ub helps in the methylation of H3K79 and H3K4 at promoter regions to induce gene transcription [149,150]. All in all, histone ubiquitination possesses roles in both transcriptional regulation and induced tumorigenesis [4].

## 6. DNA Methylation and Histone Modifications During Tumorigenesis of the GI

This section delves into how chronic inflammation mediates long-term epigenetic reprogramming of intestinal epithelial cells, serving as a bridge between environmental stress and genetic dysregulation. Epigenetic mechanisms such as DNA hypermethylation of tumor suppressor genes (e.g., *MLH1*, *SEPT9*), histone modifications (loss of acetylation, H3K27me3), and the dysregulated expression of non-coding RNAs (e.g., *miR-21*, lncRNA HOTAIR) are discussed in the sections below. The contribution of inflammation-induced reactive oxygen species (ROS) and immune cell infiltrates to these alterations is also discussed. These changes not only disrupt epithelial homeostasis and promote carcinogenesis, but also provide biomarkers and therapeutic targets.

### 6.1. DNA Methylation in IBD, CAC and CRC

In GI cancers and most other cancer types, global DNA hypomethylation commonly occurs alongside aberrant regional hypermethylation. The latter has been widely investigated across nearly all cancer types and is thought to promote tumorigenesis by repressing tumor suppressor gene expression. Hypomethylation, although also recognized as a hallmark of cancer, has a less clearly defined role. Its contribution is believed to arise from promoting genomic instability, which in turn may activate the expression of parasitic elements, or the expression of oncogenes or cancer germline genes (germline genes with mutation(s) that are [70,151,152,153,154]. However, despite the lack of knowledge as to its functional role in cancer, results have shown that global hypomethylation, which generally occurs on transposable elements (e.g., long interspersed nuclear element 1 (LINE1) or L1 or short interspersed nuclear element/Alu (SINE/Alu), occurs in many cancer types, including CRC.

Studies have demonstrated that L1 hypomethylation occurs widely in CRC patients and is associated with clinically relevant bio-pathological features [155] and correlates with poor prognosis and early onset (<60 years) [155,156,157,158,159]. In fact, LINE-1 hypomethylation was found to be significantly correlated with shorter overall survival (OS), disease-free survival (DFS) and cancer-specific survival (CSS). Importantly, a shorter OS that was found to be associated with L1 hypomethylation was also identified in early-stage colorectal cancers [155,160].

These correlative findings represent promising tools for prognosis prediction. Moreover, a large-scale study involving 1317 colon and rectal carcinoma cases demonstrated a significant association between LINE-1 (L1) hypomethylation and increased colorectal cancer-specific mortality, with the effect being more pronounced in proximal colon cancers compared to distal colon or rectal cancers [161]. Furthermore, patients exhibiting low LINE-1 methylation who received adjuvant chemotherapy showed longer survival compared to those treated with surgery alone, indicating a survival advantage linked to oral fluoropyrimidine therapy. Contrastingly, no survival benefit from chemotherapy was observed among patients with high LINE-1 methylation levels.

This suggests that L1 hypomethylation versus L1 hypermethylation can be used as a predictive marker for the survival benefit of adjuvant chemotherapy with oral fluoropyrimidines [162]. Furthermore, detection of L1 hypomethylation levels in plasma cell-free DNA (cfDNA) was recently proposed as a novel biomarker for detection of CRC in the early stages (biomarker for CRC, particularly for early-stage detection).

DNA methylation in physiological cells takes place predominantly within repetitive genomic regions, including satellite DNA and parasitic elements like long interspersed nuclear elements (LINEs) and short interspersed nuclear elements (SINEs), thereby contributing to the maintenance of genomic integrity [70,163,164]. In contrast to these regions, CpG islands—especially those located within gene promoters—are generally unmethylated in normal cells, which allows access for transcription factors and chromatin-associated proteins to drive the expression of most housekeeping genes as well as other regulated genes. However, a subset of these CpG islands can undergo methylation in a tissue-specific manner, particularly during early developmental processes and/or within differentiating tissues, where the DNA methylation level at some CpG sites reaches approximately 6% [70,165]. DNA hypermethylation promotes tumorigenesis and progression of colitis-associated CRC (CAC). IBD patients show DNA methylation changes both at the cell and at the tissue level [12]. These changes also differ between UC and CD patients [166,167,168,169,170,171,172]. In the following paragraphs of this section, examples of changes in DNA methylation levels and DNA methyltranferases associated with IBD and inflammation-associated CRCs will be summarized. Table 2 lists the DNA methylation status of certain genes that have been associated with certain inflammatory bowel conditions and their use as diagnostic and/or therapeutic biomarkers.

The expression of DNA methyltransferase 1 (DNMT1), which has a crucial role in maintaining DNA methylation patterns in the cell generations, is higher in CAC samples than in those of tumor tissue samples of patients with sporadic CRC. Generally speaking, sporadic cancer refers to cancer that arises due to random DNA damage and subsequent genetic mutations in cells, acquired during a person’s lifetime, rather than being inherited from a parent. These mutations typically occur in somatic cells (non-reproductive cells), do not have a clear pattern of inheritance within families and are not passed on to future generations. Specifically, sporadic CRCs are cancers that arise from the colorectum without known contribution from germline causes (germline inherited mutation(s)) or significant family history (inherited, familial) or inflammatory bowel disease. The increased levels of the DNA methyltransferase, DNMT1, indicate increased DNA methylation levels in CAC tumor tissues [109,173].

Examples of genes of importance, which are tumor suppressor genes and related to cell cycle events that were found to be hypermethylated in tumor tissues of CAC patients, are the cell cycle inhibitor gene, p16 [174,175], and the gene involved in the regulation of *p53*, i.e., *p14* [176]. P14 binds to *MDM2* and stabilizes the *MGM2-p53* complex, which holds inactive p53. There is an inverse relationship between p14 expression and p53 function in tumor cell lines. Indeed, down-regulation of the *p14* gene by DNA methylation is a relatively early event in ulcerative colitis-associated colorectal carcinogenesis [177]. Furthermore, the DNA methylation levels of the genes *TFP12* (tissue factor pathway inhibitor), *ITGA4* (integrin alpha 4) and *VIM* (vimentin) are increased in inflamed colon tissue. These results strongly imply a high risk for development of inflammation—induced CRC and that the methylation levels of these genes can be used as risk markers for inflammation-associated colon cancer [178]. Altered DNA methylation patterns in tumors have been termed DNA methylation valleys (DMV). These regions extend over several kilobases of DNA, are strongly hypomethylated in most normal tissues, and are enriched in genes for transcription factors and development [176]. DMVs have been shown to become hypermethylated in colorectal cancer and may thus contribute to the aberrant epigenetic programming of tumor cells [179]. Specifically, in a colitis-induced mouse colon cancer model, investigators found hypermethylation of DMVs leading to silencing of the DMV-related genes, thus facilitating inflammation-induced cell transformation. Based on the above, the authors [180] proposed that the DNA methylation status of a specific subset of DMVs may be a promising early detection biomarker of inflammation-induced CRC.

Other tumor suppressor genes that have been reported to be hypermethylated in colorectal cancer have also been proposed to be used as early diagnostic markers of CRC for detection in stool or blood samples of patients. Among the genes that undergo repression in CRC due to CpG hypermethylation, the most extensively studied for their impact in cancer diagnosis or prognosis are MGMT, SEPT9, HLTF, NDRG4, BMP3, CDH13, APC, MLH1, CDKN2A, RASSF1A and RUNX3 [155].

More importantly, and in terms of the mechanisms and signalling pathways involved, inflammation-induced oxidative stress, reactive nitrogen species, and cytokine-driven transcriptional activity promote aberrant DNA methylation patterns in epithelial cells. Hypermethylation of CpG islands in promoter regions leads to transcriptional silencing of tumor suppressor genes such as *MLH1*, *CDKN2A* (p16), and *SEPT9* [180,181]. Notably, Septin 9 (*SEPT9*) promoter methylation has been utilized as a clinical biomarker for early CRC detection and has been linked to increased inflammatory signaling via NF-κB activation [182].

In IBD, global DNA hypomethylation is also observed, contributing to genomic instability. Simultaneously, promoter hypermethylation selectively targets genes involved in DNA repair and apoptosis, tipping the balance toward uncontrolled cell survival [183]. Furthermore, DNA repair protein, O6-methylguanine-DNA methyltransferase (*MGMT*) promoter hypermethylation has been detected early in CRC, while hypermethylation of Helicase-like Transcription Factor (*HLTF*) was detected in the serum of CRC patients and is associated with an increased risk of disease recurrence and death. Another study confirmed the positive correlation of serum positive HLTF and transmembrane protein containing epidermal growth factor and follistatin domains (*HPP1/TPEF*) DNA hypermethylation with tumor size, stage, grade and metastatic disease [155,184]. Hypermethylation of N-myc down-regulated family member 4 (*NDRG4*) was correlated with CRC clinical features [185], while Bone Morphogenetic Protein 3 (*BMP3*) hypermethylation was correlated with microsatellite instability [186]. Cyclin Dependent Kinase Inhibitor 2A (*CDKN2A/p16*) hypermethylation has been associated with worse prognosis in CRC, i.e., reduced OS, presence of lymph node metastasis and lymphovascular invasion [155,187,188].

Several additional studies have demonstrated that the methylation of specific genes is linked to inflammatory conditions, dysplasia, and malignant transformation, thereby underscoring its role in inflammation-driven cellular transformation. Numerous proinflammatory cytokines, secreted as a consequence of the activation of the NF-κB and STAT3 transcription factor signaling pathways, become upregulated and facilitate the progression from inflammation to CRC [189]. A notable example is interleukin-6 (IL-6), which silences the expression of the suppressor of cytokine signaling 3 (SOCS3) through the induction of elevated levels of DNMT1. Since SOCS3 functions as a critical negative regulator of cytokine-mediated STAT3 signaling, its silencing ultimately promotes the onset of CRC [190]. Furthermore, IL-6 has been shown to enhance methylation levels within promoter regions of genes associated with tumor suppression, cell adhesion, and resistance to apoptosis. Importantly, these IL-6–induced increases in methylation could be reversed by treatment with the DNMT1 inhibitor, 5-azadeoxycytidine (Figure 1) [173]. In addition, IL-6 produced during intestinal inflammation can modulate the expression of certain genes, including CYP2E1 and CYP1B1 (members of the cytochrome P450 enzyme family). This modulation alters the metabolic capacity of epithelial cells, which may enhance the activation of dietary carcinogens and promote DNA damage, thereby contributing to CRC development [109].

### 6.2. Histone Modifications in Inflammation-Related Cancer Progression

Histone modifications are also major epigenetic determinants of chromatin structure and function that can be dysregulated by inflammatory cytokines such as IL-6 and TNF-α. Pro-inflammatory conditions often result in loss of histone acetylation (see Section 6.2.1 for more details), particularly at tumor suppressor gene loci, through the upregulation of histone deacetylases (HDACs). For instance, reduced H3K9 and H4K16 acetylation have been reported in colonic tissues during chronic inflammation and dysplasia [191].

Moreover, repressive methylation marks like H3K27me3, catalyzed by the Polycomb Repressive Complex 2 (PRC2), are enriched at loci encoding differentiation and apoptotic regulators during early CRC development. These modifications, initially reversible, become fixed under continued inflammatory pressure, locking cells into a dedifferentiated, proliferative state [192].

#### 6.2.1. Histone Acetylation in CRC

Aberrant histone acetylation patterns are strongly linked to CRC pathogenesis. One of the earliest deregulated marks identified was the global loss of histone H4 acetylation at lysine 16 (H4K16ac), observed in both CRC cell lines and primary tumor samples [131,193]. Global hypoacetylation of H4K12 and H3K18 was further associated with poorly differentiated colorectal adenocarcinomas [194], whereas overall histone acetylation levels, such as H4K12ac, were elevated in moderately differentiated tumors, showing a gradual rise from normal tissue to carcinoma. Loss of H3K9 acetylation was directly linked to the silencing of the tumor suppressor gene E-cadherin in CRC cell lines [195,196]. In addition, a CRC cell line stably transformed with oncogenic Harvey-Ras—which drives EMT—showed global H3K9/14 acetylation at the promoters of E-cadherin and cyclin D1, genes crucial for EMT and cell cycle control. This modification ultimately reduced their protein expression through activation of Ras signaling [197]. Acetylation of both H3 and H4 is also required for transcriptional activation of 15-lipoxygenase-1 (15-LOX-1), whose gene is typically silenced in CRC cells [198]. Reduced transcription of the tumor suppressor p21WAF1, which inhibits cyclin-dependent kinases, has been associated with H3 hypoacetylation and altered histone-modifying enzymes in CRC [199]. Likewise, diminished acetylation contributes to silencing of N-myc downregulated family member 1 (NDRG1), a metastasis suppressor gene, in the highly metastatic SW620 colon cancer cell line compared with the less metastatic SW480, in which higher levels of H4 acetylation were found [200].

#### 6.2.2. Histone Methylation in Inflammatory Signaling and CRC Progression

Histone methylation also plays a pivotal role in inflammatory signaling. For instance, histone H3 lysine 9 (H3K9) methyltransferases (HMTs) and demethylases (HDMs) act in opposition to maintain the dynamic balance of H3K9 methylation. A key histone demethylase, Jmjd3—also referred to as KDM6B—is responsible for removing specific histone marks and regulating differentiation and cell identity in macrophages. Through this function, Jmjd3 bridges the connection of inflammation and epigenetic reprogramming [5,201]. When macrophages are exposed to bacterial components or inflammatory cytokines, Jmjd3 is induced, whereupon it associates with polycomb group target genes and modulates their repressive H3K27me3 levels, thereby influencing transcriptional activity [202]. Moreover, continuous stimulation with the cytokine IL-4 activates Jmjd3 demethylase activity, leading to the removal of the H3K27me3 repressive mark from the STAT6 promoter. Once STAT6 is activated, it positively regulates Jmjd3 through direct promoter binding. In addition, Jmjd3 promotes the expression of other inflammation-related genes by erasing their repressive H3K27me marks [5,203].

Thus, histone methylation can affect inflammatory signaling through the above as well as through other inflammatory signaling pathways in many forms of cancer. Cooperative interactions between DNA methylation and histone methylation have also been shown in severe systemic inflammation (SSI). In many forms of cancer, including CRC, deregulation of these inflammatory pathways, through deregulated histone methylation, has been ascertained. In colon cancer, the production of Th1-type chemokines such as CXC chemokine ligand 9 (CXCL9) and CXCL10, which are crucial for T cell recruitment, is suppressed by H3K27me3 modification at their gene promoters [204]. In contrast, the chemokine receptor CXCR4 is upregulated through EZH2-mediated suppression of miR-622, thereby creating conditions that favor tumor cells in evading immune surveillance [5,205,206]. In CRC (among other malignancies), a classic histone methylation mark is the loss of tri-methylation at lysine 20 of histone H4 (H4K20me3), along with the global loss of DNA methylation and acetylation at lysine 16 [193]. In addition, mono-, di- and tri-methylation of H3K4 (activating) are targets of SMYD3 (SET and MYND domain-containing protein 3) HMT and LSD1 (lysine-specific demethylase 1) HDM, both of which are highly expressed in CRC [131]. Furthermore, genome-wide analyses of histone methylation in colorectal cancer (CRC) revealed that the activating H3K4me3 and the repressive H3K27me3 marks exhibit similar patterns in both normal colon and tumor tissues, with notable differences arising only in CRC cell lines. Tumor-associated genes marked with H3K4me3 in normal colon tissue became hyperactivated in tumors, while genes carrying H3K27me3 with low expression in normal tissue became further silenced in CRC tumors [131,207]. Additionally, the presence of H3K4me3, the loss of H3K27me3, and increased H3 acetylation were all linked to the reactivation of previously silenced genes in CRC. With respect to CRC metastasis, a decreased level of H3K4me3 (along with decreased acetylation, see above in Section 6.2.1 “Histone acetylation in CRC”) in the coding region of the NDRG1 gene in the highly metastatic colon cancer cell line, SW620, was found and associated with the gene’s down-regulated expression [131,200].

A crucial histone mark of heterochromatin linked to transcriptional repression, H3K9me3, was found to be increased in cancer types, possibly promoting gene silencing of tumor suppressor genes [208]. Overexpression of the protein-lysine methyltransferase G9a and H3K9-specific p53 methyltransferase has been reported in CRC, as well as in other cancer types, associated with suppressive alterations in gene expression [131].

#### 6.2.3. Histone Phosphorylation in CRC

Histone phosphorylation is abnormally regulated in colorectal cancer (CRC), resulting in an imbalance in gene transcription [209]. This modification directly influences the expression of CRC-related genes, thereby facilitating tumor development and progression. Studies [209] have shown that EZH2 and anti-silencing factor 1 (Asf1) phosphorylate histones H2B and H4, respectively, which in turn activate the transcription of autophagy-related genes, induce autophagy in CRC cells, and contribute both to disease progression and drug resistance [210]. In addition, VprBP—a kinase that is markedly overexpressed in CRC cells—plays a direct role in epigenetic gene silencing through histone H2A phosphorylation. By regulating transcription of growth-related genes in this manner, VprBP substantially enhances carcinogenesis and promotes the proliferation of cancer cells [209,211]. Moreover, researchers showed that phosphorylation of H2A.X (an H2A class subtype) is elevated in CRC tissues and was correlated with a more aggressive type of tumor and poor CRC patient survival [139,212]. Another example of aberrantly regulated histone phosphorylation in CRC is the observation of the downregulation of the expression of the dual specificity phosphatase 22 (*DUSP22*) in CRC specimens and reduced *DUSP22* expression in stage IV patients who mainly showed poor survival outcomes [139,213]. Moreover, Chen and co-workers [214] found that PKCε is a kinase that phosphorylates *MIIP-S303*. There is an induction of PKCε-dependent phosphorylation of migration and invasion inhibitory protein (MIIP), which is localized at serine 303 (Ser303), and stimulated by the Epidermal growth factor (EGF). This phosphorylation enhances the nuclear interaction between MIIP and RelA, whereby MIIP inhibits histone deacetylase 6 (HDAC6)-mediated deacetylation of RelA, ultimately advancing RelA transcriptional activity and promoting tumor metastasis. Conversely, protein phosphatase 1 (PP1) acts as a counter-regulator by mediating MIIP-Ser303 dephosphorylation, with its expression levels inversely associated with the metastatic potential of colon tumor cell lines such as HCT116 and CaCo2. Clinical analyses further demonstrate that phosphorylation levels of MIIP at Ser303 strongly correlate with colorectal cancer (CRC) metastasis and patient prognosis [139,214].

Table 3 shows representative histone modifications whose levels were found to change in different inflammatory bowel conditions (different samples), including CAC and CRC, and thus, could be considered potential biomarkers.

## 7. Exploring the Role of Non-Coding RNAs in Epigenetic Regulation of IBD and CRC

MicroRNAs (miRNAs) and long non-coding RNAs (lncRNAs) act as post-transcriptional regulators and chromatin remodelers, respectively. Chronic inflammation modulates the expression of several oncogenic miRNAs, such as *miR-21*, which is overexpressed in IBD and CRC tissues and associated with inhibition of tumor suppressors such as *PDCD4* and *PTEN* [128,129]. MicroRNA-21 is now under investigation as a diagnostic biomarker and therapeutic target.

Similarly, inflammation-sensitive lncRNAs such as *HOTAIR* and *LINC-PINT* participate in epigenetic gene silencing through interaction with histone-modifying complexes, contributing to the persistence of an oncogenic transcriptional landscape. The next subsections shed light on related research in this subject that has been reported in scientific literature.

Additionally, in the current review, recent versions of ncRNA-oriented databases and biotools were employed by the first author, such as LncRNA2Target versions V2.0 and V3.0, Open Targets Genetics v22.10, The Human Reference Protein Interactome Mapping Project (HuRI) NONECODE V6, lncRNAfunc, LncRNADisease v2.0, starBase v2.0, EVLncRNAs2.0, lncRNAfunc, LncRNAWiki 2.0, and the updated LNCipedia_5.2, for the investigation of ncRNA contributions to the disease related to this study [215,216,217,218,219,220,221,222,223].

### 7.1. MicroRNAs—Molecular Insights of microRNA Dysregulation or Aberrant Function and Its Involvement in IBD and CRC

Aberrant expression of microRNAs (miRNAs) and other non-coding RNAs (ncRNAs) has been documented across numerous tumor types, including colorectal cancer (CRC) [155,224,225,226,227]. In CRC, part of this deregulation arises from epigenetic modifications affecting the regulatory regions of miRNA and ncRNA genes. For instance, researchers [228] demonstrated that hypermethylation of *miRNA-124-1/3* in colon cancer and other solid tumors leads to reduced levels of mature *miRNA-124a*, accompanied by increased expression of its target *CDK6* gene and enhanced phosphorylation of the Rb protein—both key regulators of cell cycle progression [228]. In CRC tissues specifically, hypermethylation of *miRNA-124* family genes was detected in over 70% of cases. Additional miRNAs have also been reported to undergo aberrant methylation in the early stages of CRC, including *miR-137* [229], the *miR-200* family [230], as well as *miR-129* and *miR-9* [231]. Together, these findings highlight the critical role of epigenetic regulation of miRNA expression in maintaining tumor-suppressive functions [155].

Evidence has also accumulated showing miRNAs’ critical contribution to the disease onset and progression of IBD, supporting further investigation as to the possible role(s) of miRNA as markers in differential diagnosis. MicroRNA expression patterns have been found to differ significantly between IBD patients and healthy controls, between CD patients and UC patients, as well as between patients in remission and those in the active stages of the illness [13]. Moreover, CD patients always displayed increased levels of *miR-340* in peripheral blood. In another study, four specific miRNAs (*miR-20b*, *miR-98*, *miR-125b-1** and *let-7e**) were identified in colonic mucosa of UC patients, which were differentially upregulated by more than 5-fold in active UC as compared to inactive UC, active CD, inactive CD and healthy controls. These results not only corroborate accumulated evidence that microRNAs contribute to disease onset and progression, but also support the use of specific miRNAs as diagnostic markers of the differential states of colon inflammatory conditions [232].

In line with disease onset and progression, numerous microRNAs have been implicated or identified as advancing transformation by participating in NF-κB and STAT3 signaling pathways that play an important role in transformation of inflammation into cancer [233]. NF-κB and STAT3 are transcription factors that regulate the expression of a variety of genes that coordinate innate and adaptive immune responses, and responses to cellular stimuli, respectively. MicroRNAs’ roles in GI tract cancers, namely, colon, gastric and liver cancers, have been investigated and found to play key roles in cell growth and apoptosis [234]. Their activation and the interaction of their signaling pathways play vital roles in control of the communication between cancer cells and inflammatory cells [234]. This interaction can lead to the transformation of inflammation into cancer [109]. Numerous microRNAs promote this transformation by participating in these signaling pathways. TNF-α (Tumor Necrosis Factor-alpha, a multifunctional cytokine) increases the expression of *miR-105*, which targets *RAP2C* (a Ras-related protein subfamily of the Ras GTPase superfamily that regulates cell proliferation, differentiation and apoptosis) and activates NF-κΒ signal transduction by IKK (central core element of the NF-κΒ cascade), which ultimately contributes to CRC progression [235]. An additional study demonstrated that TNF-α induces high expression of *miR-19a*, which in turn activates NF-κB signaling, thereby aggravating colitis and facilitating the development of CAC. STAT3, a downstream effector of IL-6, can interact with several regulators, including *miR-21*, *miR-181b-1*, *PTEN*, and *CYLD*, forming a potential epigenetic switch that links inflammation with tumorigenesis [236]. Through the activation of *miR-21*, STAT3 further drives TGF-β-dependent EMT in CRC [237]. In CRC samples and cell lines, elevated STAT3 expression is also correlated with higher levels of *miR-572*, which suppresses the pro-apoptotic protein Modulator of Apoptosis 1 (*MOAP-1*), thereby contributing to tumor progression [238]. Moreover, natural killer (NK) cells enhance tumor cell apoptosis by releasing large amounts of cytokines, including IFN-γ and TNF-α [239]. Notably, CRC patients exhibit increased levels of *miR-24* in NK cells. This is thought to play a causative role in the decreased levels of cytokines, including TNF-α and IFN-γ. In this manner, the increased miR-24 levels inhibit the cytotoxic effects of NK cells on CRC cells [240].

### 7.2. MicroRNAs as Biomarkers in GI Diseases

MicroRNAs are found circulating in human peripheral blood in a stable form and are also detectable in other body fluids, including urine, saliva, milk, cerebrospinal fluid, and feces [241]. Alterations in miRNA expression profiles have been explored for potential applications in early detection, prognosis, and diagnostic classification of IBD. Recent investigations in this field have analyzed circulating miRNAs in body fluids as well as in homogenized tissue biopsies, employing microarray, RT-qPCR, and next-generation sequencing (NGS) methodologies (Table 4) [242,243]. Notably, miRNAs are increasingly being investigated and will be more scrutinized in the future as non-invasive markers for CRC.

*MicroRNA-21* and *miR*-*155* have been identified repeatedly and appear to be the most studied miRNAs associated with IBD [243,244]. *MicroRNA-21* is potentially the most interesting miRNA involved in IBD, with associations between *miR-21* and disease. It has been replicated in several studies, and functional significance has been reported in mouse models of IBD [245]. *MicroRNA-21* is elevated in both UC and CD patients and is involved in several proinflammatory functions, such as modulating T-cell responses and controlling epithelial tight junction proteins [244]. It should be emphasized that deactivation of *miR-21* reduced inflammatory responses and improved survival rate in a mouse model of DSS-induced colitis [246].

In another study, only *miR-150* was downregulated out of the 25 miRNAs specifically expressed in the serum of UC patients. Additionally, a significant increase in *miR-29a* was observed in the blood of UC patients, which plays an important role in regulating both innate and adaptive immune responses by directly targeting interferon (IFN)-γ. Supporting this functional role, two independent studies demonstrated elevated expression of miR-29a in colonic tissues from patients with both active and inactive UC. Importantly, serum levels of *miR-29a* have been proposed to hold strong promise as a novel non-invasive biomarker for the early detection of colorectal cancer. Since colorectal cancer represents a well-recognized complication of long-standing UC, the association of *miR-29a* with both active and inactive UC reinforces its value as a biomarker for early CRC detection [247,248] along with the increased expression of *miR-127-3p* in both UC and CD patients, suggesting that *miR-127-3p* could be a potential biomarker for IBD [249].

Two more studies reported that (a) serum *microRNA146b2-5p* (*miR2-146b2-5p*) expression was 2.872- and 2.722-fold higher in CD and UC patients than in healthy controls [250], and (b) a study, which is dealing with the miRNA family, *miR-125*, consisting of *miR-125a* and *miR-125b* found that only *miR-125a* is reduced in patients with active disease and negatively correlates with disease severity and inflammatory cytokines in patients with CD [251].

Rashid and co-workers showed that patients with active disease exhibit a distinct miRNA profile and that *miR*-*223* and *miR-1246* are generally present at high levels in feces and are upregulated in active patients with IBD. However, the results are not the same in serum samples from the same patients (in serum samples, *miR-223* shows a greater increase in patients with CD, as discussed above). This increase was seen in patients with UC as well as CD; it is thus concluded that these miRNAs are generally associated with intestinal inflammation [243].

Moreover, Schaefer and co-workers found that serum samples from patients with IBD showed higher levels of *miR-16*, *miR-21*, and *miR-223* than controls and were higher in CD patients. In more detail, distinctive changes in miRNA expression were observed in stool samples from patients with IBD for all tested miRNAs, with the highest expression of *miR-155* and *miR-223* in the control groups. In conclusion, *miR-21*, *miR-155*, and *miR-223* exhibit significant levels and could potentially be considered biomarkers for IBD [252].

Further research has reported the identification of a cluster of nine miRNAs that are dysregulated in the rectal tissue of pediatric IBD patients. Within this group, four miRNAs (*miR-192*, *miR-194*, *miR-200b*, and *miR-375*) were significantly downregulated, while four others (*miR-21*, *miR-142-3p*, *miR-146a*, and *let-7i*) were markedly upregulated in pediatric IBD cases compared to healthy controls. A major clinical challenge lies in distinguishing UC from CD in the pediatric population, as diagnostic clarity is not always achieved even after endoscopic evaluation. Notably, three serum miRNAs showed significant alterations in children with UC, including *miR-192* and *miR-21*, both of which had previously been reported to be raised in pediatric CD [253]. The third miRNA was *miR-142-3p*. However, the study found that colitis-associated miRNA levels could not distinguish UC from CD [254].

Furthermore, regarding the utility of miRNA biomarkers in the treatment of IBD, a recent study was able to identify 8 miRNAs in serum samples that are associated with clinical response to anti-TNF-α and glucocorticoid (GC) therapy [255]. These are *miR-146a*, *miR-146b*, *miR-320a*, *miR-126*, and *let-7c*. Although *miR-146a* and *miR-146b* are elevated in serum and biopsies of individuals with IBD, they appear to be reduced by anti-TNF and GC treatment. As mentioned above, these miRNAs have also been reported as diagnostic biomarkers of IBD, showing a high correlation with endoscopic disease activity. In parallel, *miR-320a*, *miR-126*, and *let-7c* also show downregulation [256]. However, research in this area is limited, and results are mixed. Therefore, further studies are needed to fully investigate and validate the utility of miRNAs as predictive markers for treatment outcomes in IBD [257].

Importantly, Shi and co-workers showed that alterations in *miR-31* levels in TNBS-induced colitis and in IL-10 knockout mice could regulate the IL-12/23 pathway, resulting in improvement or aggravation of colitis. Furthermore, the therapeutic effects of *miR-31* inhibitor were eliminated after inducing IL-25 (interleukin-25) overexpression in the colon in mice [258]. In addition, it was proven that *miR-223* interacts with the IL-23 pathway by targeting claudin-8 (*CLDN8*), which is involved in the formation of tight junctions in the gastrointestinal tract. Intraperitoneal injection of *antagomiR-223* activated *CLDN8* and reduced intestinal permeability in mice with colitis [259].

Interestingly, using a DSS-induced IBD mouse model, a study on *miR-133a* and its target UCP2 (mitochondrial uncoupling protein 2) revealed that miR-*133a* levels were reduced following DSS treatment. The DSS-induced IBD was also alleviated by introducing a *miR-133a* mimic, indicating that miRNA mimics could also serve as therapeutics for IBD [260].

*MicroRNA-223* and *miR-451* have been identified as strong markers for distinguishing CRC patients from healthy individuals, while *miR-223* was also highlighted as a biomarker for IBD [243,261]. Additionally, *miR-135b* has been identified as a marker of cancer origin, further suggesting its potential role in the detection of CRC [262]. Furthermore, *miR-421*, *miR130b-3p*, and *miR27a-3p* were shown to be elevated in CRC patients’ fecal samples. An important addition has demonstrated that a developed algorithm has supported the identification of patients with advanced colorectal neoplasia based on a 5-criterion task force, i.e., fecal levels of two microRNAs (*miR-421* and *miR-27a-3p*), fecal hemoglobin concentration, patient age, and patient sex [263,264].

### 7.3. LncRNAs in IBD and CRC—A General Overview

LncRNAs are important in cancer biology, generally causing abnormal expression of gene products, which can be involved in the progression of various human tumors [229,265]. They are also involved in the transformation of inflammation into CRC. A representative example of this is indicated in the study by Hanisch et al. [266], who showed that the interaction between lncRNA *PRINS* and *miR-491-5p* regulated the pro-apoptotic factor *PMAIP1*, a pro-apoptotic member of the Bcl-2 protein family with a specific domain (BH3 domain) that allows it to interact with other Bcl-2 family proteins, ultimately triggering apoptosis.

This interaction between PRINS and *miR-491-5p* was shown to enhance the anti-apoptotic effect of Trefoil Factor 3 (TFF3)—a small peptide recognized for its role in mucosal protection and wound healing, particularly within the gastrointestinal tract—by counteracting the pro-apoptotic effects of IFN-γ and TNF-α in colorectal cancer (CRC) cells [266]. In addition, the expression of the lncRNA *FEZF1-AS1* was found to be elevated in CRC tissues compared with normal tissues, and this overexpression was associated with poor prognosis [267]. Mechanistic investigations revealed that *FEZF1-AS1* can bind to the pyruvate kinase 2 (PKM2) protein, thereby enhancing its stability, which leads to increased levels and activity of PKM2 in both the cytoplasm and nucleus. This *FEZF1-AS1*–induced upregulation of PKM2 further activates STAT3 signaling, thereby accelerating the transition from inflammation to cancer [267]. Similarly, research has demonstrated that the lncRNA *AB073614* promotes EMT in CRC by regulating the JAK/STAT3 signaling pathway [268]. Moreover, accumulating evidence suggests that lncRNAs not only contribute to the transformation of inflammation into cancer but also play a role in mediating chemotherapy resistance in CRC through the modulation of inflammatory signaling pathways [269]. For example, the lncRNA, *HOTAIR*, has been shown to induce resistance to 5-fluorouracil (5-FU) by repressing *miR-218* and enhancing the activity of NF-κB/TS signaling pathway in CRC. Notably, HOTAIR expression has been strongly correlated with tumor progression, reduced survival, and poor prognosis across multiple cancer types, including CRC. Its interaction with chromatin-modifying complexes and key signaling pathways has solidified its role as a novel prognostic indicator and therapeutic target in CRC; thereby, HOTAIR has been identified as a novel prognostic and promising therapeutic biomarker for CRC [109,270,271].

Further on the importance of lncRNA—miRNA interactions, a detailed functional annotation of the newly characterized lncRNAs, *FIGNL2-DT* and *GAS5-AS1*, is presented in Appendix A), and lncRNA—miRNA interactions are visualized (for 3 important miRNAs) in Appendix A). Appendix A illustrates the plethora of lncRNA—miRNA interactions involved in CD, with different levels of significance for the disease, according to their evaluation by the miRNA Recognition Element (MRE) score [272].

In more detail, the first author performed the analysis and mapped lncRNA—miRNA interactions through the ncFANs-NET module of the ncFAN v2.0 network analysis biotool, which is an updated and full-featured platform for noncoding RNA (ncRNA) functional annotation, that comprises the following three major modules for the lncRNA analysis, i.e., lncFANs-CHIP, ncFANs-NET and ncFANs-eLnc [272]. Conducting this research, three (3) ncRNA transcripts were found in our query list by this platform, as shown in Appendix A, i.e., FIGNL2-DT, *GAS5-AS1* (Appendix A) and Pseudogene *GOLGA2P8* (Triantaphyllopoulos et al.,, unpublished results). Notably, the results of the two lncRNAs involved in the CD, such as *GAS5-AS1* and *FIGNL2-DT* displayed high centrality and high Max_miRANDA_score, in the lncRNA-miRNA interactions’ list in the subnetwork (while the estimated edge number in the network was 169 miRNA ID targets), and linked to significant targets, as shown in Appendix A, e.g., (a) *hsa-mi-205-3p*—a pro-inflammatory miRNA in IBD (miRANDA_score: 162) and (b) *hsa-miR-23b-3p* (miRANDA_score: 152), significant to inhibiting gastric cancer by regulating *miR-23* and/or *hsa-miR-200c* (miRANDA_score: 155), which is related to EMT in inflammation and carcinogenesis, respectively. In Appendix A are also presented the interacting miRNAs, *hsa-miR-1184*, *hsa-miR-8082* and *hsa-miR-6741-5p*, with high Max_miRANDA_score (157, 157 and 163, respectively), as visualized in the lncRNA-miRNA network (Appendix A). Interpreting these results and their impact, we summarize that *GAS5* directly interacts with *miR-23a/b*, as evidenced by pull-down and Ago2-RIP assays, acting to de-repress tumor suppressors such as *GSK3β* and *PTEN* in both oncological and neuronal contexts [273]. Given the involvement of *miR-23* family in intestinal epithelial barrier maintenance and cytokine signaling, the potential regulatory role of *GAS5-AS1* or *GAS5* through these miRNA axes may represent a novel layer of post-transcriptional control during the inflammation-to-cancer transition in IBD. The latter findings are both critical in CD-associated mucosal immunity that may function as tumor suppressors or anti-inflammatory regulators. These lncRNA hubs may also act as molecular sponges, fine-tuning miRNA availability during chronic intestinal inflammation. Importantly, these findings support their potential as regulators of epithelial immune responses but warrant further experimental validation in gastrointestinal disease models to confirm their functional impact and biomarker potential in intestinal pathology.

Furthermore, in order to explore the epigenetic modulation of lncRNA expression in gastrointestinal tumorigenesis, the DNA methylation profiles of cancer-associated lncRNA loci were also searched using the Lnc2Meth database [274]. The analysis highlighted widespread differential methylation of promoters and gene bodies of key lncRNAs in colorectal, colon cancer, gastric cancer, colorectal neoplasia and gastrointestinal stromal tumor (Appendix A). Notably, *ZNF582-AS1*, *TP53TG1*, *MEG3* showed consistent promoter hypermethylation, correlated with transcriptional silencing in tumor tissues of esophageal cancer, suggesting epigenetic inactivation during the malignant progression [275,276,277].

Conversely, hypomethylation of *H19* promoter was associated with lncRNA overexpression, indicating potential oncogenic roles, especially in esophageal and gastric cancer [278,279,280]. These patterns underscore the relevance of methylation-dependent lncRNA dysregulation in the inflammation-to-cancer axis and offer potential for targeted epigenetic diagnostics.

#### LncRNA as Biomarkers in GI Diseases

The lncRNA profile data of colonic biopsy and blood samples differ significantly between patients with IBD and healthy groups. This suggests that lncRNAs have potential as valuable diagnostic biomarkers for IBD [76,281]. Table 5 shows representative lncRNA biomarkers in IBD, CD, UC and CRC, and their gene expression changes compared to healthy individuals. As summarized in Table 5, several studies have demonstrated that lncRNA *Mirt2* and lncRNA *IFNG-AS1* exhibit inverse expression patterns in ulcerative colitis (UC), with *Mirt2* being significantly downregulated and *IFNG-AS1* upregulated in the plasma of UC patients compared with control subjects [76,282]. Furthermore, interleukin-22 (IL-22), which is well recognized for its anti-inflammatory functions, plays a crucial role in suppressing intestinal inflammation. Of particular interest is the observed positive correlation between IL-22 and *Mirt2* levels in UC patients. Even more significant is the finding that both IL-22 and *Mirt2* levels in plasma show an inverse correlation with CRP levels, an acute-phase protein (APP) widely used as a biomarker to evaluate inflammatory status. These results indicate that combined measurement of IL-22, *Mirt2*, and CRP plasma levels may provide improved diagnostic accuracy for UC [76,283].

Among long non-coding RNAs, one that has drawn substantial attention is *HOTAIR*, which regulates multiple target genes through sponging mechanisms and epigenetic modulation. *HOTAIR* influences a wide range of oncogenic cellular processes and signaling pathways, including those governing metastasis and resistance to chemotherapy. It has been reported that *HOTAIR* reprograms chromatin organization and facilitates breast cancer metastasis. Moreover, *HOTAIR* is associated with genome-wide reprogramming of Polycomb Repressive Complex 2 (PRC2) function not only in breast cancer but also in CRC, where its upregulation is thought to represent a key element driving metastatic progression [270].

It has been reported that the expression level of intestinal mucosa and peripheral blood mononuclear cells (PBMCs) *lnc-ITSN1-2* was higher in patients with active UC and patients with UC in remission compared to healthy controls, showing excellent prognostic value in active UC and efficiency in distinguishing patients with active UC from patients with UC in remission [284]. Similar results were obtained with the intestinal mucosal inflammation, where upregulated *ITSN1-2* levels from CD patients showed positive effects in predicting the risk of active CD compared to healthy controls (Table 5). Another interesting observation was that *lnc-ITSN1-2* was decreased after infliximab treatment in active CD patients [284]. Likewise, lncRNA *THRIL* is upregulated in UC as well as in CD patients, which is implicated in innate immunity by regulating the expression level of TNF-α, forming a complex that binds to the promoter region of the TNF-α gene, resulting in its induction [76,285]. Owing to its wide range of cellular processes, including cell proliferation, survival and death, THRIL could be used as a potential biomarker for the diagnosis and prognosis of UC and CD [76,286].

Although the dysregulation of lncRNAs in IBD tissue or plasma samples is a potentially valuable diagnostic biomarker, the pathophysiology of IBD is very complex and has not been fully elucidated; currently, there is not a single gold standard for the diagnosis of IBD. Therefore, a combination of several ncRNAs may be necessary to provide an accurate diagnosis. For example, lncRNA *CDKN2B-AS1* (Table 5), which negatively correlated with increased expressions of inflammatory mediators specific to UC (TNF-α, IL-6, and sIL-2R), was an excellent marker in distinguishing UC as well as CD patients from healthy controls. However, *CDKN2B-AS1* in UC, and in combination with miR-16-5p and miR-195-5p, could greatly improve the diagnostic efficiency for UC (Table 5) [76,287]. In addition, changes in *CDKN2B-AS1* expression are associated with response to infliximab treatment in patients with CD, as *CDKN2B-AS1* expression in infliximab treatment responders increased, whereas that of non-responders remained stable. Thereby, lncRNA *CDKN2B-AS1* may serve as a biomarker to assess the response of patients to this therapy [76,288].

Also, of equal importance is lncRNA *H19*, which has been the subject of several studies due to its association with the development of inflammatory diseases, such as osteoarthritis. *H19* lncRNA is transcribed from the *H19* gene on chromosome 11, is highly expressed in multiple tissues during the embryonic stage but is largely inactivated after birth, and is upregulated in mouse models of colitis and in inflammatory colonic tissues from CD patients. An important aspect of *H19*’s role in disease is its interaction with the vitamin D receptor (VDR), i.e., 1,25 (OH)2D3 (calcitriol), the active form of vitamin D, which is crucial in protecting the intestine from certain damaging agents. VDR also plays a significant role in regulating inflammation and carcinogenesis in various tissues. In the context of UC, overexpression of lncRNA *H19* can decrease VDR levels, disrupting the intestinal epithelial barrier’s function, which contributes to UC pathogenesis. Therefore, the interaction between lncRNA *H19* and VDR signaling may offer potential targets for future therapeutic intervention in UC [76,289].

Additionally, it was found that lncRNA *CRNDE* was involved in colonic epithelial cell apoptosis in IBD and it was highly expressed in tissues from DSS-induced murine colitis models and human colon epithelial cells. In the DSS-induced murine model, *CRNDE* was found to suppress miRNA-495 and increase the suppressor of cytokine signaling (*SOCS1*). MicroRNA-495, which is reduced in UC, normally helps prevent apoptosis of IECs via the JAK/STAT3 signaling pathway, while *SOCS1*, on the other hand, restricts cytokine receptor signaling and promotes IFN-γ-induced apoptosis in these cells. After *CRNDE* intervention in a murine model, the clinical signs were reduced, showing improvement in weight loss and reduction in bloody stools, suggesting that *CRNDE* could be a promising target for the treatment of IBD [75,76].

Moreover, lncRNA *NEAT1* (Table 5) is a remarkable therapeutic biomarker, and *NEAT1* expression was found to be involved in the inflammatory response and elevated in serum and tissue samples from mouse models of IBD. This response is mediated through the regulation of the intestinal epithelial barrier and the exosome-mediated polarization of macrophages. Notably, downregulation of *NEAT1* suppressed the inflammatory response by affecting the same pathways. These findings suggest that targeting *NEAT1* could be a promising strategy for the treatment of IBD [76,290].

As far as the pediatric UC patients are concerned, investigation has shown an association between lncRNA *GAS5* (growth arrest-specific 5) (Table 5 and Appendix A) and the response to glucocorticoid therapy. Thus, it has been observed that the long non-coding RNA *GAS5* is significantly upregulated in peripheral blood mononuclear cells (PBMCs) of ulcerative colitis (UC) patients who display an unfavorable response to glucocorticoid treatment. This finding indicates that *GAS5* may serve as a potential pharmacogenomic biomarker, offering promise for its use in guiding personalized glucocorticoid therapy in these patients [76,291].

Additionally to Table 5, the aforementioned lncRNAs as above (i.e., *CDKN2B-AS1*, *H19*, *IFNG-AS1*, *MALAT1*, *Mirt2*, *TUG1*) have been the subject of vigorous research, and an update with more detailed information is provided by Appendix A). For the newly characterized lncRNA mentioned above, various analysis platforms have been employed by the first author, such as NONECODE V6, EVLncRNAs2.0 [221,222] Ensembl https://www.ensembl.org/index.html, (accessed on 3 February 2025), NCBI, https://www.ncbi.nlm.nih.gov/gene/, (accessed on 3 February 2025) (biotool details in Section 8), for more updated information. Thus, details are shown in Appendix A and include chromosome location, exons, interaction type, interaction target, NCBI accession numbers, and description for interaction and function, supported by citation ID (PMID). The aforementioned data (Appendix A) are also associated with Appendix A). Furthermore, the UC network shown in Appendix A, built under the EVlncRNAs 2.0 database by the first author, presents the following lncRNAs: CDKN2B-AS1, H19, IFNG-AS1, MALAT1, Mirt2 and TUG1 with their first-degree interaction molecules, i.e., significant miRNA interaction targets, coding genes, proteins, protein complexes, etc., as first-degree node interactors. The interaction column distinguishes the types (i.e., U: unknown type; Regulation: shows that lncRNA can regulate the expression of other biomolecules in the same physiological process; Co-expression: to indicate positively or negatively correlated expression of lncRNA with other molecules in the same physiological process; Binding: indicates that lncRNA has direct physical contact with other biomolecules).

As an example of pointing out important interactors in these direct networks, we refer to lncRNA MALAT1, which promotes UC by upregulating lncRNA CDKN2B-AS1 (ANRIL), since both lncRNAs are significantly and positively correlated in UC patients but not in healthy controls, while the latter (CDKN2B-AS1) relieves inflammation of ulcerative colitis via sponging *miR-16* and *miR-195* [284,289]. Notably, the later reported diagnostic miRNAs in UC (*miR-16* and *miR-195*), and potentially therapeutic, are shown in Table 4 [cited by PMIDs: 29668922 and 21546856, respectively], and there are also serum biomarkers for IBD (Table 4). Additionally, miR-200 [230] interaction with MALAT1 was inferred in our network analysis and can be observed in the MALAT1-miR interaction targets network (Appendix A).

The lncRNA known as Metastasis-associated lung adenocarcinoma transcript 1 (*MALAT1*) (Table 5), misregulation has been linked with a lot of autoimmune diseases and it is abundantly expressed in many tissues in various biological processes, including cancer development and metastasis. A study also shows potential interaction between *MALAT1* and IL-6, which was found to be upregulated, contributing to inducing apoptosis and inflammation; thus, it shows *MALAT1*’s potential value as a target in diagnosis and treatment for IBD patients [76,292]. *MALAT1* was also reported to play a role in CRC, while a study demonstrated that the inflammation-associated long non-coding RNA *MALAT1* and *miR-663a* form an endogenous RNA (ceRNA) network in colorectal cancer (CRC) cells via sequence-dependent binding. Specifically, MALAT1 downregulated *miR-663a* expression through a ceRNA mechanism, thereby preventing the degradation of multiple *miR-663a* targets, including P53, PIK3CD, P21, CXCR4, TGFB1, and JUND, in both CRC cells and tissues. These findings suggest that MALAT1 and *miR-663a* may play a critical role in inflammation-driven tumor formation and CRC development [76,293]. In addition, the lncRNAs *KIF9-AS1*, *LINC01272*, and *DIO3OS* (Table 5) were selected for evaluation as potential diagnostic biomarkers for IBD in a study, and the findings showed that tissue and plasma samples from IBD patients had considerably higher levels of *KIF9-AS1* and *LINC01272* mRNA expression than the healthy controls. In contrast, tissue and plasma samples from IBD patients had considerably lower levels of DIO3OS mRNA expression than the healthy controls [76,294]. Furthermore, *MIR4435-2HG* (Table 5) suppression inhibits macrophage M1 polarization while promoting M2 polarization, thereby alleviating intestinal inflammation in DSS-induced mice with ulcerative colitis through JAK1/STAT1 signaling and can be considered as a potential therapeutic target for UC treatment [76,295].

LncRNAs are also being investigated as non-invasive markers for CRC. A bioinformatics-based investigation revealed that *FHIP1A-DT* expression was reduced in colorectal cancer (CRC), and patients exhibiting low levels of *FHIP1A-DT* had poorer prognostic outcomes. Consequently, the lncRNA *FHIP1A-DT* (Table 5) is linked to epigenetic modifications and plays a role in regulating multiple cancer-associated pathways, thereby highlighting a potentially important opportunity for future research in CRC diagnosis and therapeutic strategies [296]. Also, several studies were engaged with lncRNA *PVT1* (Table 5), which is highly upregulated in CRC patients. This upregulation positively correlates with cell proliferation, invasion, tumor stages, and lymph node metastasis. A study showed *PVT1* is highly expressed in CRC patients, and its level is closely related to vascular invasion and lymph node metastasis, while down-regulation of *PVT1* induces apoptosis and inhibits proliferation in CRC cells [297].

Collectively, epigenetic dysregulation plays a pivotal role in the stepwise transformation from inflammation to CRC. To better illustrate the stage-specific accumulation of epigenetic disruptions during colitis-to-carcinoma progression, the following Table 6 summarizes key alterations in DNA methylation, histone modifications, and non-coding RNA profiles across each pathological stage. In more detail, Table 6 summarizes stage-specific epigenetic alterations observed during the progression from normal intestinal epithelium to colitis, dysplasia, and colorectal cancer. Importantly, Table 6 illustrates how specific epigenetic changes emerge and accumulate across the stages of IBD-associated CRC, offering a roadmap for biomarker discovery and targeted therapies.

Key molecular events are categorized into four major epigenetic regulatory layers: DNA methylation, histone acetylation, histone methylation, and non-coding RNA modulation. For each pathological stage, representative changes are presented based on recent literature, including promoter hypermethylation of tumor suppressor genes (e.g., *p16INK4a*, *MLH1*) in dysplasia, decreased global acetylation of histones H3 and H4 during inflammation, aberrant H3K27me3 deposition in advanced lesions, and upregulation of oncogenic microRNAs (e.g., *miR-21*) and lncRNAs (e.g., *MALAT1*). Progression is illustrated using directional arrows that reflect temporal and molecular transitions between disease stages.

Although this time-dependent presentation of the pathological stages is not entirely confined to the specific conditions depicted in Table 6, it highlights the sequential accumulation of epigenetic aberrations and their potential use as biomarkers or therapeutic targets in inflammation-driven colon carcinogenesis. In this respect, Table 6 illustrates how specific epigenetic changes emerge and accumulate across the stages of IBD-associated colorectal cancer, offering a blueprint for biomarker discovery and targeted therapies. This multi-layered approach, which takes into account the associations of the diverse entities with the disease, e.g., causal genes, epigenetic modulators and environmental cues in their networks of interactions, which are the most critically important for the pathophysiology of the disease phenotype, are summarized on the whole, in Table 6.

### 7.4. Other ncRNAs in GI Diseases

Small RNAs are abundant in eukaryotic organisms and play a crucial role in regulating gene expression through mRNA degradation or gene silencing. Another class of small RNAs, the Piwi sub-family of Argonaute proteins, specifically binds to piRNAs [298]. The latter gene regulation complex is vital because it is involved in biological processes such as cell renewal and differentiation of stem cells [299], animal development [300], germline cell development [301], and certain types of human cancer [302].

#### 7.4.1. PiRNAs

PIWI-interacting RNAs (piRNAs) are small RNA molecules approximately 26 to 31 nucleotides in length [303]. Their functions include regulating mRNA expression of transposable elements through degradation, potentially inhibiting translation, and modifying chromatin [304]. This surveillance system is essential for controlling and degrading cancer cells [305]. Beyond this, piRNAs contribute to the silencing of transposable elements through methylation, participate in the deadenylation of Drosophila transcripts [306], promote de novo DNA methylation [307], and play a role in imprinting of the Rat sarcoma guanine nucleotide releasing factor 1 (*Rasgrf1*) locus in rats [308].

PIWI-interacting RNAs are derived from single-stranded transcripts of non-coding regions, which often contain a high density of truncated transposable element sequences or 3′ UTR transcripts from protein-coding genes. The bulk of these transcripts is derived from genomic regions that serve as stable sources of piRNA precursors, designated as clusters [309]. These clusters correspond to the repertoire of transposable elements, which are then recognized and silenced. The transcripts from these clusters are processed within the cell into a large pool of piRNAs in a largely random fashion. Despite interspecies and individual sequence differences, certain features remain conserved, including the presence of a uridine at the 5′ end and an adenosine at the 10th nucleotide position [310]. The genomic localization of piRNA clusters is also conserved in mammals [311].

Two main models explain piRNA biogenesis. In the first model, a piRNA cluster transcript undergoes cleavage, producing a 5′ end. The transcript may be cleaved at almost any position, though cleavage tends to favor a 5′ uridine terminal. Following binding to a Piwi protein, a second cleavage defines the RNA’s 3′ end [312]. The second model, known as the ping-pong amplification cycle, begins with the production of large numbers of piRNAs from both sense and antisense strands of the cluster, similar to the first model. When the Piwi–piRNA complex encounters its target, it cleaves the transcript 10 nucleotides downstream of the piRNA’s 5′ end. This cleavage both inactivates the target mRNA and generates a new RNA fragment with a 5′ end that associates with the Argonaute protein AGO3. This new complex then binds complementary sequences, such as cluster transcripts, leading to multiple piRNA fragment products identical to the initial ones. This ping-pong mechanism, therefore, enables the amplification of vast numbers of piRNAs from a limited original sequence pool [312]. The Piwi–piRNA complexes assemble in the cytoplasm and are subsequently transported into the nucleus, where they function to repress transposons and recruit histones to these loci. Importantly, sustained Piwi–piRNA activity is required to maintain transposon silencing, since diminished Piwi function rapidly results in reactivation of transposable elements [309]. Small RNAs, such as Piwi, are present in eukaryotic organisms and are highly active in regulating gene expression through mechanisms like mRNA degradation and gene silencing. Argonaute proteins direct these mature small RNAs to their specific targets [313]. Within this family, the Piwi subfamily of Argonaute proteins shows a stronger and more specific association with piRNAs [298]. The regulatory role of this complex is crucial, as it is linked to essential biological processes, including cellular renewal and stem cell differentiation [299], organismal development [300], germline cell development [301], and the progression of certain types of human cancers [302].

Table 7 below shows representative results of the involvement of piRNAs in IBD-related syndromes as well as relevant citations for further reading. All the databases employed by the first author in these analyses of piRNAs, concerning the associations between piRNAs and disease phenotypes, were a few, e.g., piRDisease v1.0, piRNAQuest V.2, piRPheno [314,315,316], and were based on the manually curated associations of experimentally supported piRNAs with CRC. The piRNAs involved in colorectal cancer are listed with their main characteristics (i.e., SNP information, SNP expression info, function and supportive publication reference (PMID)). For the piRNA expression profile of the top 5 most abundant piRNAs in colorectal-related carcinogenesis, the reader can refer to Appendix A), as explained below.

Further research can be provided by piRNA online databases and biotools for analysis and updated curated information. The latter approach has become increasingly critical due to the importance of relationships between piRNAs and disease phenotypes [316]. Therefore, the first author employed piRBase V2.0 database, which systematically integrates epigenetic and post-transcriptional regulation data to support piRNA functional analysis [317,318]. Also, piRPheno v2.0, a manually curated database platform, which provides experimentally supported associations between piRNAs and disease phenotypes with validated novel class of biomarkers and potential drug targets for disease diagnosis, therapy and prognosis. The work performed on the aforementioned databases, including also search and analysis of available and pertinent GEO datasets to colorectal-associated neoplasmatic pathologies at the gene expression level, as shown in Appendix A). Results of the piRNA expression profile of the top 5 among 200 most abundant piRNAs, which were identified in the samples, are shown in Appendix A, under the column piRNA ID. In more detail, Appendix A presents the piRNA expression profile of the top 5 most abundant piRNAs detected in the samples (GEO, NCBI) by RNA-seq analysis in colorectal-related carcinogenesis. For the specific analysis, piRNAQuest V.2 database was additionally employed, which is a comprehensive updated resource for piRNAs of 28 species with 9,277,689 unique piRNAs, density-based cluster prediction and piRNA expression, corresponding to different tissues and diseases [315]. These new findings suggest that further research is of utmost importance as a goal to elucidate the nature and involvement of piRNAs in diagnosis and/or potential therapy of these diseases by scrutinizing their action as clinical markers.

#### 7.4.2. CircRNAs

Circular RNAs (circRNAs) represent a subclass of ncRNAs characterized by cell- or tissue-specific expression and evolutionary conservation across species [319]. Compared to their linear counterparts, circRNAs are more stable, predominantly localized in the cytoplasm, and play important roles in human development and disease [319]. Initially discovered decades ago and dismissed as splicing errors, they were later recognized—by the advent of next-generation sequencing—as abundant, stable, and conserved among eukaryotes, and structurally form covalently closed continuous loops through back-splicing or lariat mechanisms [319]. To date, bioinformatic analyses have predicted tens of thousands of circRNAs, broadly categorized into exonic circRNAs, circular intronic RNAs (ciRNAs), and exon-intron circRNAs (EIciRNAs) [319]. They have been implicated in a wide range of physiological and pathological processes, including tumorigenesis, neurodegeneration, cardiovascular diseases, immune dysregulation, and metabolic disorders, and are also known to accumulate during aging. In the GI tract, circRNAs have recently garnered attention for their potential roles in epithelial homeostasis, inflammation-driven signaling, and CRC progression, functioning as miRNA sponges, RBP scaffolds, or even templates for translation. Given their structural stability and conservation, circRNAs are increasingly investigated as diagnostic biomarkers and therapeutic targets in GI diseases. In this section, pre-analyzed human colon tissue expression datasets by the first author are integrated to highlight the Top 30 most abundantly expressed circRNAs, quantified as CPM (counts per million) in a one-step RNA-seq analysis pipeline (Figure 3), searched under the circAtlas database [320]. These circRNAs represent highly transcribed and potentially functional molecules relevant to mucosal biology, inflammation, and colorectal carcinogenesis. CircAtlas 3.0, integrates more than 3.1 million circRNAs across 10 species (human, macaque, mouse, rat, pig, chicken, dog, sheep, cat, rabbit) as well as a variety of tissues, thus integrating the most comprehensive circRNAs, their expression and functional profiles in vertebrates.

The profile of the top 30 most highly expressed circRNAs in normal human colon tissue (Figure 3) shows abundances ranging from approximately 5.5 to 1 CPM. While many of these circRNAs have not yet been characterized in the context of GI pathology, circRNAs such as *circITGA7* (and others not included in Figure 3, such as *hsa_circ_0000711*, *hsa_circ_0000231*, and *circ_0062682*)—though not necessarily among the most expressed in normal tissue—have well-documented roles in colorectal cancer. For instance, *circITGA7* is downregulated and correlates with better prognosis (AUC ≈ 0.88), sponging *miR-370-3p* to inhibit Ras signaling via NF1; *hsa_circ_0000231* is significantly upregulated (~4.6-fold) in CRC and promotes proliferation; *circ0062682* is overexpressed across multiple CRC cohorts and associates with poor survival, via miR-940/PHGDH regulatory circuit [321,322,323,324].

In Table 8 below are presented circRNAs involved in colorectal cancer. These findings underscore a potential functional divergence: highly expressed circRNAs in normal colon may maintain tissue homeostasis, whereas dysregulated circRNAs in CRC often show altered expression rather than absolute abundance. Future work should map overlaps between the expressed circRNAs in this dataset and the dysregulated CRC-associated circRNAs to identify candidates for functional validation and biomarker utility.

Table 8 presents examples of circRNAs involved in colorectal cancer, which are listed with their main characteristics (i.e., circRNA names, predicted interactions with RBP, expression pattern, function, database recorded, method of detection and supported by publication reference ID (PMID) from NCBI).

#### 7.4.3. Functions and Implications of circRNAs

CircRNAs have emerged as important regulators of transcription [320]. They can function as miRNA sponges or bind RNA-associated proteins to form RNA–protein complexes that influence gene transcription [319]. Certain ciRNAs and EIciRNAs also interfere with pre-mRNA splicing, a key step in post-transcriptional gene regulation, while their lack of 3′ termini renders them resistant to RNase R-mediated degradation, giving them higher stability than linear RNAs [319]. Importantly, their abundance in blood, saliva, and exosomes underscores their potential as diagnostic biomarkers, with growing evidence linking them to a range of diseases, including Alzheimer’s disease, diabetes, gastrointestinal disorders, such as IBD and colitis, and most notably colorectal cancer (CRC) [325,326].

To further explore the landscape of circRNAs in GI inflammation and tumorigenesis, the first author searched and analyzed circRNA profiles derived from extracellular vesicles (EVs), which are known mediators of intercellular communication, using the online biotool exoRBase 2.0 [327]. Thus, Appendix A) presents circRNAs identified from EVs in healthy colon and small intestine tissues, while Appendix A highlights circRNAs enriched in EVs isolated from colorectal cancer (CRC) samples, all detected through RNA-seq analysis searched and retrieved from exoRBase 2.0. These datasets provide insight into the differential expression, genomic origin, and potential regulatory roles of circRNAs in both homeostatic and malignant contexts. Their EV-based localization suggests roles in systemic signaling and possibly in non-invasive biomarker development. The aforementioned tables show and expand the biological relevance of circRNAs beyond intracellular activity, supporting their potential as circulating molecular indicators in IBD and CRC progression.

Notably, it is clear that circHlPK3 is the highest CPM signal, as shown in Figure 3, along with high fold-changes or altered expression in CRC compared to normal tissues (Table 8). The latter provides dual evidence—i.e., circHlPK3 baseline high abundance and disease-specific overexpression (Table 8)—which underscores its biological relevance in gastrointestinal tissue homeostasis and malignant transformation. According to a key study (PMID: 33536039, Table 8), circHIPK3 promotes CRC progression by sponging miR-7, thereby modulating the expression of multiple oncogenes, including Focal Adhesion Kinase (*FAK*), epidermal growth factor receptor (*EGFR*), and the transcription factor (TF), Yin Yang 1 (*YY1*). These findings suggest that *circHIPK3* may act as an endogenous miRNA sponge, contributing to tumor cell proliferation, migration, and invasion.

#### 7.4.4. Circular RNA-Protein Interactions: Functional Significance and Binding Site Density

Furthermore, circRNAs modulate gene expression and cellular pathways through interactions with TFs and RBPs. Strictly speaking, for circRNA binding capacity, the number and affinity of RBP binding sites on circRNAs significantly influence their regulatory potential: multiple binding sites enable circRNAs to act as molecular “scaffolds,” sequestering RBPs away from pre-mRNAs, modulating splicing, translation, or protein stability [328,329,330,331].

These interactions often hinge on the number of binding sites, which can modulate the efficiency and strength of circRNA–protein interactions. For instance, a circRNA with multiple HuR (Human Antigen R)-binding motifs may more effectively sequester HuR, altering mRNA stability and downstream gene expression [330]. This ribonucleoprotein assembly could influence key processes, including RNA splicing, translation, and cellular localization, thereby contributing to oncogenic or tumor-suppressive pathways in inflammation-driven CRC. The density of RBP binding sites correlates with competitive binding kinetics—higher counts increase the likelihood of functional RBP sequestration or scaffold formation. Additionally, some RBPs may recruit chromatin-modifying complexes, linking circRNA–protein interactions to epigenetic regulation. For instance, abundant binding sites on oncogenic circRNAs retain RBPs involved in tumor suppression or RNA processing, thus tipping the cellular balance toward proliferation and survival. Moreover, circRNAs may compete with parental linear RNAs for RBP binding, further altering gene expression profiles. Therefore, mapping RBP-binding landscapes on circRNAs is critical to understanding circRNAs’ functional complexity in tumor progression and their roles in inflammation-induced cancers like IBD-CAC and CRC, while RBPs may serve as novel biomarkers or targets in CRC therapeutics [332,333,334].

In Table 8, information is also shown concerning the predicted interacting RBP (Nο of binding sites) as a score-dependent evaluation of the circRNA’s potential binding; thus, correlating with their potential competitive binding affinity to RBPs and sequentially recruiting chromatin-modifying complexes, as linking circRNA–protein interactions expand epigenetic regulation.

### 7.5. Inflammation-Driven ncRNA Modulation in CRC

Chronic inflammation is a well-established contributor to the initiation and progression of colorectal cancer (CRC), particularly in the context of inflammatory bowel disease (IBD). Among the key mediators of inflammation-driven tumorigenesis, various types of non-coding RNAs (ncRNAs) are involved, which some have already discussed, mainly including miRNAs, lncRNAs, piRNAs and circRNAs. These ncRNAs act as post-transcriptional and epigenetic regulators of gene expression, responding to pro-inflammatory signals and reshaping the cellular transcriptome to favor survival, proliferation, and malignant transformation.

In Table 9, known examples of inflammation-associated ncRNA clinical markers of various types, involved in inflammation-driven colorectal cancer, altered in patients with CRC, their functions, and supporting evidence (PMID) are highlighted. Thus, Table 9 contains a concise selection list of ncRNAs that have revealed potential or are under investigation for validation as biomarkers/clinical markers, emanated from Table 4, Table 5, Table 6 and Table 7, in IBD/CRC conditions.

In the inflamed colonic microenvironment, immune cells release cytokines such as IL-6, IL-1β, and TNF-α. These cytokines activate transcriptional programs through pathways like NF-κB, STAT3, and Wnt/β-catenin, which in turn influence the expression of oncogenic or tumor-suppressive ncRNAs. For example, *miR-21*, a highly upregulated miRNA in both IBD and CRC, is transcriptionally induced by NF-κB and STAT3 activation. More importantly, it targets tumor suppressors such as *PTEN*, *PDCD4*, and Reversion-Inducing Cysteine Rich Protein With Kazal Motifs (*RECK*), thereby enhancing PI3K/AKT signaling and resistance to apoptosis [335,336].

Similarly, *miR-155* is induced in macrophages and epithelial cells during chronic inflammation. It modulates immune tolerance and tumor progression by targeting negative regulators of cytokine signaling like Suppressor Of Cytokine Signaling 1 (*SOCS1*) and Src homology 2 (SH2) domain containing inositol polyphosphate 5-phosphatase 1 (*SHIP1*), contributing to a feed-forward loop that sustains STAT3 activation and inflammatory gene expression [337,338].

Beyond miRNAs, inflammation-responsive lncRNAs play pivotal roles in chromatin remodeling and transcriptional repression of tumor suppressor genes. The lncRNA *HOTAIR*, for instance, is upregulated in colonic epithelial cells under inflammatory stress. It recruits PRC2 to silence genes involved in epithelial homeostasis and immune control, such as *CDH1* (E-cadherin), facilitating EMT and metastasis [270]. The inflammation-sensitive lncRNA, *MALAT1*, is elevated in CRC and modulates alternative splicing of genes linked to migration and invasion [339]. Notably, both *HOTAIR* and *MALAT1* participate in chromatin remodeling and transcriptional regulation, further enhancing inflammation-linked oncogenic transformation.

Importantly, circular RNAs have recently emerged as key regulatory molecules in the inflammation–cancer axis. An important paradigm, circular RNA *HIPK3* (Table 8 and Table 9), is highly expressed in CRC, and acts as a sponge for 1207-5p, thereby derepressing oncogenes like formin-like 2 (*FMNL2*) [340]. In more detail, *circHlPK3* is reported to confer chemoresistance via sponging *miR-637* and promoting autophagy in CRC, suggesting its interaction with RBPs like *IGF2BP1* and HuR, which may stabilize target mRNAs related to autophagy and survival pathways (Table 8 and Table 9). Furthermore, *circCCDC66*, another circRNA elevated during chronic inflammation, sequesters tumor-suppressive miRNAs and enhances the expression of MYC Proto-Oncogene, BHLH Transcription Factor (*MYC*), Zinc finger E-box binding homeobox 1 (*ZEB1*), and other targets promoting tumor growth [341]. It has oncogenic roles in proliferation, migration, therapy resistance, and promotes autophagy by sponging *miR-3140*. Binding to HuR and Polypyrimidine Tract Binding Protein 1 (*PTBP1*) may similarly modulate mRNA stability of proliferative or apoptotic genes (Table 8).

Taken together, inflammation-driven modulation of ncRNAs constitutes a critical epigenetic layer of carcinogenic regulation in IBD-associated CRC. These ncRNAs function both upstream and downstream of canonical signaling pathways, creating a dynamic regulatory network that shapes cell fate in the inflamed epithelium, illustrated in Figure 4.

Figure 4 summarizes key molecular circuits where inflammation-induced epigenetic events interface with non-coding RNAs to regulate tumorigenesis in colorectal cancer (CRC). Inflammatory cytokines such as IL-6 and TNF-α, often elevated in IBD and tumor microenvironments, activate transcription factors including NF-κB and STAT3. These factors orchestrate oncogenic signaling cascades like the Wnt/β-catenin pathway, central to CRC progression. The diagram shows how epigenetic silencing of tumor suppressors, including *PTEN*, through promoter hypermethylation or histone deacetylation (e.g., H3K27me3 loss), intersects with dysregulated ncRNA networks.

MicroRNAs such as *miR-21* and *miR-155*—frequently upregulated in CRC—are downstream targets of NF-κB/STAT3 signaling and are known to repress tumor suppressors, including *PTEN* and pro-apoptotic genes. In more detail, *miR-21*, commonly upregulated in various cancers, promotes cell survival by suppressing Apoptotic Protease Activating Factor 1 (*Apaf-1*) and Fas cell surface death receptor (Fas) ligand, key mediators of intrinsic and extrinsic apoptosis [342]. Its anti-apoptotic role is supported by evidence that ectopic *miR-21* expression protects cells from gemcitabine-induced apoptosis [343]. Similarly, *miR-155*, often overexpressed in malignancies, is activated by TGF-β/SMAD4 signaling and facilitates EMT by targeting RhoA GTPase, a key regulator of cell polarity and junctions. Silencing *miR-155* inhibits TGF-β-induced EMT and cell invasion [344]. In contrast, miR-200 and *miR-203* [230]—suppressors of EMT—are downregulated by TGF-β, highlighting the dual regulatory role of miRNAs in metastasis. These miRNAs are further regulated by circular RNAs like *circHIPK3* and *circCCDC66*, which function as competing endogenous RNAs (ceRNAs), sequestering oncogenic miRNAs and thus fine-tuning gene expression, but also inhibiting the expression of TGF-β, inactivating SMAD signaling pathway, and reversing EMT in gastric cancer (GC) cells; this is suggesting that *circCCDC66* is an important regulator of EMT in gastric cancer [341]. *circCCDC66*, which contains target sites for *miR-33b*, *miR-93*, *miR-510-5p* and *miR-338-3p*, by sponging different miRNAs may perform various functions while promoting CRC growth and metastasis, including induction of drug-resistance [325]. Interestingly, 88% of randomly chosen patients with colon cancer have higher levels of *circCCDC66* than randomly chosen normal subjects, indicating that the expression level of *circCCDC66* is a good indicator for the detection of colon cancer [325]. *circCCDC66* has also been reported to protect *PTEN* by sponging miRNAs such as *miR-93* or *miR-33b*, and circHIPK3 that may promote tumorigenesis via NF-κB feedback loops. The latter can be explained and supported by the fact that the knockdown of hsa_*circ0000284* (*circHIPK3*) markedly suppresses CRC cell proliferation, migration, and invasion, inducing apoptosis in vitro and inhibiting CRC growth and metastasis in vivo [325]. The diagram (Figure 4) supports the central thesis that non-coding RNAs serve as both downstream effectors and modulators of inflammation-driven carcinogenesis. These interactions—combined with stable epigenetic changes—create a self-reinforcing tumor-promoting microenvironment. Identifying and targeting these RNA-mediated axes could offer translational potential for diagnostic biomarkers and therapeutic interventions in CRC.

To expand upon the network logic depicted in Figure 4, the first author explored the complex interactions of ncRNAs by using the LncACTdb3.0 platform [345], assisted by aforementioned biotools such as EVlncRNAs 2.0, NDE IQuery and FunCoup 5 [222,346,347] to construct a large-scale, colorectal cancer-centered ceRNA interaction network. For the latter gene interaction network analysis, LncACTdb3.0 was selected as the main biotool, for its validated and experimentally supported ceRNA interactions across human and mouse models.

This interaction network, which was created from our analysis (Triantaphyllopoulos et al., unpublished results), is presented in Appendix A), by integrating three molecular tiers: long non-coding RNAs (lncRNAs), microRNAs (miRNAs), and protein-coding genes, all connected through disease-associated regulatory interactions. In more detail, by performing initially an RNA-seq meta-analysis of differentially expressed genes in the AOM/DSS model (Section 3.2.1) and cross-species comparison to the human UC, we found strong similarities. The aim was to decode the molecular landscape associated with inflammation-induced colorectal tumorigenesis and identify evolutionarily conserved, upregulated gene signatures that may act as potential biomarkers or therapeutic targets. A selection of commonly upregulated protein-coding genes and ncRNAs was further prioritized for their mechanistic involvement in CRC pathogenesis. Our query was informed by candidate protein-coding genes and non-coding RNAs identified from our unpublished RNA-seq meta-analysis results of AOM/DSS-induced CAC mouse model datasets, incorporating key gastrointestinal inflammatory disease phenotypes, such as Colon Adenocarcinoma, Colorectal Cancer, Colorectal Adenocarcinoma, Ulcerative Colitis, and Colorectal Cancer Liver Metastases. The inclusion of only human and mouse orthologs for the complex network construction was also taken into account. The total protein-coding gene list and the lncRNA genes that were used for the complex network construction, derived from the first author’s unpublished RNA-seq meta-analysis (results of the AOM/DSS), are related to CRC-related carcinogenesis (Appendix A).

The strong overrepresented candidate genes which were used for the complex network construction such as *CTNNB1*, *CD44*, *EZH2*, *AXIN2*, *MMP10*, and *WIF1* (False Discovery Rate (FDR)-adjusted *p*-values < 0.001, *** *p* < 0.001), consist of a subset of a larger discovered gene list, which formed the core input for the ceRNA network analysis (Appendix A). These top enriched genes with well-established mechanistic roles in colorectal tumor biology show concomitant upregulation in both human CRC sources (TCGA, GEP databases, etc.) as well as in the studied AOM/DSS mouse model. The findings include transcription factors, pro-inflammatory mediators, angiogenic drivers, and enzymes critical to tissue remodeling in the constructed network, accompanied by reference, underscoring their validated translational significance in CRC biology. The final network map highlights densely connected regulatory hubs and crosstalk nodes involving known CRC drivers (e.g., *TP53*, *CTNNB1*, *CD44*, *EZH2,* etc.) and inflammation-associated lncRNAs and miRNAs targeting mRNAs (e.g., lncRNAs such as *BCYRN1*, *NEAT1*, *ZEB1-AS1,* and miRNAs such as *miR-34b-5p*, *miR-221*, *miR-204-3p*, *miR-139-5p*), respectively (Appendix A). Additional technical details of the node types and biological context are provided in the accompanying Appendix A). Notably, the inferred network revealed multiple lncRNAs acting as competing endogenous RNAs (ceRNAs), potentially sequestering disease-relevant miRNAs known to regulate pro-inflammatory and immune-modulatory transcripts.

Finally, the complex gene interactions could serve as a molecular bridge between preclinical and clinical settings, providing a foundation for planning future studies of emerging strategies, targeting the most significant pathways by therapeutic interventions in colitis-associated CRC.

## 8. Diagnostic and Therapeutic Implications: Biomarkers and Epigenetic Drugs

This section integrates and highlights the translational potential of inflammation-related epigenetic changes in colorectal cancer, outlining actionable diagnostic and therapeutic strategies. As graphically depicted in Figure 1 (Section Inflammation-Induced Cancers Associated with the GI Tract), it reviews clinically validated and experimental key biomarkers such as methylated *SEPT9* promoter (approved for blood-based CRC screening), overexpressed *miR-21* (linked to disease severity and prognosis), as well as experimental lncRNA signatures such as *CRNDE*, *HOTAIR* and *MALAT1*. From a therapeutic standpoint, the section discusses anti-inflammatory agents like 5-aminosalicylic acid (5-ASA—Mesalamine) and epigenetic drugs including HDAC inhibitors (e.g., Vorinostat) and DNMT inhibitors (e.g., Decitabine), which are under clinical evaluation for CRC treatment.

One important scope, in the current review, is to present an overview of how understanding the epigenetic landscape in inflammation-associated colorectal cancer can inform precision medicine approaches to improve patient outcomes. Moreover, this integrative approach underscores how inflammation-epigenetic pathways offer novel opportunities for early detection and personalized therapeutic intervention.

### 8.1. Translational Epigenetics and RNA-Based Therapeutics in IBD/CRC

The increasing understanding of chromatin epigenetics and non-coding RNAs (ncRNAs) in the pathogenesis of inflammatory bowel disease (IBD) and its progression to colorectal cancer (CRC) has opened new avenues for clinical translation. The therapeutic strategies in IBD-associated colorectal cancer (IBD-CRC) now include molecular tools that directly target epigenetic regulators and non-coding RNAs (ncRNAs), particularly miRNAs and lncRNAs. Among the most promising of these are epigenetic-targeting drugs and RNA-based therapeutics, which are reshaping personalized medicine strategies in gastrointestinal oncology. These interventions are being investigated both preclinically and in early-phase clinical trials to halt chronic inflammation or reverse the neoplastic transformation of inflamed intestinal tissues. Within this context, Table 10 below shows top translational therapies targeting epigenetic regulators in colitis-associated colorectal cancer (CAC), including emerging agents such as 5-azacytidine and their molecular targets, also visualized in Figure 1.

### 8.2. Epigenetic Drugs and Clinical Trials in IBD-Associated CRC

DNA methyltransferase inhibitors (DNMTis) such as azacitidine and decitabine, historically used in hematologic malignancies, have shown potential in reactivating silenced tumor suppressor genes in solid tumors, including CRC. Although not yet standard in IBD-associated CRC, preclinical studies demonstrate that DNMTis can reverse inflammation-induced methylation patterns and modulate immune responses—crucial in the dysplastic transformation of IBD lesions [348]. Similarly, HDACis, including vorinostat and romidepsin, have demonstrated anti-inflammatory and anti-tumor effects by altering histone acetylation states and suppressing pro-inflammatory gene transcription [349].

Of particular interest are BET (bromodomain and extra-terminal) inhibitors, which target chromatin “readers” like BRD4. These proteins regulate oncogenic transcriptional programs, including Myc-driven pathways in CRC. BET inhibitors (e.g., JQ1, Pelabresib (CPI-0610), etc.) are undergoing clinical trials and have been shown to suppress cytokine-mediated inflammation and reduce tumor growth in preclinical models of colitis and CAC [350]. BET proteins also modulate T-cell responses, offering additional therapeutic potential in chronic inflammation and immune dysregulation.

### 8.3. RNA-Based Therapeutics in Preclinical and Clinical Use

Several ncRNAs implicated in IBD-CRC serve as both diagnostic biomarkers and therapeutic targets. *MicroRNA-21*, upregulated in inflamed and cancerous colonic tissues, targets tumor suppressors like *PDCD4* and is currently under investigation for *anti-miR-21* oligonucleotide-based therapies [336,351]. For instance, *miR-21*, frequently upregulated in colonic inflammation and early neoplasia, enhances NF-κB signaling and suppresses tumor suppressor *PDCD4* [335]. Antagomirs targeting *miR-21* have demonstrated efficacy in reducing colitis severity and tumor burden in murine models [352]. *MicroRNA-92a*, another oncogenic miRNA elevated in serum and stool samples of CRC patients, is also being explored as a therapeutic target and non-invasive biomarker [353]. These RNA therapeutics are being evaluated using mouse models of DSS- and AOM/DSS-induced colitis/CAC—mouse model analogous to the human disease—alongside human CRC organoids and patient-derived xenograft models that preserve tumor heterogeneity [351].

In parallel, *miR-155* and *miR-223* are under investigation as modulators of immune cell polarization and epithelial integrity, with recent studies confirming their differential expression and regulatory roles in ulcerative colitis and colorectal cancer models [354,355], while lncRNAs like *HOTAIR* and *CCAT1* have shown oncogenic properties in CRC. Therapeutic silencing of these molecules using antisense oligonucleotides (ASOs) or locked nucleic acid (LNA)-modified inhibitors is currently in early-phase clinical trials or advanced preclinical stages [356].

Furthermore, RNA aptamers and small interfering RNAs (siRNAs) offer precise gene silencing potential and are being explored to target inflammatory mediators, oncogenes, and DNA methyltransferases directly [357]. For example, the siRNA-based drug ALN-PCS, although initially developed for hypercholesterolemia, exemplifies the clinical viability of RNA interference-based platforms [358].

### 8.4. Diagnostic and Prognostic Utility of Epigenetic and RNA Biomarkers

Several circulating epigenetic and RNA markers have shown promise as non-invasive tools for early detection and surveillance. Stool- (Cologuard in USA and ColoAlert in Europe, approved by FDA) or blood-based assays detecting methylated *SEPT9***,**
*VIM***,** or *SFRP2* genes [359,360], as well as panels combining *miR-21*, *miR-92a*, and *miR-135b*, have demonstrated high sensitivity for early CRC detection [353]. Stool DNA methylation testing, specifically Cologuard, is an FDA-approved assay for colorectal cancer (CRC) screening. It detects DNA markers and occult hemoglobin in stool samples, helping identify the presence of colorectal cancer or advanced precancerous lesions. The test has been updated with Cologuard Plus, a next-generation version that improves sensitivity and stability. Moreover, patterns of miRNA expression are increasingly being integrated into risk models for dysplasia surveillance in patients with UC and CD.

### 8.5. Comparative Human-Mouse Evidence

Preclinical mouse models of colitis and colorectal cancer remain indispensable for dissecting molecular mechanisms that are often challenging to study directly in patients. By mirroring key pathological features of human IBD and CAC, these systems provide a platform to interrogate genetic, epigenetic, and transcriptomic changes under controlled conditions. Importantly, cross-species comparisons allow validation of candidate pathways and biomarkers, strengthening their translational relevance for diagnostics and therapeutics.

For example, in DSS-induced mouse colitis models, HDAC inhibition reduced NF-κB activity and prevented histological inflammation, paralleling human biopsy studies showing HDAC1/2 overexpression in UC patients [348,361]. Similarly, *miR-21* overexpression has been validated in both AOM/DSS murine CAC (Figure 2), zebrafish tissues, as well as human colorectal carcinoma biopsies, linking inflammation and carcinogenesis via shared transcriptomic patterns [335,336,351].

Referring to the expanded overview of murine models in Section 3 and among the various animal models for IBD and CAC, the AOM/DSS mouse model stands out as one of the most widely used, as it successfully combines inflammation-driven carcinogenesis with genetic mutation induction. Its broad adoption highlights not only its reliability in reproducing colitis and colitis-associated cancer but also its strong translational value for uncovering molecular signatures relevant to human CRC—one of the key reasons the first author selected it for conducting the RNA-seq meta-analysis study.

## 9. Concluding Remarks and Future Outlook

The interplay between genetic predisposition, chronic inflammation, and epigenetic dysregulation is central to the pathogenesis of inflammatory bowel disease (IBD)-associated colorectal cancer (CRC). As detailed in this review, chromatin-modifying enzymes, DNA methylation machinery, and regulatory non-coding RNAs orchestrate dynamic transcriptional changes in both immune and epithelial compartments, driving the progression from inflammation to neoplasia.

At the forefront of translational breakthroughs are several clinically actionable genetic markers such as *TP53*, *KRAS*, *APC*, *BRAF*, and *SMAD4*, which not only serve as predictors of malignant transformation but also guide treatment decisions. Simultaneously, epigenetic regulators such as *DNMT1*, *HDACs*, *EZH2*, and BET proteins are gaining attention as both biomarkers and druggable targets, particularly as next-generation epigenetic inhibitors advance through clinical pipelines. Adding another dimension, proteomic studies have unveiled dysregulated pathways involving key inflammation-related proteins (e.g., IL-6, STAT3, TNF-α), tight junction components (e.g., claudins, occludin), and tumor microenvironment factors (e.g., MMP9, VEGF), many of which are now being profiled longitudinally to monitor disease progression and therapeutic response. Moreover, the incorporation of advanced imaging modalities—including CT, MRI, positron emission tomography (PET)/CT, and emerging Electron Paramagnetic Resonance (EPR)-based molecular imaging—into clinical practice allows for early lesion detection and real-time biomarker mapping. Novel imaging biomarkers, particularly those targeting integrin expression or metabolic pathways, are now integrated into multimodal biomarker discovery platforms, offering a powerful complement to genetic and transcriptomic profiling.

Importantly, the convergence of multi-omics technologies, liquid biopsy tools (e.g., cell-free DNA methylation and circulating ncRNAs), and artificial intelligence-based imaging analytics is transforming biomarker discovery from siloed datasets to holistic, integrative diagnostics. These advancements enable not only early CRC detection in IBD patients but also real-time stratification and tailored therapeutic decisions.

Thus, the sidebar infographic (Figure 5) below provides a visual overview of the socioeconomic and translational dimensions of precision medicine in IBD and CRC, highlighting key areas such as cost-effectiveness, healthcare optimization, and implementation strategies. This is a brief review of the Socioeconomic impact of precision medicine in IBD and CRC.

Future efforts must prioritize clinical validation of composite biomarker panels, refinement of RNA delivery technologies, and the integration of omics data with radiogenomic signatures. Together, these efforts will bridge the bench-to-bedside gap, allowing for truly personalized, predictive, and preventive strategies in IBD-associated CRC.

Socioeconomic and Health System Impact: Beyond scientific innovation, the integration of epigenetic and genetic biomarkers into routine clinical care carries profound socioeconomic implications.

In more detail, this infographic summarizes key socioeconomic metrics and the potential of precision medicine in IBD and CRC management. Most precision interventions are cost-effective compared with standard care (PMID: 31650223). Evidence also shows that tailored precision strategies, such as targeted antibiotic-based interventions, can reduce hospitalization (PMID: 40001454). In Europe, financial incentives and innovative reimbursement frameworks are emerging as important drivers of sustainable implementation (PMID: 37623911). Broader value frameworks further highlight the importance of capturing patient-centered, long-term, and societal benefits when assessing returns on investment in precision medicine (PMID: 32389217). Together, these findings suggest that precision approaches can minimize unnecessary interventions, improve resource allocation, and strengthen healthcare system resilience.

The integration of epigenetic and genetic biomarkers into clinical practice carries profound socioeconomic implications. Furthermore, precision medicine in IBD and CRC promises to alleviate rising healthcare costs, reduce disparities in access to molecular diagnostics, and optimize therapeutic efficacy. Evidence suggests that precision approaches can be cost-effective compared to standard care (PMID: 31650223). In the European context, financial incentives and innovative reimbursement frameworks are emerging as crucial catalysts for sustainable implementation (PMID: 37623911). Broader value frameworks emphasize the need to capture patient-centered, long-term, and societal benefits when evaluating return on investment (PMID: 32389217). Beyond IBD and CRC, the wider applicability of targeted care has been demonstrated in infectious disease settings, such as urinary tract infections, where biomarker- and precision-based strategies can reduce hospitalization and improve cost efficiency (PMID: 40001454).

Although upfront investments in infrastructure—including interoperable digital health platforms, genomic repositories, and workforce training—are substantial, the long-term benefits are considerable. By decreasing ICU admissions, shortening hospital stays, and minimizing recurrence rates, precision medicine strengthens system resilience. The Organisation for Economic Co-operation and Development (OECD) has emphasized that targeted technologies can be cost-effective and that policy modernization is essential for efficient resource allocation. Taken together, these considerations highlight how predictive and preventive medicine can redefine cost-efficiency, moving healthcare systems toward early intervention strategies that are not only relevant for IBD and CRC but broadly applicable across human disease.

## Figures and Tables

**Figure 1 ijms-26-09498-f001:**
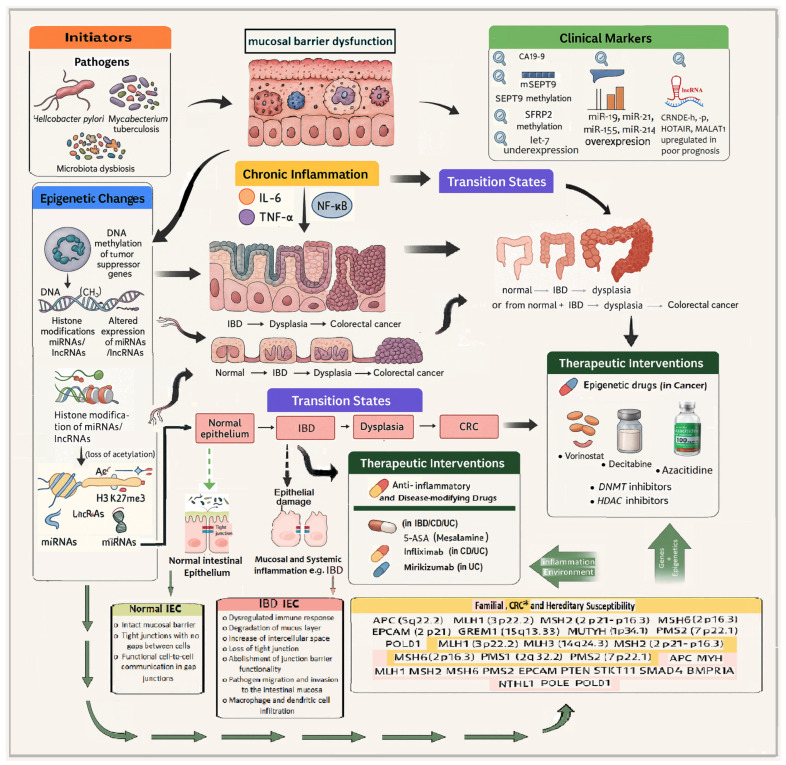
Mechanistic Pathway—From Inflammation to Colorectal Carcinogenesis. This schematic illustrates the multi-step cascade of molecular and cellular mechanisms involved in the transition from chronic intestinal inflammation to inflammation-driven colorectal cancer (CRC) development, particularly in the context of inflammatory bowel disease (IBD), with a focus on genetic and epigenetic regulation. Notes: Bottom right call-out. Familial disease (in pink background); CRC* predisposition of tumor suppressor genes promoted by epigenetic changes (in orange background); Hereditary Susceptibility genes to colorectal neoplasia (in white background). (see details in the text).

**Figure 2 ijms-26-09498-f002:**
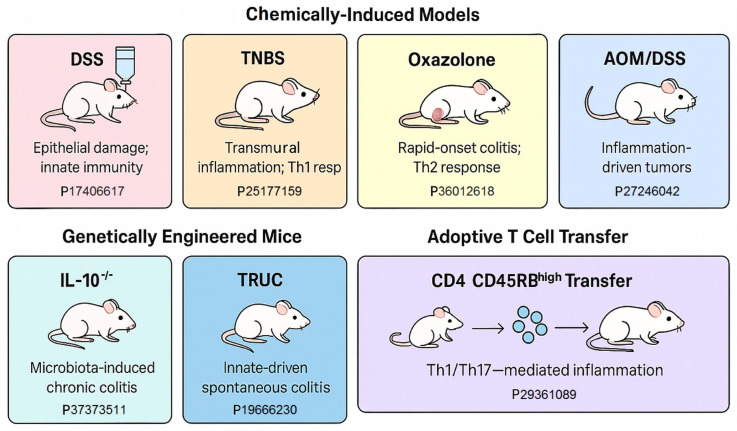
Sidebar Infographic: Commonly Used Animal Models for Studying Inflammatory Bowel Disease and CAC. A schematic overview of widely employed murine models used in IBD and colitis-associated colorectal cancer (CAC) research. These include chemically induced models (DSS, TNBS, Oxazolone, AOM/DSS), genetically engineered mice (*IL-10^−/−^*, *TRUC*, *APC^Min/+^*), and adoptive T cell transfer models. Each model has distinct immune engagement and phenotypic features, supporting diverse aspects of inflammation and tumorigenesis. Notes: P number in the bottom of each rectangular corresponds to the published reference (PMID). The blue dots in the bottom right rectangle represent the adoptive T cell injection to the mice.

**Figure 3 ijms-26-09498-f003:**
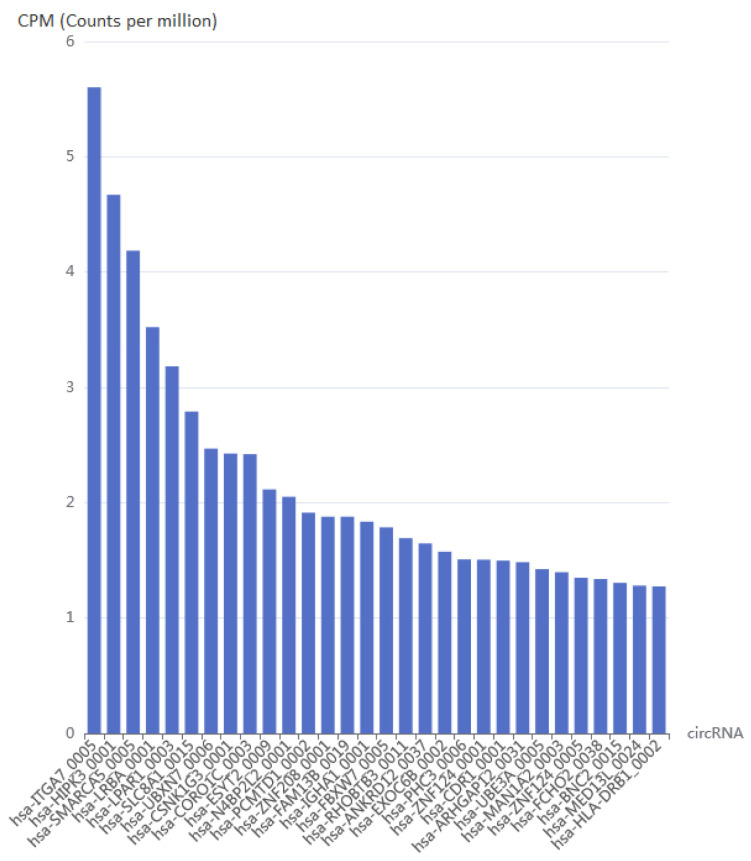
Thirty most highly expressed circRNAs from preanalyzed datasets of human colon tissue (one-step analysis) in CPM (counts per million). Data Accessed 16 June 2025, from circAtlas 3.0 database [320] (see details in the text).

**Figure 4 ijms-26-09498-f004:**
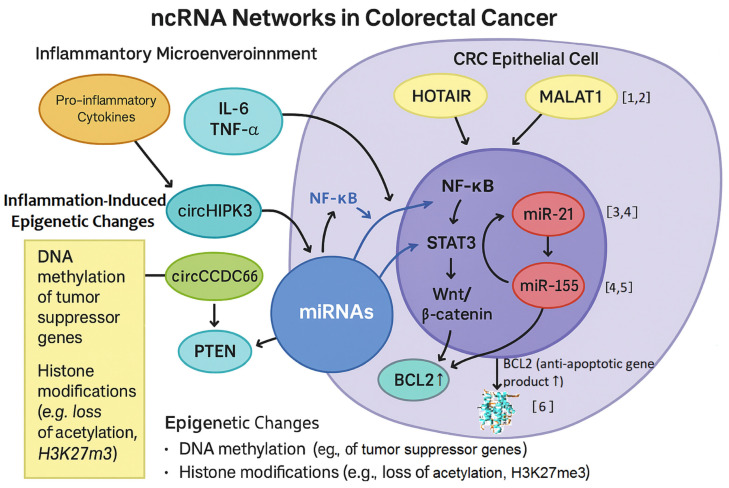
Non-coding RNA Networks in Inflammation-Associated Colorectal Cancer. The above diagram illustrates the interactions between key ncRNAs and inflammation-related signaling cascades within the colorectal tumor microenvironment. The upper region depicts inflammatory cytokines (e.g., IL-6, TNF-α) activating intracellular pathways such as *NF-κB*, *STAT3*, in CRC epithelial cells, which in turn regulate the expression of oncogenic miRNAs and interact with Wnt/β-catenin in colonic epithelial cells. These signals modulate the transcriptional expression of lncRNAs (*HOTAIR*, *MALAT1*), miRNAs (*miR-21*, *miR-155*) and circRNAs (*circHIPK3*, *circCCDC66*) in CRC pathogenesis. Circular RNAs act as sponges modulating miRNA activity, such as *circCCDC66* protecting *PTEN* from miRNA-mediated repression, while *circHIPK3* affecting NF-κB signaling. Downstream key targets include tumor suppressor genes (e.g., *PTEN*, *PDCD4*) and oncogenes (e.g., *BCL2*, Wnt/β-catenin), leading to increased cell survival, proliferation, and evasion of apoptosis. The diagram portrays chromatin-level epigenetic changes, including DNA methylation and histone modification (e.g., H3K27me3), which are orchestrated by ncRNA-interacting complexes such as PRC2. This integrated view underscores the central role of ncRNAs in mediating inflammation-associated carcinogenesis. Notes: The references in brackets refer to PMIDs, which support the interactions. PMIDs: [1], [35305641]; [2], [35961438]; [3], [29263891]; [4], [36733201]; [5], [24165275]; [6], [32380907]. Bold black fonts (NFκB, STAT3, Wnt/β-catenin) indicate transcriptional hubs activated in the nucleus, connected by black arrows (canonical activation). Light blue fonts (e.g., NFκB) denote transcription factors when directly targeted by miRNAs, shown by thin blue inhibitory arrows. The circHIPK3 black arrow “attacking” the miRNAs means that circHIPK3 functions as a competing endogenous RNA (ceRNA): it binds and sponges miRNAs, thereby reducing their availability to repress targets like NFκB or PTEN. Black arrows from extracellular factors (IL-6, TNFα) reflect cytokine-induced pathway activation feeding into the nucleus.

**Figure 5 ijms-26-09498-f005:**
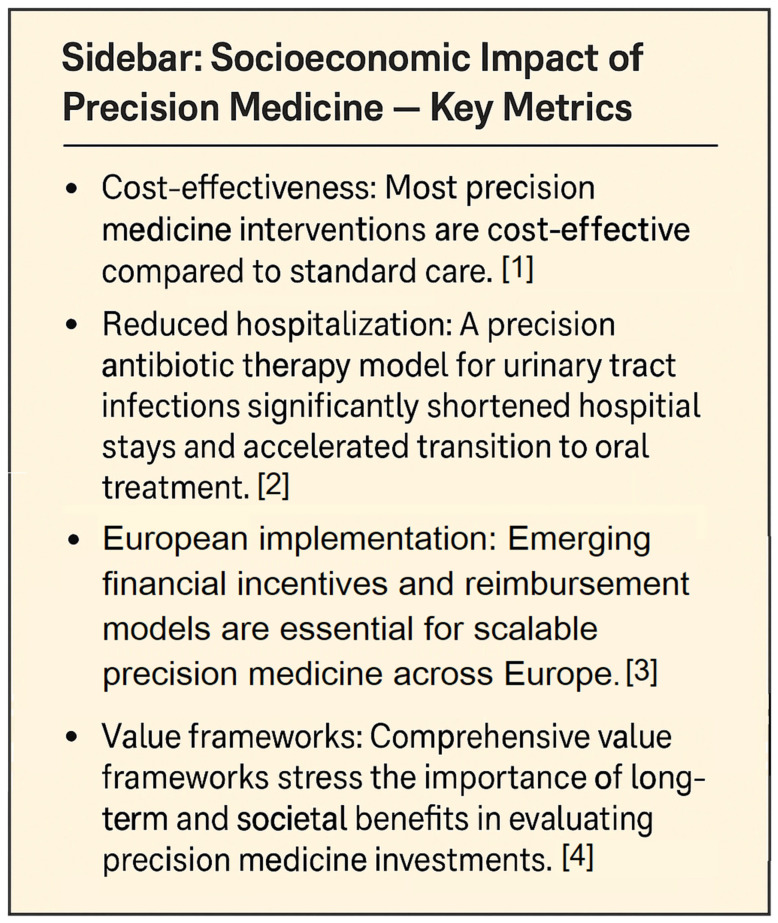
Sidebar Infographic: Socioeconomic Impact and Precision Medicine in IBD and CRC. Notes: PMIDs, [1]: 31650223, [2]: 40001454, [3]: 37623911, [4]: 32389217.

**Table 1 ijms-26-09498-t001:** Murine Models of Inflammatory Bowel Disease (IBD) and Colitis-Associated Colorectal Cancer (CAC).

Model	Method	Immune System Involvement	Advantages	Limitations	Reference (PMID)
DSS	Oral DSS in drinking water	Innate	Rapid, simple, epithelial injury	No adaptive immune involvement	17406617 36012618 34440615
AOM/DSS	AOM injection + DSS cycles	Innate + DNA damage	Models CAC pathogenesis	Less suitable for sporadic CRC	27246042
TNBS	Rectal instillation of TNBS	Adaptive (Th1)	Crohn’s-like inflammation	Variability, toxic risk	25177159 34440615
Oxazolone	Chemical	Adaptive (Th2)	Mimics UC; rapid onset	Short-lived; strain-dependent	36012618
IL-10^−/−^	Genetic deletion of IL-10	Adaptive	Spontaneous colitis	Microbiota dependence	37373511
TRUC (T-bet^−/−^ × RAG^−/−^)	Double knockout	Innate	Innate immunity-driven model progressing to colonic dysplasia and rectal adenocarcinoma	Complex breeding, microbiota sensitive	19666230
APC^Min/+^	Genetic (multiple intestinal neoplasia)	Sporadic Intestinal Tumors	FAP model; Wnt pathway activation	Small intestine focus	30887153
CD4^+^ CD45RB^high^ T Cell Transfer to RAG^−/−^ Mice	Adaptive cell transfer	Adaptive	T-cell-driven colitis	Requires expertise, chronic model	29361089 34440615 25989337

**Table 2 ijms-26-09498-t002:** DNA Methylation as diagnostic and therapeutic biomarkers according to the subtype of IBD.

Disease Type	Sample Type	Methylated Markers	Methylation Status	Reference (PMID)
IBD	Rectal biopsies	*THRAP2*, *FANCC* and *GBGT1*	↑	22419656
CD	Blood	*WDR8* and *ITGB2*	↑	27886173
CD	Rectal biopsies	*DOK2* and *TNFSF4*	↓	22419656
CD	Blood	*VMP1*	↓	27886173
UC	Rectal biopsies	*CARD9* and *CDH1*	↑	22419656
UC	Blood	*WDR8*	↑	27886173
UC	Rectal biopsies	*ICAM3*, *DOK2* and *TNFSF4*	↓	22419656
UC	Blood	*VMP1*	↓	27886173
UC	Colon biopsies	*EBI3*	↓	22419656
CRC	Rectal biopsies	*TGFB2*, *SLIT2*, *HS3ST2*, *TMEFF2*,	↑	27886173
CRC	Colon biopsies	*FOXE1*, *SYNE1*	↑	22419656
CRC	Colonic mucosa	*APC*, *CDH13*, *MGMT*, *RUNX3* and *MLH1*	↑	27886173
CRC	Colon biopsies	*ITGA4*	↑	34069352

Notes: ↑, DNA methylation increase; ↓, DNA methylation decrease.

**Table 3 ijms-26-09498-t003:** Histone modifications as diagnostic and therapeutic biomarkers according to the subtype of IBD.

Disease Type	Sample Type	Histone Modifying Enzymes/Histone Markers	Modification Type/Status	Reference (PMID)
IBD	Intestinal tissue	KAT2B	Acetylation ↓	26802082
UC/CAC	Colon Tissue	H3K27ac	Acetylation ↑	29983891
CD	Colonic biopsy	H2Bub1	Ubiquitination ↓	34088983
IBD	Colon Tissue	HDAC	Acetylation ↑	38903915
IBD	Colon Tissue	HDAC8	Acetylation ↑	36558966
CRC	Primary cancer tissue	H3K9me2	Methylation ↑	22076537

Notes: ↑, increase in Post-translational modification; ↓, decrease in Post-translational modification.

**Table 4 ijms-26-09498-t004:** MicroRNAs as diagnostic and therapeutic biomarkers according to the subtype of IBD and CRC.

Disease Type	Sample Type	miRNAs	Gene Expression Change	Reference (PMID)
UC	Serum	*miR-29a*, *miR-196b*, *miR-127-3p*	↑	23607522
CD	Serum	*miR-140-3p*, *miR-127-3p*	↑	23607522
UC	Serum	*miR-150*	↓	23607522
CD	Blood	*miRNA* *-125a*	↓	29209130
UC	Blood	*miRNA* *-19a*	↑	25886994
UC	Blood	*miRNA* *-146a*	↓	25886994
CD	Blood	*miR-31*	↓	25886994
CD	Blood	*miR-101*	↑	25886994
IBD	Serum	*miR* *-146b* *-5p*	↑	30734320
IBD	Feces	*miR-223, miR-1246*	↑	32850969
IBD	Serum	*miR-16*, *miR-21*, and *miR-223*	↑	29668922
IBD	Feces	*miR-223*, *miR-155*	↑	29668922
IBD	Serum/Feces	*miR-21*, *miR-142-3p*, *miR-146a* and *let-7i*	↑	24613022
IBD	IECs	*miR-192*, *miR-194*, *miR-200b*, *miR-375*	↓	24613022
IBD	Serum	*miR-192*, *miR-195*, *miR-20a*, *miR-30a*, *miR-484* and *let-7b*	↑	21546856
IBD	Serum	*miR-146a*, *miR-146b*, *miR-320a*, *miR-126* and *let-7c*	↓	32793975
IBD	Colonic tissue	*miR-133a*	↓	28104982
CRC	Feces	*miR-135b*, *miR-223* and *miR-451*	↑	24841830
CRC	Feces	*miR-29a*, *miR-223* and *miR-224*	↓	26756616
CRC	Feces	*miR-421*, *miR130b-3p miR27a-3P*	↑	31622624

Notes: ↑, increase in expression; ↓, decrease in expression.

**Table 5 ijms-26-09498-t005:** LncRNAs (#) as diagnostic and therapeutic biomarkers according to the subtype of IBD and CRC.

Disease Type	Sample Type	lncRNAs	Gene Expression Change	Reference (PMID)
UC	Plasma	*Mirt2*	↓	31687015
CRC	Blood	*HOTAIR*	↑	21862635, 24583926
CRC	COAD tissue	*FHIP1A-DT*	↓	37703762
UC	Plasma	*IFNG-AS1*	↑	34970354
UC	Plasma	*ITSN1-2*	↑	32547537
IBD	Serum	*THRIL*	↑	36206229
UC	Blood	*CDKN2B-AS1*	↓	33182065
CD	Blood	*CDKN2B-AS1*	↑	30665494
UC	Colonic tissue	*H19*	↑	27661667
IBD	Colonic tissue	*CRNDE*	↑	31251902
IBD	Serum	*NEAT1*	↑	30132508
UC	Blood	*GAS5*	↑	28722800
IBD/CRC	Intestinal tissue	*MALAT1*	↑	38085149
IBD	Plasma/Tissue	*KIF9-AS1 and LINC01272*	↑	29207070
IBD	Plasma/Tissue	*DIO3OS*	↓	29207070
UC	Human intestinal epithelial Caco2 cells and murine macrophage RAW264.7 cells	*MIR4435-2HG*	↑	37597495
CRC	Colon/Rectal biopsies	*RVT1*	↑	28381186

Notes: (#) Adapted from [76], ↑, Up-regulated; ↓ Down-regulated.

**Table 6 ijms-26-09498-t006:** Epigenetic Alterations Across the Stages of Colon Disease Progression.

Condition	DNA Methylation	Histone Acetylation	Histone Methylation	Non-Coding RNAs (miRNAs/lncRNAs)	References (PMID)
Normal Colon	Homeostatic balance of methylation	Balanced H3/H4 acetylation regulating gene expression	Physiological levels of H3K4me3 and H3K27me3	Normal expression of regulatory miRNAs and lncRNAs	26220502
Inflammatory Bowel Disease (IBD)	↑ Promoter hypermethylation of anti-inflammatory genes	↑ H3K27ac	↑ H3K4me3, ↑ H3K9me3	↑ Inflammatory miRNAs (e.g., *miR-155*), altered lncRNAs	27886173, 37936149, 32464322
Colitis (IBD)	Global hypomethylation; hypermethylation of SOCS3	↓ H3/H4 acetylation; HDAC overexpression	↑ H3K9me3 (heterochromatinization); ↓ H3K27me3	↑ *miR-155*, *miR-21*; ↓ *let-7*; ↑ lncRNA *HOTAIR*	22739025, 30137272
Dysplasia	↑ CIMP phenotype, ↑ Promoter hypermethylation of MLH1, CDKN2A, APC	Loss of histone acetylation at tumor suppressor loci	↑ H3K27me3, H3K9me3 by EZH2; ↓ H3K4me3 at differentiation genes	↑ *MALAT1*, *CRNDE*, ↓ *MEG3*; ↑ oncogenic lncRNAs, miRNAs, ↓ tumor-suppressive circRNAs	16804544, 33930428
Colorectal Cancer	Global hypomethylation; ↑ Promoter CpG island hypermethylation	Aberrant HDAC recruitment; hypoacetylation at TSGs; ↓ H4K16ac	↑ H3K9me2/3; ↑ EZH2-mediated H3K27me3; ↓ H3K4me3	↑ Oncogenic miRNAs (e.g., *miR-21*, *miR-135b*); ↑ lncRNAs *NEAT1*, *PVT1*, *CCAT1*; ↓ lncRNAs like *GAS5*	15765097, 32464322

Notes: ↑, increase; ↓, decrease. Abbreviations: *TSG*, tumor suppressor gene; *HDAC*, histone deacetylase; *EZH2*, Enhancer of Zeste Homolog 2; *SOCS3*, suppressor of cytokine signaling 3. PubMed IDs (PMIDs) are provided for traceability independent of citation numbering.

**Table 7 ijms-26-09498-t007:** piRNAs, involved in colorectal cancer, showing SNP information and function, curated by piRNA databases (#) and reported in PubMed (PMID).

Name	SNP Expression Info	Function	Database (#)	PubMed (PMID)
*piR-hsa-679*	rs34383331, base change: A > T	May be involved in the development of CRC	piRBase	25740697
*piR-hsa-7400*	rs2070766f, base change: C > G	-“-	piRBase	25740697
*piR-hsa-21417*	rs2070766f, base change: C > G	-“-	piRBase	25740697
*piR-hsa-29786*	rs2070766f, base change: C > G	-“-	piRBase	25740697
*piR-hsa-21517*	rs11776042, base change: T > C	May be involved in the development of CRC	piRBase	25740697
*piR-hsa-29056*	rs9368782, base change: A > G	-“-	piRBase	25740697
*piR-hsa-2363*	rs12483859, base change: A > G	-“-	piRBase	25740697
*piR-hsa-8401*	rs10433310, base change: C > T	-“-	piRBase	25740697
*piR-hsa-3789*	rs12910401, base change: G > A	-“-	piRBase	25740697
*piR-hsa-1245*	up-regulated	It is a novel oncogene and a potential prognostic biomarker in colorectal cancer	piRBase	29382334
*piR-hsa-1282*	up-regulated	It interacts with HSF1 to promote Ser326 phosphorylation and HSF1 activation, enhancing CRC cell proliferation and suppressing cell apoptosis	piRBase	28618124
*piR-hsa-17444*	up-regulated	Formation of PIWIL2/STAT3/phosphorylated-SRC (p-SRC) complex, which activates STAT3 signaling and promotes proliferation, metastasis and chemoresistance of CRC cells	piRBase	30555542
*piR-hsa-1077* (*)	up-regulated	Ontology ID: EFO_1001951	piRNAdb, piRPheno v2.0	16751776

Notes: (*) associated with colorectal carcinoma; (#) piRBase v2.0 and piRPheno v2.0 were mainly employed for the analysis; -“- the same as above (see details in the text).

**Table 8 ijms-26-09498-t008:** Important circRNAs involved in colorectal cancer (CRC) and their characteristics (*).

circRNA Name	Synonyms	Predicted Interacting RBP (Nο of Binding Sites) (†)	Methods	Expression Pattern	PubMed ID
*hsa_circ_0020397*	*hsa_circRNA_100722*	EIF4A3(32); HuR(3); IGF2BP1(2); AGO2(2); SFRS1(1); PTB(1); LIN28A(1); IGF2BP2(1); FUS(1)	qRT-PCR; dual luciferase reporter assay; in vitro knockdown; in vitro overexpression; Western blot; etc.	up-regulated	28707774
*circ-BANP*	*hsa_circRNA_101902*; *hsa_circ_0003098*	EIF4A3(12); HuR(1); AGO2(1)	Microarray; RT-PCR; qRT-PCR; in vitro knockdown; ISH; Western blot; etc.	up-regulated	28103507
*hsa_circ_0000069*	*hsa_circRNA_100213*; *hsa_circ_001061*	EIF4A3(10); AGO2(9); IGF2BP3(7); PTB(6); IGF2BP2(6); IGF2BP1(4); HuR(2); FMRP(2); SFRS1(1); FXR2(1)	qRT-PCR; in vitro knockdown; etc.	up-regulated	28003761
*hsa_circ_001569*	N/A	N/A	in vitro knockdown; in vitro overexpression; qRT-PCR; Western blot; luciferase reporter assay, etc.	up-regulated	27058418
*hsa_circ_0001451*	*hsa_circ_001988*	EIF4A3(6); HuR(2); LIN28A(1); IGF2BP3(1); IGF2BP2(1); DGCR8(1); AGO2(1)	RNA-seq; qRT-PCR	down-regulated	26884878
*circHlPK3*	*circ_0000284*, *hsa_circ_0000284*	IGF2BP1 (5), HuR (ELAVL1) (8), FUS (4)	RNA-seq; Microarray	up-regulated	33536039
*circCCDC66*	*circ_0001313*	HuR (ELAVL1) (6), PTBP1 (7), FUS (5)	RNA-seq; Microarray; droplet digital PCR	down-regulated	33536039
*circZFR*	*hsa_circRNA_103809*; *hsa_circ_0072088*	FMRP(21); EIF4A3(7); AGO2(7); IGF2BP3(6); HuR(6); IGF2BP1(4); AGO1(4); ZC3H7B(1); U2AF65(1); PTB(1); LIN28B(1); IGF2BP2(1)	Microarray; qRT-PCR	down-regulated	28349836
*circPTK2*	*hsa_circRNA_104700*; *hsa_circ_0005273*	AGO2(5); EIF4A3(2); IGF2BP3(1); IGF2BP2(1); FUS(1)	Microarray; qRT-PCR	down-regulated	28349836
*CDR1as*	*Cdr1as*; *ciRS-7*; *hsa_circRNA_105055*; *hsa_circ_0001946*	AGO2(43); FUS(26); IGF2BP1(11); IGF2BP2(10); IGF2BP3(9); AGO1(6); TNRC6(2); TDP43(2)	qRT-PCR; in vitro overexpression; in vivo overexpression; IHC; Western blot; etc.	up-regulated	28174233

Notes: (*) predicted interactions with RBP, their expression pattern, method of detection, employed database and cited in PubMed (PMID); (†) Nο of Binding Sites are shown in parenthesis next to the name; N/A, Not/Any synonym known.

**Table 9 ijms-26-09498-t009:** Overview of Inflammation-Associated key clinical markers of ncRNAs in Colorectal Cancer (CRC).

ncRNA Type	Name	Function in CRC	References (PMID)
lncRNA	*HOTAIR*	Recruits PRC2 to silence tumor suppressor genes (e.g., CDKN1A); promotes invasion and metastasis.	21862635; 28701486
lncRNA	*MALAT1*	Enhances β-catenin nuclear translocation; regulates alternative splicing and epithelial–mesenchymal transition (EMT).	12970751; 34144008
circRNA	*circHIPK3*	Acts as a *miR-1207* sponge, which is downregulated in CRC; enhances formin-like 2 (*FMNL2*) in CRC; contributes to chemoresistance and proliferation.	32046858
circRNA	*circCCDC66*	Functions as a ceRNA; sequesters tumor-suppressive miRNAs (e.g., *miR-33b*); enhances c-MYC and YAP1 pathways.	28249903
miRNA	*miR-21*	Overexpressed in CRC; targets *PTEN*, *PDCD4*, suppressing apoptosis; regulated by NF-κB and IL-6 inflammatory stimuli.	17968323; 20797623; 34771727
miRNA	*miR-155*	Induced by NF-κB and STAT3; promotes tumor cell survival and immune evasion by targeting *SOCS1* and *TP53INP1*.	17242365; 17911593; 32702393
mRNA target	*PTEN*	Tumor suppressor inhibited by *miR-21*; regulates PI3K/AKT signaling.	32104279
mRNA target	*BCL2*	Anti-apoptotic protein regulated by multiple miRNAs (e.g., *miR-15*, *miR-16*); supports resistance to cell death.	16166262; 28984869

**Table 10 ijms-26-09498-t010:** Top Therapeutics in Development Through Epigenetic Research for Cancer.

Drug/RNA	Type	Target	Disease Phase	Function/Application Notes	Reference (PMID)
OTX015	BET inhibitor	*MYC*, *NF-κB* pathways	Phase I/II clinical trials	Solid tumors, anti-inflammatory profile	37207401
5-Azacytidine	DNMT inhibitor	DNA methylation reversal	Approved (hematologic); off-label, investigational in CRC	Reverses methylation silencing	24583822; 26317465; 33359448
Anti-miR-21 Oligonucleotides	Antisense oligo	*miR-21* suppression	Preclinical studies	Targets *PDCD4*, inhibits tumor invasion	17968323
miR-92a sponge	miRNA decoy	*miR-92a* suppression	Preclinical studies	Biomarker and therapeutic target	28957811; 33620640
Romidepsin	HDAC inhibitor	HDAC1/2 (Histone deacetylation)	Approved (T-cell lymphoma); clinical trials for CRC	Suppresses inflammation and tumor growth	27599530

## Data Availability

Further data in this study are available upon request from the authors.

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
