# Peer review of "Integrating Inflammatory and Epigenetic Signatures in IBD-Associated Colorectal Carcinogenesis: Models, Mechanisms, and Clinical Implications"

_ijms, 2025, doi:10.3390/ijms26199498_

Round 1

Reviewer 1 Report

Comments and Suggestions for Authors

The review is very informative and well written. I only have a limited amount of points the authors should address.

  1. Combine sections 2 and 4.
  2. Line 152: add innate lymphoid cells (ILC)
  3. Section 2.3: include this reference: Rios Martini et al (PMID 37749331)
  4. Line 634: replace "bacteria" with "microbes"
  5. Section 4.2: add this reference: Clay et al. (PMID: 35166235) and mention relevant microbes in the text
  6. Section 4.3: ILC have also been associated with CRC. Add these references: Schroeder et al. PMID: 35618586 and Wu et al. PMID: 35720302

Author Response

Reviewer #1

I sincerely thank the reviewer for the valuable and constructive feedback. Below, I provide detailed responses to each comment individually, and the answers are in blue fonts (Times New Roman fonts).

Points raised by the Reviewer (in black fonts)

  1. Combine sections 2 and 4.

I thank the reviewer for this significant point of improvement and constructive comment that enhances the readability of our manuscript. The sections 2 and 4 have been combined in one unified Section 2 (entitled: 2. Infectious Agents, Immune Responses, and the Inflammatory Basis of Gastrointestinal Carcinogenesis) and old sections/subsections arranged accordingly to the content of each subsection to keep relevance and logical subsection title as much as possible.

We processed this important suggestion by combining the former Sections 2 and 4 into a single broad Section 2. In doing so, we did not only merge them but also updated the titles and reorganized subsections to ensure a clear, cohesive progression of topics (infection → immune response → inflammation → carcinogenesis). This restructuring provides a cleaner narrative and keeps the scientific focus intact. Figure 2 has been reassigned under the new Section 2 (sub-section 2.5.1.) title at the end of the section 2, where its placement is now more contextually appropriate.

There are now four subsections after the merge of section 2 and 4 under the new section 2, i.e.

2.1. Infection and Host Defense Mechanisms

2.2. Pathogen Evasion and Immune Dysregulation

2.3. Infectious Agents Linked to GI Inflammation and Cancer

2.4. Chronic Inflammation as a Driver of Colorectal Carcinogenesis

In more detail the changes in headings to support contextually the new section 2 is shown below (In red fonts are the new sections/subsections replacing the old sections)

  1. The role of infectious agents in inflammation
  2. Infectious Agents, Immune Responses, and the Inflammatory Basis of Gastrointestinal Carcinogenesis

2.1. The physiology of infection: A Brief Overview

2.1. Infection and Host Defense Mechanisms

2.1.1 The Physiology of Infection: A Brief Overview

2.1.1.   Infection stages

2.1.2. Infection Stages

2.1.2.   The immune response

2.1.3. The Immune Response

2.1.3. Factors influencing the outcome of the infection

2.1.4. Factors Influencing the Outcome of the Infection

2.2.  Physiological Human-Bacteria Interactions

2.1.5. Physiological Human-Bacteria Interactions

2.3.  Antigens and Subversion of Immune Responses

2.2. Pathogen Evasion and Immune Dysregulation

2.2.1. Antigens and Subversion of Immune Response

2.4. Bacterial Infection and Immune Dysregulation

2.2.2. Bacterial Infection and Immune Dysregulation

2.5.Gastrointestinal tuberculosis

2.3. Infectious Agents Linked to GI Inflammation and Cancer

2.3.1. Mycobacterium tuberculosis and GI Inflammation

2.6. Colonic Tuberculosis + 2.7. Gastrointestinal tuberculosis in animals

2.3.2. Other Bacterial Drivers of GI Carcinogenesis

  1. Immune Defense in Gastrointestinal (GI) Chronic Inflammation and Carcinogenesis

2.5. Mechanistic and Historical Perspectives on Inflammation-Induced Cancer

4.1. Inflammation-induced cancers associated with the gastrointestinal tract.

2.5.1. Inflammation-Induced Cancers Associated with the GI Tract

4.2. The involvement of bacteria in the mechanisms of carcinogenesis

2.3.3. The Involvement of Microbes in the Mechanisms of Carcinogenesis

4.4.. Effect of bacterial infection on gastrointestinal cancer

2.3.4. Effect of bacterial infection on gastrointestinal cancer

4.4.1. Colorectal cancer

2.3.5..Colorectal cancer

4.3. Chronic Inflammation as a Driver of Colorectal Carcinogenesis.

2.4. Chronic Inflammation as a Driver of Colorectal Carcinogenesis

All the following correction/additions follow the new section numbering system after the combination of sections 2 and 4 in one (section 2) and are located from line 102 – line 608 (please refer to the corrected manuscript).

Thereby, the present Figure 1 (“Mechanistic Pathway – From Inflammation to Colorectal Carcinogenesis”) is located in subsection 2.5.1. and took the number of former Figure 1 (“Sidebar Infographic: Commonly Used Animal Models for Studying Inflammatory Bowel Disease and CAC”) which is now has remained in section 3 but changed number to Figure 2.

The internal corrections/changes suggested by the reviewer(s), within the whole manuscript, including section 2, are annotated in red fonts for visibility as well as all the other suggestions suggested by the reviewer(s), which are shown below.

  1. Line 152: add innate lymphoid cells (ILC)

Apologies to the Reviewer 1 that ILCs are missing from this manuscript and I thank the reviewer very much for the addition of this important family (ILCs) important regulators of mucosal immunity and inflammation. The additional paragraphs associated with the Rios Martini et al reference are highlighted with red fonts, in subsection 2.1.2. in the beginning of the subsection, at lines 143-145 (please refer to the corrected manuscript).

  1. Section 2.3: include this reference: Rios Martini et al (PMID 37749331)

I thank the reviewer for this important reference. The Rios Martini et al reference 26, and the additional sentences associated with the above reference are highlighted with red fonts, in the new subsection 2.2.1. in 2 specific areas, in order to introduce a contextual meaning and keep the text flow. They are located at lines 219-222  and 235-237 (please refer to the corrected manuscript).

.

  1. Line 634: replace "bacteria" with "microbes"

Thank you for spotting this narrow term located in the old subsection

4.2. The Involvement of Bacteria in the Mechanisms of Carcinogenesis. Now in the new subsection (2.3.3.) has been corrected as:

2.3.3. The Involvement of Microbes in the Mechanisms of Carcinogenesis (please refer to the corrected manuscript), which is located at line 315.

  1. Section 4.2: add this reference: Clay et al. (PMID: 35166235) and mention relevant microbes in the text

I thank the reviewer also for this important reference. The additional reference 51 and the sentence associated with the Clay et al reference are highlighted with red fonts, in the new subsection 2.3.3. in 3 specific areas, at lines 321, 344 and 367-368) (please refer to the corrected manuscript).

  1. Section 4.3: ILC have also been associated with CRC. Add these references: Schroeder et al. PMID: 35618586 and Wu et al. PMID: 35720302

.

I finally thank the reviewer for these important references. The additional references 59 and 64 and paragraphs associated with the Schroeder et al., 2022 and Wu et al., 2022 references, are highlighted with red fonts, in subsection 2.4. in 4 specific areas, also introducing small paragraphs pertinent to the references in order to keep the flow with relevant content, which are shown at lines 422-427, 441-442, 448-452 and 455-458 (please refer to the corrected manuscript).

Please note that all new additions/corrections/modifications in the manuscript are shown in red fonts, while the blue paragraphs were rephrased to avoid unintended plagiarism as requested.

Thank you

Thank you

Yours sincerely,

Kostas A. Triantaphyllopoulos

Reviewer 2 Report

Comments and Suggestions for Authors

Enclosed in the pdf are my revision! Thank you!

Author Response

Reviewer #2

I sincerely thank the reviewer for the thoughtful and constructive comments. I have addressed each of these points individually below, and the answers are in blue fonts (Time New Roman fonts).

Points raised by the Reviewer (in black fonts)

  1. Ensure consistent terminology: IBD-associated CRC/CAC (pick one convention and use throughout)

I sincerely thank the reviewer for this important and constructive comment regarding terminology: IBD-associated CRC.

This dual mention basically reflects the scope of the review, which integrates evidence from preclinical CAC models (e.g., AOM/DSS) alongside clinical CRC findings, ensuring clarity for both experimental and translational audiences.

To ensure clarity and consistency, we have now adopted the terminology IBD-associated CRC throughout the manuscript when referring to the human disease, while specifying CAC only when explicitly describing findings derived from preclinical models. This distinction allows us to maintain precision without ambiguity for readers.

Thank you.

  1. Standardize symbols and punctuation: TNF-α (not “á”), NF-κB (not “êB”); one comma: “DNA methylation, Histone modifications”; fix duplicated commas (Line 34-“methylation,, Histone”).

I thank the reviewer’s constructive comments concerning the symbols and punctuation

We have corrected the “êB” in NF-κB. TNF-α with Greek alpha. I have also fixed the duplicated commas in line 34 (line remained the same) and checked all the manuscript.

  1. Reduce redundancy; merge overlapping sentences about innate/adaptive responses and chronic inflammation leading to cancer.

I thank the reviewer’s constructive comments concerning the merge of overlapping sentences in the old sections 2 & 4 regarding the innate/adaptive responses and chronic inflammation leading to cancer. In response, the former Sections 2 and 4 have been consolidated into a unified Section 2. During this revision, the overlapping sentences on innate/adaptive responses and chronic inflammation leading to cancer were eliminated to improve clarity and flow.

  1. Keep journal style for abbreviations on first use and maintain uniform reference formatting per IJMS

I thank the reviewer for highlighting this important point. Abbreviations have now been defined consistently at first use and checked carefully for compliance with IJMS style. In addition, all references have also been reviewed and reformatted to ensure uniformity according to journal guidelines. Thank you

  1. Define all abbreviations at first use; keep section headings parallel.

I thank the reviewer for this valuable suggestion to improve readability. All abbreviations are now defined at their first appearance and used consistently thereafter. Section headings have also been revised to follow a parallel structure for improved coherence. Thank you

  1. A strong, well-referenced review with high translational value. With minor revisions, principally scope tightening around IBD/CAC, added mechanistic clarity for epigenetic regulators/ncRNAs, consolidation of the socioeconomic section, and formatting clean-up, the manuscript will be suitable for publication.

I thank the reviewer for the positive evaluation and constructive recommendations. In line with the suggestions, the scope has been tightened around IBD/CAC, mechanistic explanations for epigenetic regulators and ncRNAs have been clarified, and the socioeconomic section has been consolidated. We also refined formatting throughout, and we trust that the revised manuscript is now substantially improved.

Yours sincerely,

Kostas A. Triantaphyllopoulos

Reviewer 3 Report

Comments and Suggestions for Authors

The current manuscript surveys how chronic intestinal inflammation (NF-κB/STAT3 cytokine circuits) intersects with epigenetic programs (DNA methylation, histone marks, ncRNAs) to drive the IBD, dysplasia, CAC continuum, tying mechanisms to translational angles.
It catalogs animal models used to study IBD/CAC (DSS, AOM/DSS, TNBS/oxazolone, IL-10⁻/⁻, TRUC, APC^Min/+, T-cell transfer), claiming an internal RNA-seq meta-analysis comparing DSS/AOM-DSS mice with human UC/CRC, and outlines diagnostic/prognostic epigenetic & RNA biomarkers (e.g., methylated SEPT9/VIM/SFRP2; miR-21/92a/135b panels; stool DNA platforms). In general, the topic relevance is high but originality is somehow moderate since its novelty hinges on the promised cross-species RNA-seq “meta-analysis,” which is referenced as “manuscript in preparation” without methods/results here, so the paper’s incremental contribution remains unsubstantiated in its current form. The review repeatedly leans on an internal meta-analysis to motivate conserved signatures/biomarkers but provides no data, methods, or reproducible details. That undercuts the manuscript’s unique “integration” promise. The inclusion of off-topic infection/TB content is somehow misleading and should be considered for removal. Besides, several translational statements (diagnostic test status; “high sensitivity” panels) need up-to-date, primary validation sources and precise wording. Tighten sourcing around any strong clinical or regulatory claims.

Author Response

Reviewer #3

I sincerely thank the reviewer for the thoughtful and constructive comments. I have addressed each of these points individually below, and the answers are in blue fonts (Times New Roman fonts).

Points raised by the Reviewer (in black fonts)

The current manuscript surveys how chronic intestinal inflammation (NF-κB/STAT3 cytokine circuits) intersects with epigenetic programs (DNA methylation, histone marks, ncRNAs) to drive the IBD, dysplasia, CAC continuum, tying mechanisms to translational angles.
It catalogs animal models used to study IBD/CAC (DSS, AOM/DSS, TNBS/oxazolone, IL-10⁻/⁻, TRUC, APC^Min/+, T-cell transfer), claiming an internal RNA-seq meta-analysis comparing DSS/AOM-DSS mice with human UC/CRC, and outlines diagnostic/prognostic epigenetic & RNA biomarkers (e.g., methylated SEPT9/VIM/SFRP2; miR-21/92a/135b panels; stool DNA platforms). In general, the topic relevance is high but originality is somehow moderate since its novelty hinges on the promised cross-species RNA-seq “meta-analysis,” which is referenced as “manuscript in preparation” without methods/results here, so the paper’s incremental contribution remains unsubstantiated in its current form.

1) The review repeatedly leans on an internal meta-analysis to motivate conserved signatures/biomarkers but provides no data, methods, or reproducible details. That undercuts the manuscript’s unique “integration” promise.

2) The inclusion of off-topic infection/TB content is somehow misleading and should be considered for removal.

3) Besides, several translational statements (diagnostic test status; “high sensitivity” panels) need up-to-date, primary validation sources and precise wording. Tighten sourcing around any strong clinical or regulatory claims.

I would like to sincerely thank Reviewer 3 for their constructive comments. I address each point individually below. Please note that the following division of responses is intended solely to separate the different suggestions and comments, thereby ensuring a clear and comprehensive reply to all questions raised. Accordingly, I provide specific answers to each of the reviewer’s remarks.

1)

We thank the Reviewer for this observation. We agree that our RNA-seq meta-analysis work is still unpublished and therefore cannot be fully detailed in the present review, since including raw methods or results here would compromise the originality and eligibility of our upcoming research article. This practice is consistent with how review articles often contextualize very recent findings without disclosing full datasets.

The references to these results are limited (mainly in subsection 7.5., to highlight the translational relevance and value of mouse models, reasoning this selection in our research) and are not the main basis of the manuscript, which remains grounded in published peer-reviewed data. To avoid ambiguity, we have revised the text by replacing “manuscript in preparation” with “unpublished results,” which more accurately reflects their status. We were also open to removing redundant mentions and therefore reduced them to the minimum providing only details in subsection 7.5.

Importantly, the novelty of the current review does not rely on unpublished data but on the integration of published evidence and database-driven insights. For example, in subsection 7.3 we expanded the discussion using literature-based analyses e.g. ncFAN v2.0, which revealed key lncRNA–miRNA interactions (e.g., GAS5-AS1, FIGNL2-DT, GOLGA2P8) or analyzed DNA methylation profiles of cancer-associated lncRNA loci for novel epigenetic signatures or used circRNAs analysis with the exoRBase 2.0 biotool, in GI inflammation and tumorigenesis etc, suggesting that these results may represent novel layer of post-transcriptional controls in the inflammation-to-cancer transition, illuminating possibly hidden regulatory switches in IBD-associated CRC. Certainly these additions require further validation in laboratory and translational settings, while their inclusion adds conceptual depth to the review. We, the authors, believe this comprehensive study demonstrates that our integrative framework goes beyond the unpublished work, which we have synthesized into the narrative and distilled into comprehensive visual figures to provide a simplified yet translational perspective.

2)

We thank the Reviewer for raising this concern. We respectfully note that gastrointestinal tuberculosis and related Mycobacterium species infections are not off-topic. They directly exemplify the link between pathogens, chronic intestinal inflammation, and carcinogenesis, which is the predominant theme of our review. This inclusion is intended to illustrate the broader infectious contributors to GI inflammation and highlight how different infectious agents may initiate immune dysregulation pathways relevant to CAC.

Nevertheless, to improve clarity and focus, we have now reorganized the structure of the manuscript (Sections 2 and 4 combined into a single streamlined Section 2, as requested by another Reviewer), making the role of pathogens in inflammation-driven carcinogenesis more explicit.

Therefore, we have included Mycobacterium sp. infections and tuberculosis subtypes affecting the GI system in the subsection 2.3.2. Other Bacterial Drivers of GI Carcinogenesis. We believe this revised presentation resolves the Reviewer’s concern by ensuring that infection-related subsections are more tightly integrated into the central narrative.

3)

We thank the Reviewer for pointing out the need for precision in translational claims. We have carefully re-checked all statements concerning diagnostic platforms (including methylated DNA markers such as SEPT9, VIM, and SFRP2, as well as miRNA panels and stool DNA tests). Each of these is now clearly supported by updated primary sources, including recent validation studies and regulatory status reports where available.

Where strong claims such as “high sensitivity” were made, we have revised the wording to be precise and consistent with the cited evidence, e.g., by specifying reported sensitivity/specificity ranges and the type of validation (clinical trial, FDA-approved platform, etc.). This ensures the translational section remains both accurate and evidence-based.

Thank you

Yours sincerely,

Kostas A. Triantaphyllopoulos

Round 2

Reviewer 1 Report

Comments and Suggestions for Authors

The authors addressed all the points I raised, and now I am happy to endorse the manuscript for publication.